# MIX-MAXENT: CREATING HIGH ENTROPY BARRIERS TO IMPROVE ACCURACY AND UNCERTAINTY ESTIMATES OF DETERMINISTIC NEURAL NETWORKS

## ABSTRACT

We propose an extremely simple approach to regularize a single deterministic neural network to obtain improved accuracy and reliable uncertainty estimates. Our approach, on top of the cross-entropy loss, simply puts an entropy maximization regularizer corresponding to the predictive distribution in the regions of the embedding space between the class clusters. This is achieved by synthetically generating between-cluster samples via the convex combination of two images from *different* classes and maximizing the entropy on these samples. Such a data-dependent regularization guides the maximum likelihood estimation to prefer a solution that (1) maps out-of-distribution samples to high entropy regions (creating an entropy barrier); and (2) is more robust to the superficial input perturbations. Via extensive experiments on real-world datasets (CIFAR-10 and CIFAR-100) using ResNet and Wide-ResNet architectures, we demonstrate that Mix-MaxEnt consistently provides much improved classification accuracy, better calibrated probabilities for in-distribution data, and reliable uncertainty estimates when exposed to situations involving domain-shift and out-of-distribution samples.

## 1 INTRODUCTION

A particularly thriving sub-field of research in Deep Neural Networks (DNNs) concerns devising efficient approaches towards obtaining reliable predictive uncertainty. NNs are known to be overconfident for both, in- and out-distribution samples (i.e. for samples coming from the same distribution from which the training distribution has been sampled (IND samples) and for samples not coming from such distribution), leading to highly unreliable uncertainty estimates. They can be wrong with very high confidence (Guo et al., 2017) not only on the test data similar to the one they have been trained on, but also when facing previously-unseen conditions (Taori et al., 2016). The overconfidence problem becomes even more concerning when just slight changes in illumination, atmospheric conditions or in the image capturing process (domain-shift) can severely damage the actual accuracy of the model (Taori et al., 2016). A desirable property of any model is to be robust to such superficial changes that do not affect the label of the classified image, and to become uncertain (or indecisive) when exposed to samples from a distribution different from the training distribution.

While the literature suggests that classifiers whose predictive distributions increase their entropy the further away the test input gets from the training data are desirable, the implementation of models satisfying such a property is not scalable, and can only occur at the cost of performing crude approximations and by non-trivial modifications to the architecture of the neural network (Liu et al., 2020a; Kristiadi et al., 2020). Such modifications sometimes lead to degraded accuracy.

Our motivation behind this work is based on the following observations. (1) We find that a trained network (DNN) do not project out-of-distribution (OOD) and domain-shifted (DS) samples **arbitrarily away** from the training data. In fact, all the OOD and domain-shifted inputs that we considered (detailed discussion in Section 3) are projected within the smallest hypersphere that contains all of the IND test data. Therefore, it might not be necessary to enforce the network to be uncertain *everywhere* away from the data, and perhaps, in-distribution data can be used to mimic the regions where OOD and DS samples are being projected. (2) DNNs tend to embed OOD and DS samples in high confidence (low-entropy) regions (see Figure 1). This contradicts the desired ideal behaviour (Liu

et al., 2020a; Pereyra et al., 2017), that would require a model to map such samples in high-entropy regions.

Given the observations above, we propose a simple entropy maximization regularizer (called *Mix-MaxEnt*), that induces a high entropy barrier between class clusters, while the cross-entropy optimization objective keeps the entropy low close to the class clusters. The entropy regularizer is enforced for samples *synthesized* using the convex combination of a pair of samples from two **different** classes of the in-distribution training data. When combined with the cross-entropy loss, this regularizer prefers a maximum likelihood solution such that the uncertainty of the network increases while moving away from the embeddings of one class in the direction of another, and learns features that are robust to local input perturbations.

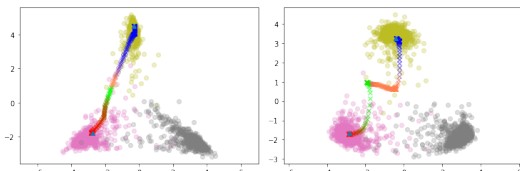

Figure 1: **DNN (Left), Ours (Right)**. Interpolation experiment on CIFAR10 to show embeddings of the linear interpolation of two randomly picked input samples from class 1 (purple) and class 2 (yellow). Red and green samples are classified as class 1, and orange and blue samples as class 2. As the color changes from red to green, the predictive entropy increases. Same for the color change from blue to orange. Note, DNN classifies interpolated points with very high confidence (low entropy) even if the samples shift drastically from the data. However, Mix-MaxEnt maps these samples into a wide high-entropy region. Details of this visualization are provided in Appendix F. More exhaustive visualizations of this phenomenon are provided in Section 5.2.2.

Through extensive experiments using WideResNet28-10 and ResNet50 architectures on CIFAR10 and CIFAR100 datasets, we demonstrate that our method outperforms *all* existing single model baselines in providing clean data accuracy. On Domain-Shift experiments, it provides remarkably improved accuracy compared to all the baselines including Deep Ensembles (DE) (Lakshminarayanan et al., 2017). For instance, it provides $4.8\%$ and $4.7\%$ improvements over the highly competitive DE and SNGP (Liu et al., 2020a), respectively, on CIFAR-10 using WideResNet. In terms of reliable uncertainty estimates, it is either the best or, in a few cases, extremely competitive with respect to the best performing one, with only slightly inferior performance. Overall, our experiments show that Mix-MaxEnt is by far the best performing one compared to the existing single model approaches.

We would like to highlight that, along with its effectiveness, one of the core strengths of our approach is its simplicity. As opposed to the recently proposed SNGP and DUQ (van Amersfoort et al., 2020), it does not require any modifications to the architectures and does not trade accuracy in order to improve uncertainty estimates. And, as opposed to the extremely competitive DE, it is a single deterministic model, hence, extremely efficient.

## 2 RELATED WORKS

Modern NNs have been shown to be miscalibrated (Guo et al., 2017), implying, a mismatch between model's confidence and its accuracy. While recent approaches have been proposed to fix this issue on in-distribution data (Guo et al., 2017; Mukhoti et al., 2020; Lakshminarayanan et al., 2017), such calibrated models do not necessarily behave as well under domain-shift (Ovadia et al., 2019).

**Uncertainty estimation methods** The current state-of-the-art in producing reliable uncertainties is the extremely expensive Deep Ensembles (Lakshminarayanan et al., 2017), whose training and inference costs scale linearly with the number of members in the ensemble. This significantly limits its applicability in real-world applications, where time and compute are of the essence. But even in this case, the accuracy and calibration dramatically drop when facing data-shift. A current research trend is to either emulate deep ensemble behaviour while avoiding its disadvantages (Havasi et al., 2021; Wen et al., 2020; Huang et al., 2017) ) or combining it with other methods (Ashukha et al., 2020; Rahaman & Thiery, 2020; Wen et al., 2021).

An alternative approach to obtain a reliable predictive distribution is to rely on Bayesian NNs (Chen et al., 2014; Zhang et al., 2020; Durmus et al., 2016). However, obtaining the exact posterior for NNs is computationally infeasible, for this reason, it is necessary to rely on approximate inference schemes whose effectiveness is in question (Hron et al., 2018; Foong et al., 2020). A recent trend in

literature has tried reducing the application of Bayesian inference only to the last layers to get the benefit of the effectiveness of training deterministic networks with standard training procedures, and the improved uncertainty estimation provided by Bayesian approaches (*e.g.* SNGP (Liu et al., 2020a), KFAC-LLLA (Kristiadi et al., 2020), BM (Joo et al., 2020), DUQ (van Amersfoort et al., 2020)).

Unfortunately, many of these models often trade accuracy (due to changes in architecture or training procedures) in exchange of marginal improvements in uncertainty estimation, and are hard to use in a plug-and-play fashion since they are extremely sensitive to hyperparameter tuning.

**Learning better features via augmentation or normalisation**  It has been shown (Shah et al., 2020) that if NNs can identify simple (spurious) features to separate training data, they will prefer those to semantically meaningful features. NNs also heavily rely on textures (Geirhos et al., 2018). This agrees with recent literature, that has shown self-supervision (Winkens et al., 2020; Hendrycks et al., 2019) or simple augmentation techniques like mixup (Thulasidasan et al., 2019) can produce representations that yield better uncertainty estimation. A way to make NNs insensitive to noise is imposing constraints on their sensitivity. An extensive literature studies how controlling Lipschitzness impacts on the robustness and generalization performance (Bartlett et al., 2017; Neyshabur et al., 2018; Yoshida & Miyato, 2017; Gouk et al., 2020; Arora et al., 2018).

## 3 ANALYSIS AND OBSERVATIONS

Consider the supervised task of learning a predictive distribution $p(y|\mathbf{x}; \theta)$, where $y \in \{1, \cdots, K\}$ is the set of K-classes, $\mathbf{x} \in \mathcal{X} \subset \mathbb{R}^p$ is the space of input data, and $\theta$ respresents the parameters of the underlying model (*e.g.* a neural network). To obtain an optimal $\theta$, the standard approach is to collect a training data set $\mathcal{D} = \{\mathbf{x}_i, y_i\}_{i=1}^n$ , and maximize the log-likelihood $p(y|\mathbf{x} \in \mathcal{X}; \theta)$ under $\mathcal{D}$. Therefore, the nature of the predictive distribution heavily relies on the training data and the model under consideration. In practice, however, because of the high dimensionality of the input space $\mathcal{X}$, the training data $\mathcal{D}$ is normally collected from a subset $\mathcal{X}_I \subset \mathcal{X}$, where $\mathcal{X}_I$ denotes the in-domain data manifold. Let us denote the out-domain data manifold as $\mathcal{X}_O = \mathcal{X} \setminus \mathcal{X}_I$.

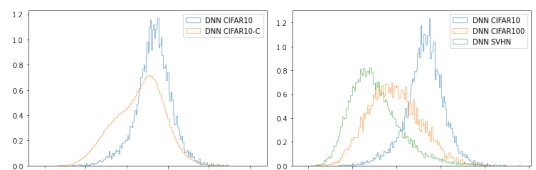

Figure 2: Histograms of the Euclidean distance of the embeddings (model trained on CIFAR-10). (Left) CIFAR-10 vs CIFAR-10-C. (Right) CIFAR-10 (IND) vs CIFAR-100 (OOD) and SVHN (OOD). Note, both corrupted and OOD embeddings lie within the hypersphere $\mathcal{S}$ as their histograms are left shifted relative to the histogram of CIFAR-10. Also, the SVHN histogram is more shifted towards the center than that of CIFAR-100, showing an implicit ordering.

**Where do neural networks map $\mathcal{X}_O$?**  Modern softmax-based neural classifiers operate under two essential assumptions: (1) a closed-world assumption (Bendale & Boult, 2015); (2) an implicit clutering assumption (Hess et al., 2020; Lee et al., 2018). The closed-world assumption implies that the classifier must select one out of $K$ classes fixed at training time, even if the input belongs to none of them. The clustering assumption depends on the presence of the softmax layer: the authors of (Hess et al., 2020) have shown that softmax neural classifiers perform K-means clustering (which, we recall, can be interpreted as a limit subcase of Expectation-Maximisation under Gaussian Mixture Model (GMM) assumptions (Hastie et al., 2001; Bishop, 2006)) with $K$ centroids at equal distance from the center of the embedding space. The authors of (Lee et al., 2018) observed the embeddings of modern neural network do empirically adapt well to GMM assumptions[1]. Following the above insights, we consider a WideResNet (WRN) trained on CIFAR-10 ($\mathcal{X}_I$) and obtain the smallest hypersphere $\mathcal{S}$ in the feature space that contains all the in-domain test samples. Note, as already mentioned in Hess et al. (2020), we also found the mean of the embeddings of $\mathcal{X}_I$ to be very close to a zero vector, therefore, $\mathcal{S}$ is practically centred at zero. Given $\mathcal{S}$, we would like to understand where do $\mathcal{X}_O$ (CIFAR-100 and SVHN) and data-shifted samples (corrupted CIFAR-10 test samples denoted as CIFAR-10-C) lie with respect to $\mathcal{S}$. These are our observations:

---

[1]It is important to observe that, given this empirical evidence, one would think that using the log-probability of a GMM could be an optimal uncertainty measure for this task. However, in Appendix E we show that the high-dimensional nature of the latent space makes the computation numerically unstable.

- The NN maps all (except one) out-of-distribution ($\mathcal{X}_O$) and data-shifted samples (corrupted) inside the hypersphere $\mathcal{S}$ that was obtained using the test data belonging to $\mathcal{X}_I$.
- There exist an implicit ordering in the sense that CIFAR-100 is mapped closer to CIFAR-10 than SVHN. This is clearly shown in the histogram (refer Figure 2) of the Euclidean distances of the OOD and data-shifted embeddings from the center of the hypersphere $\mathcal{S}$. This behaviour can be explained by the fact that convolution kernels are learned using $\mathcal{X}_I$, therefore, will produce stronger activations if the similarity between the kernels learned on $\mathcal{X}_I$ and the features present in the input is high. On the other hand, dissimilarity between kernels and features present in the input will produce weaker activations, thus driving the embeddings closer to the origin.

**Forcing classifiers to be uncertain where it truly matters**  Recall that an ideal classifier's predictive distribution $p(y|\mathbf{x} \in \mathcal{X}; \theta)$ would assume the following form

$$\underbrace{p(y|\mathbf{x} \in \mathcal{X}_I; \theta)p(\mathbf{x} \in \mathcal{X}_I)}_{\textit{maximum likelihood estimate}} + \underbrace{p(y|\mathbf{x} \in \mathcal{X}_O; \theta)p(\mathbf{x} \in \mathcal{X}_O)}_{\textit{unknown during training}}.$$

Since $\mathcal{X}_O$ is unknown during the maximum likelihood estimation (MLE) of $\theta$, the model is completely unaware of the second part of the predictive distribution that depends on the presence of $\mathcal{X}_O$. Therefore, in order for the model to be indecisive for out-of-distribution and domain-shifted samples (assuming they are away from the in-distribution data), it is desirable that the model's uncertainty increases proportionally to the test sample distance[2] from the training data $\mathcal{X}_I$. This is the prime motivation behind the recent work called SNGP (Liu et al., 2020a) where the uncertainty estimates of a NN is being improved by taking inspirations from the classic Gaussian Processes literature (Rasmussen & Williams, 2005). While techniques like SNGP (Liu et al., 2020a) and KFAC-LLLA (Kristiadi et al., 2020) try to increase the entropy *everywhere* away from the training data, they need to resort to crude approximations of the theoretical models that provide such property (both) and also have to modify the network's architecture (SNGP). Most approaches also compromise with the training accuracy because of such approximations.

Given the observations from our analysis, we find that the OOD and domain-shifted samples are mapped to a specific embedding space $\mathcal{S}$ where the network tends to concentrate all its embeddings. This indicates that correcting its uncertainty estimates within this space $\mathcal{S}$ could be sufficient to achieve better performance, rather than trying to increase entropy everywhere away from the training data (i.e. mostly in regions where the network never projects its inputs). This is precisely the motivation behind our work. In what follows, we present an extremely simple approach to regularize the embedding space $\mathcal{S}$ such that entropy barriers are being created between different classes.

## 4 Methodology

**Mix-MaxEnt**  Motivated by the evidence we collected, we define a simple regularization approach to make the maximum likelihood estimate aware (increasing the uncertainty and becoming less confident) of what is unknown to it during the training by simply putting a maximum entropy regularizer (Pereyra et al., 2017) on synthetically generated samples that leverage the knowledge of $\mathcal{X}_I$ to drift towards $\mathcal{X}_O$. Specifically, since $\mathcal{X}_O$ is unknown during the training, we take a simple approach where we synthetically create samples $\bar{\mathbf{x}} \in \mathcal{X}$ as follows:

$$\bar{\mathbf{x}} = \lambda_0 \mathbf{x}_i + (1 - \lambda_0)\mathbf{x}_j, \;\; if \; y_i \neq y_j. \tag{1}$$

Here, $\{\mathbf{x}_i, \mathbf{x}_j\} \in \mathcal{X}_I$, and $\lambda_0 \sim \texttt{Beta}(\alpha, \beta)$, where $\alpha$ and $\beta$ are the parameters of the beta distribution. Note, we only choose pair of samples belonging to two *different classes* ($y_i \neq y_j$), so that $\bar{\mathbf{x}}$ will mix features mimicking an off-data manifold image that has intermediate properties between those of each class. We further ensure this by picking $\alpha = \beta >> 1$ as, in this case, the beta distribution will have the peak in the middle and the sharpness of the peak increases as $\alpha$ grows. Let us call the collection of such synthetic intermediate samples $\bar{\mathcal{X}}_O$. Then, our final objective function adds to the usual cross-entropy optimisation term, an entropy maximisation term on such samples:

$$\min_{\theta} -\log p(y|\mathbf{x} \in \mathcal{X}_I; \theta) - \mathcal{H}_{\bar{y}}(p(.|\mathbf{x} \in \bar{\mathcal{X}}_O; \theta)), \tag{2}$$

---

[2]Note, although the theory assumes Euclidean distance, it is arguable this distance notion reflects perceptual dissimilarity in the image space.

where, $\mathcal{H}_{\bar{y}}(.)$ is the entropy defined over the label support set $\bar{y} = \{y_i, y_j\}$. It is important to observe that since the input image will contain features coming from the inputs of the two classes, it is reasonable to maximise the entropy only over such classes while keeping the probability assigned to the other classes to zero (since no input features endorses the presence of these classes). Indeed, in our experiments we show that maximising the entropy over all the classes severely degrades the performance while making the network underconfident. Since we increase the entropy only for mixed (interpolated) samples from different classes, we call our approach **Mix-MaxEnt**.

**Mix-MaxEnt vs mixup (Zhang et al., 2018)**    Though the sample interpolation in our approach is similar to the one presented in *mixup*, there are fundamental differences between the two. The mixup training goal is to reduce memorization by training around the vicinity of the samples (vicinal risk minimization). Effectively, it *slightly perturbs* the data and uses its perturbed version for the log-likelihood estimation. It optimises $-\log p(\tilde{y}|\bar{\mathbf{x}})$ where $\tilde{y} = \lambda_0 y_i + (1-\lambda_0)y_j, \bar{\mathbf{x}} = \lambda_0 \mathbf{x}_i + (1-\lambda_0)\mathbf{x}_j$, and $y_i$ and $y_j$ might be the same. Practically speaking, a small value of Beta distribution parameters ($\alpha = \beta < 1$) is typically chosen for *mixup* (Zhang et al., 2018; Thulasidasan et al., 2019; Wen et al., 2021), thus, $\lambda_0$ in this case is either $\approx 0$ or $\approx 1$. Because of this, one of the two interpolating samples heavily dominates the interpolated sample $\bar{\mathbf{x}}$. Note, this is necessary for *mixup* as otherwise the interpolated sample distribution would be different from the original data-distribution and, because of the absence of the log-likelihood term over the clean samples (as in Mix-MaxEnt), there will be a drop in the performance at test time.

However, Mix-MaxEnt (refer Eq (2)) uses *clean* unperturbed samples for the log-likelihood estimation while *interpolated samples* are being used to regularize the network so that it prefers a solution having high-entropy at regions in-between samples of different classes. Since the purpose of the interpolated samples here is only to guide the log-likelihood solution, thus, as opposed to mixup, a high interpolation factor can be used in this case. Hence, we use $\lambda_0 \approx 0.5$ to interpolate pair samples from different classes so that the interpolated sample mimic points from outside the data-manifold for entropy maximization.

## 5 EXPERIMENTS

**Training Datasets and Network Architectures**    We employ WideResNet28-10 (WRN) (Zagoruyko & Komodakis, 2016) and ResNet50 (RN50) (He et al., 2016) architectures that have been shown to produce state-of-the-art classification accuracies on real-world datasets. We train them on CIFAR-10 (C10) and CIFAR-100 (C100). For **Domain-Shift** experiments, we resort to the widely used CIFAR10-C and CIFAR100-C, corrupted versions of C10 and C100 (Hendrycks & Dietterich, 2019). For **Out-of-Distribution** detection experiments, following SNGP (Liu et al., 2020a), we use C100 and SVHN as OOD for models trained on C10. Similarly, for models trained on C100, we use C10 and SVHN as OOD.

**Methods considered for comparisons**    We consider both deterministic and Bayesian approaches for comparison. Following (Liu et al., 2020a), we also create two additional strong and simple baselines where a ResNet is enforced to be bi-Lipschitz using Spectral Normalization (SN) (Miyato et al., 2018a) and Stable Rank Normalization (SRN) (Sanyal et al., 2020). Note, we are the first to consider SRN for these experiments as it induces more compact clusters in the feature space than SN (we provide a simple mathematical proof of this in the Appendix A). Therefore, we compare our approach with the following baselines:

- DNN: Standard deterministic neural network trained using cross-entropy loss.
- DNN-SN:DNN with SN (Miyato et al., 2018a).
- DNN-SRN: DNN with SRN (Sanyal et al., 2020).
- SNGP: Spectrally Normalized Gaussian Process  (Liu et al., 2020a).
- DUQ: Deterministic Uncertainty Quantification (van Amersfoort et al., 2020).
- Mixup: Standard Mixup training (Zhang et al., 2018).
- KFAC-LLLA: KFAC-Laplace Last Layer Approximation (Kristiadi et al., 2020). A method that makes a model Bayesian at test time by taking Laplace approximation of the last layer using a Kronecker-Factored approximation (Ritter et al., 2018). For the sake of completeness, we provide a simple outline of this approach in Appendix B.

| | Clean Data | | | Domain-Shift | | | Out-of-Distribution | | | |
|---|---|---|---|---|---|---|---|---|---|---|
| | CIFAR100 (Test) | | | CIFAR100-C | | | CIFAR10 | | SVHN | |
| Methods | Accuracy (↑) | ECE (↓) | AdaECE (↓) | Accuracy (↑) | ECE (↓) | AdaECE (↓) | AUROC (↑) | AUPR (↑) | AUROC (↑) | AUPR (↑) |
| DNN | 81.58 ± 0.13 | 3.88 ± 0.25 | 3.84 ± 0.24 | 52.54 ± 0.31 | 9.96 ± 0.21 | 9.94 ± 0.21 | 81.06 ± 0.29 | 77.35 ± 0.39 | 79.68 ± 4.81 | 88.46 ± 2.53 |
| DNN-SN | 81.60 ± 0.15 | 3.94 ± 0.23 | 3.81 ± 0.21 | 52.61 ± 0.23 | 11.62 ± 0.41 | 11.59 ± 0.41 | 81.10 ± 0.35 | 77.34 ± 0.19 | 83.43 ± 3.63 | 91.01 ± 2.05 |
| DNN-SRN | 81.38 ± 0.23 | 3.82 ± 0.27 | 3.71 ± 0.26 | 52.54 ± 0.17 | 11.04 ± 0.77 | 11.00 ± 0.78 | 81.26 ± 0.18 | 77.36 ± 0.30 | 85.51 ± 1.18 | 91.84 ± 1.12 |
| Deep Ensembles | **83.85 ± 0.13** | 3.31 ± 0.12 | 3.29 ± 0.08 | 55.58 ± 0.14 | 12.43 ± 0.13 | 12.36 ± 0.15 | **83.26 ± 0.14** | **79.82 ± 0.27** | 85.07 ± 1.58 | 91.65 ± 0.97 |
| SNGP | 79.20 ± 0.21 | 1.95 ± 0.25 | 1.94 ± 0.28 | 57.23 ± 0.25 | 10.45 ± 1.56 | 10.43 ± 1.56 | 79.05 ± 0.29 | 75.09 ± 0.34 | 86.78 ± 1.90 | 93.30 ± 1.05 |
| KFAC-LLLA | 81.56 ± 0.07 | 2.20 ± 0.31 | 2.30 ± 0.32 | 52.57 ± 0.27 | 8.97 ± 0.21 | 8.99 ± 0.21 | 81.04 ± 0.35 | 77.36 ± 0.34 | 80.32 ± 4.41 | 89.05 ± 2.30 |
| Mixup | 82.60 ± 0.37 | 1.77 ± 0.49 | 1.98 ± 0.43 | 56.99 ± 0.54 | 10.32 ± 0.64 | 10.45 ± 1.57 | 78.37 ± 1.20 | 75.95 ± 0.56 | 78.68 ± 4.29 | 88.27 ± 1.89 |
| Mix-MaxEnt | 83.23 ± 0.22 | **1.67 ± 0.59** | **1.76 ± 0.62** | **59.39 ± 0.72** | **7.93 ± 0.84** | **7.93 ± 0.84** | 81.04 ± 0.48 | 77.28 ± 0.35 | **89.32 ± 1.61** | **94.45 ± 0.90** |

Table 1: WideResNet28-10 trained on C100. See Appendix C for the cross-validated hyperparameters.

| | Clean Data | | | Domain-Shift | | | Out-of-Distribution | | | |
|---|---|---|---|---|---|---|---|---|---|---|
| | CIFAR10 (Test) | | | CIFAR10-C | | | CIFAR100 | | SVHN | |
| Methods | Accuracy (↑) | ECE (↓) | AdaECE (↓) | Accuracy (↑) | ECE (↓) | AdaECE (↓) | AUROC (↑) | AUPR (↑) | AUROC (↑) | AUPR (↑) |
| DNN | 96.14 ± 0.08 | 1.26 ± 0.05 | 1.34 ± 0.03 | 76.60 ± 0.28 | 12.64 ± 0.77 | 12.62 ± 0.77 | 88.61 ± 0.34 | 88.91 ± 0.21 | 96.00 ± 1.10 | 98.08 ± 0.66 |
| DNN-SN | 96.22 ± 0.11 | 0.71 ± 0.14 | 1.12 ± 0.16 | 76.56 ± 0.24 | 11.13 ± 0.33 | 11.15 ± 0.33 | 88.56 ± 0.36 | 89.01 ± 0.34 | 95.59 ± 0.49 | 97.85 ± 0.22 |
| DNN-SRN | 96.22 ± 0.10 | 1.24 ± 0.08 | 1.36 ± 0.15 | 77.21 ± 0.39 | 11.97 ± 0.40 | 11.96 ± 0.4 | 88.46 ± 0.36 | 88.84 ± 0.37 | 96.12 ± 1.61 | 98.10 ± 0.81 |
| Deep Ensembles | 96.75 ± 0.05 | 0.81 ± 0.16 | 1.04 ± 0.08 | 78.32 ± 0.06 | 10.11 ± 0.17 | 10.31 ± 0.10 | **91.25 ± 0.14** | **91.12 ± 0.15** | **97.53 ± 0.69** | **98.84 ± 0.28** |
| SNGP | 95.98 ± 0.11 | 0.84 ± 0.13 | 0.87 ± 0.16 | 78.37 ± 0.22 | 11.33 ± 0.38 | 11.34 ± 0.39 | 90.61 ± 0.07 | 90.39 ± 0.12 | 95.25 ± 0.55 | 97.98 ± 0.18 |
| DUQ | 94.7 ± 0.02 | 3.4 ± 0.2 | - | 71.6 ± 0.02 | 18.3 ± 1.1 | - | - | 85.4 ± 1.0 | - | 97.3 ± 1.0 |
| KFAC-LLLA | 96.11 ± 0.04 | 1.06 ± 0.08 | 1.12 ± 0.07 | 76.56 ± 0.18 | 11.69 ± 0.76 | 11.67 ± 0.76 | 89.33 ± 0.23 | 88.52 ± 0.20 | 94.17 ± 1.38 | 96.99 ± 0.94 |
| Mixup | 97.01 ± 0.11 | 0.94 ± 0.21 | 1.16 ± 0.13 | 81.68 ± 0.62 | **7.54 ± 0.83** | **7.83 ± 1.2** | 83.17 ± 0.87 | 85.47 ± 0.45 | 87.53 ± 6.07 | 95.08 ± 2.12 |
| Mix-MaxEnt | **97.44 ± 0.06** | **0.63 ± 0.08** | **0.50 ± 0.08** | **83.10 ± 1.48** | 10.13 ± 1.59 | 10.08 ± 1.59 | 89.13 ± 0.18 | 88.12 ± 0.37 | 96.22 ± 0.49 | 98.01 ± 0.41 |

Table 2: WideResNet28-10 trained on C10. See Appendix C for the cross-validated hyperparameters.

- DE: Deep Ensembles (Lakshminarayanan et al., 2017) with 5 members. Note, it is almost 5x slower than all other approaches mentioned above.

**Code base** For fair comparisons, we developed our own code base for all the approaches mentioned above (except SNGP and DUQ) and performed an extensive hyperparameter search to obtain the strongest possible baselines. For SNGP, we used the available code and made sure that we follow exactly the same procedure as mentioned in their original paper. For DUQ, the original paper did not perform large scale experiments similar to ours. Unfortunately, we could not manage to make their code work on C100 as it exhibited unstable behaviour. For this reason, we borrowed numbers for DUQ from the SNGP paper. Please note that the authors of SNGP performed non-trivial modifications to the original DUQ methodology to make it work on C100. Further details provided in Appendix C.

**Optimization and Hyperparameters** We use SGD with Nesterov momentum $0.9$ and a weight decay of $5 \times 10^{-4}$. For WRN, we apply a dropout $p = 0.1$ at train time. We perform extensive cross-validation of all the hyperparameters for all the baselines. Details provided in Appendix C.

**Evaluation Metrics** For calibration, we employ: (1) the widely used Expected Calibration Error (ECE) (Guo et al., 2017), and (2) the recently proposed Adaptive ECE (AdaECE) (Mukhoti et al., 2020). For all the methods, the ECE and AdaECE are computed after performing temperature scaling (Guo et al., 2017) with a cross-validated temperature parameter. Metrics and uncertainty measures used for out-of-distribution detection are discussed in detail in Section 5.1.3.

## 5.1 EXPERIMENTAL RESULTS

### 5.1.1 ACCURACY AND CALIBRATION ON CLEAN DATA

Our first set of experiments evaluates the accuracy of the proposed method on the test sets of C100 and C10. As it can be observed from the 'Clean Data' column of Tables 1 and 2, our method provides a remarkable increase in accuracy compared to recently proposed approaches for improved uncertainty estimation. For example, in the case of C100, it is $4.03\%$ and $1.67\%$ better than SNGP and KFAC, respectively. Most of the time, while improving uncertainty estimates, such approaches degrade the accuracy. However, Mix-MaxEnt not only provides much improved accuracy compared to these approaches, it also shows either comparable or improved accuracy compared to DE as well. Note, DE is computationally much more expensive than other approaches both at training and test time.

In terms of calibration, our method remarkably improves the ECE and AdaECE outperforming all the baselines. Mixup turns out to be an extremely strong baseline in this regard. We provide ECE

and AdaECE without temperature scaling in Appendix D. Due to lack of space, results using the ResNet50 architecture are presented in Appendix D.

### 5.1.2 ACCURACY AND CALIBRATION ON CORRUPTED DATA (DOMAIN-SHIFT)

To evaluate the behaviour of various models under domain-shift, we resort to the widely used CIFAR-100-C and CIFAR-10-C datsets, corrupted versions of the C10 and C100 datasets (Hendrycks & Dietterich, 2019). The dataset is made by applying 15 synthetically generated but realistic corruptions at 5 degrees of intensity on the test sets of C100 and C10, respectively. The desired behaviour would be to preserve the classification accuracy as much as possible as these corruptions do not impact the underlying label, and to have an appropriate reduction in the confidence when the accuracy of the model degrades so that it is not incorrect with very high confidence. We report the expected accuracy, ECE and AdaECE, averaged over all the corruptions and degrees of intensities in the column 'Domain-Shift' of Tables 1 and 2. It is evident that our approach produces a remarkable improvement in the average accuracy compared to all the baselines. For instance, as shown in Table 1 (for C100), our method achieves an accuracy improvement of **6.85%** over DNN, of **3.8%** over DE, and of **2.4%** over Mixup. Similarly, in Table 2 (for C10), our method achieves an improvement of almost **6.5%** over DNN, of **4.78%** over DE, of **1.42%** over Mixup. Thus, setting new state-of-the-art with significant improvements. In terms of calibration, Mixup outperforms all the approaches for C10 experiments. Mix-MaxEnt turns out to be a strong runner-up in this case performing at least as good as the expensive DE and outperforming SNGP and KFAC. However, for C100 experiments, our approach again outperforms all the baselines. For instance, in terms of ECE (refer Table 1), Mix-MaxEnt obtains a **4.5%** improvement over DE, **2.4%** over Mixup, and **2.5%** over SNGP.

### 5.1.3 PERFORMANCE WHEN EXPOSED TO OUT-OF-DISTRIBUTION SAMPLES

Following the standard evaluation methodology (Liu et al., 2020a), we report the performance in terms of Area Under Receiver Operating Characteristic (AUROC) and Area Under Precision-Recall (AUPR) curves for the binary classification problem between in- and out-distribution samples. The uncertainty of the prediction of the model is normally used to obtain these curves. Given an uncertainty measure, it is important for models to be able to provide reliable uncertainty estimates to obtain good AUROC and AUPR. However, there is no consensus in the literature regarding which uncertainty measure to use. Choice of the measure itself can have significant impact on the quality of the uncertainty estimation. In fact, there are various such metrics and they are not yet compared and evaluated properly on large scale experiments. To fill this gap, we performed extensive experiments with the following metrics to understand their behaviour and choose the one that is stable and provides the strongest possible baseline:

- **Entropy**: $H(\mathbf{p}(\mathbf{x})) = -\sum_{i=1}^{K} \mathbf{p}_i \log \mathbf{p}_i$.
- **Dempster-Shafer** (Sensoy et al., 2018): $\text{DS}(\mathbf{x}) = K/_{K + \sum_{i=1}^{K} \exp(\mathbf{s}_i)}$.
- **Energy**: $E(\mathbf{x}) = -\log \sum_{i=1}^{K} \exp(\mathbf{s}_i)$ (ignoring the temperature parameter). This metric was used in (Liu et al., 2020b) for OOD.
- **Maximum Probability Score**: $\text{MPS}(\mathbf{x}) = \max_i \mathbf{p}_i$.
- **Feature Space Density Estimation** (FSDE): Assuming that the features of each class follow a Gaussian distribution, there are several ways one can estimate the *belief* of a test sample belonging to in-distribution data and treat it as a measure of uncertainty. One such approach is to compute the Mahalanobis score $\arg\min_{i \in y} (\phi(\mathbf{x}) - \mu_i)^T \Sigma_i^{-1} (\phi(\mathbf{x}) - \mu_i)$, where $\mu_i$ and $\Sigma_i$ are class-wise mean and the covariance matrices of the *train* data. We define and use other variants as well: a detailed discussion is provided in Appendix E.

**Remarks regarding various metrics:** We would like to highlight a few important observations that we made regarding these metrics. **(1)** DS and $E$ are equivalent as they are both decreasing functions of $\sum_{i=1}^{K} \exp(\mathbf{s}_i)$, and since log does not modify the monotonicity, both will provide the same ordering of a set of samples. Hence, will give the same AUPR and AUROC values. **(2)** We observed DS and $H$ to perform similarly to each other except in a few situations where DS provided improved results for all the approaches. **(3)** MPS, in many situations, was worse. **(4)** We found Gaussian assumption based density estimation to be *highly unstable*. Though it provided extremely competitive results for C10 experiments, sometimes slightly better than the DS based scores, it performed very poorly on C100. We found this score to be highly unstable as it involves large matrix

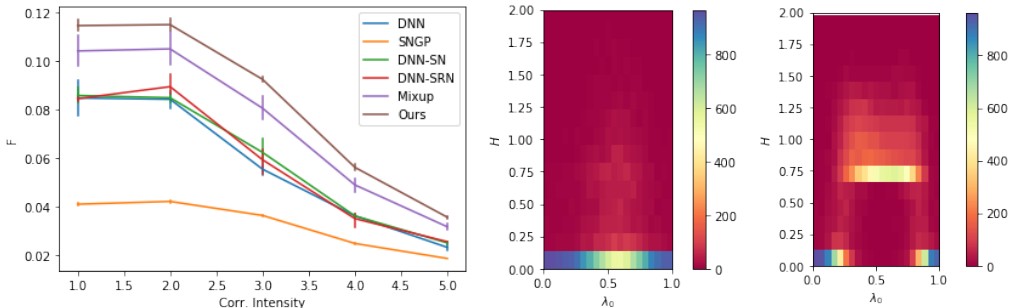

Figure 3: **First figure**: Fisher criterion ($\alpha$) on the embedding space for various degrees of elastic corruption intensity levels for C10 (domain-shift). Higher $\alpha$ indicates more compact and separated clusters in the feature space which is desirable. **Right figures**: heatmaps of the profile of entropies as the interpolation factor between samples of two classes varies; left-most heatmap: DNN WRN on C10, the entropy is mostly zero; right-most heatmap: Mix-MaxEnt WRN on C10, there is a high entropy barrier separating the two classes, while the entropy is low close to the class clusters.

inversions. We applied the well-known tricks such as perturbing the diagonal elements and the low-rank approximation with high variance-ratio, but the results were extremely sensitive to such stabilization and there is no clear way to cross-validate these hyperparameters. We provide analyses based on this metric and its variants in Appendix E. Based on these findings, *we report all the results in the main paper using* $\text{DS}^3$, and report the performance using other scores in Appendix E.

**OOD results:** In the case of SVHN as the OOD dataset (refer Tables 1 and 2 'Out-of-Distribution' column), our method either outperforms all the existing approaches ($2.54\%$ higher AUROC than the runner-up SNGP in C100 experiments) or it turns out to be a runner-up with the gap of $1.31\%$ compared to the top performing and expensive DE. Note, Mix-MaxEnt outperforms all the single model approaches in these experiments as well. In the case of CIFAR as the OOD dataset, DE turns out to be the best performing one. Mix-MaxEnt, along with KFAC-LLLA, is the runner-up in the case of C100 experiments (refer Table 1) while SNGP is the runner-up in C10 experiments (refer Table 2).

### 5.1.4 WHICH METHOD IS THE BEST PERFORMING ONE?

We presented extensive results using a variety of experimental scenarios and metrics. In most of the experiments, Mix-MaxEnt outperformed all the approaches including DE. However, there indeed were a few cases where it was not the best performing one. Following observations further suggest that Mix-MaxEnt is the best performing approach compared to the existing single deterministic alternatives. Out of **20** different evaluations presented in Tables 1 and 2:

- Mix-MaxEnt outperformed all the approaches (including DE) in **11** of them.
- Mix-MaxEnt outperformed all the single model approaches in **14** scenarios.
- No existing single deterministic baseline outperformed all other approaches in more than **2** scenarios.

### 5.2 ANALYSING MIX-MAXENT

### 5.2.1 MIX-MAXENT ENCOURAGES COMPACT AND SEPARATED CLUSTERS

We use the well known Fisher criterion (Bishop, 2006, Chapter 4) to quantify the compactness and separatedness of the feature clusters for various models. Let $\mathcal{C}_k$ denotes the indices of samples for $k$-th class. Then, the overall *within-class* covariance matrix is computed as $\mathbf{S}_W = \sum_{k=1}^{K} \mathbf{S}_k$, where $\mathbf{S}_k = \sum_{n \in \mathcal{C}_k} (\phi(\mathbf{x}_n) - \mu_k)(\phi(\mathbf{x}_n) - \mu_k)^\top$, $\mu_k = \sum_{n \in \mathcal{C}_k} \frac{\phi(\mathbf{x}_n)}{N_k}$, and $\phi(\mathbf{x}_n)$ denote the feature vector. Similarly, the *between-class* covariance matrix can be computed as $\mathbf{S}_B = \sum_{k=1}^{K} N_k (\mu_k - \mu)(\mu_k - \mu)^\top$, where $\mu = \frac{1}{N} \sum_{k=1}^{K} N_k \mu_k$, and $N_k$ is the number of samples in $k$-th class. Then, the Fisher

---

[3] for Mixup we report results using entropy as it provided the best performance

criterion is defined as $\alpha = \texttt{trace}(\mathbf{S}_W^{-1} \mathbf{S}_B)$. Note, $\alpha$ would be high when the within-class covariance is small and between-class covariance is high, thus, a high value of $\alpha$ is desirable. In Figure 3, we compute $\alpha$ over the C10 dataset with varying degrees of domain-shift. As the amount of corruption increases, $\alpha$ gradually decreases for all the models, indicating that the model is not able to differentiate different classes anymore and is projecting them too close to each other. This also explains why the accuracy and the calibration of all the models decreases as the domain-shift increases. However, *Mix-MaxEnt consistently provides the best $\alpha$*. This also explains why Mix-MaxEnt performed so well under domain-shift. We provide further plots and visualizations in Appendix J.

### 5.2.2 MIX-MAXENT CREATES A HIGH ENTROPY BARRIER BETWEEN CLASSES

In Figure 3, we provide heat-maps to visually show the entropy barrier. The heat-map is created as follows. We randomly choose 1000 pairs of samples $\{\mathbf{x}_i, \mathbf{x}_j\}$ such that $y_i \neq y_j$. For each pair, we synthesize 20 samples $\bar{\mathbf{x}}$s using Eq (1) where $\lambda_0$ is equally spaced between 0 to 1. The heat map is then created using all the 20K samples. The intensity of each ($\lambda$, H) bin in the heat-map indicates the number of samples in that bin. It is clearly visible that for standard DNN, although an entropy peak between the two classes exist, the location of the peak is scattered and its intensity is inconsistent. For most of the interpolated samples the entropy is close to zero, thus, making the model overconfident even for samples with very high degree of interpolation. However, Mix-MaxEnt clearly creates a high-entropy barrier that spans most of the space between the two classes. At the same time, the entropy close to the clean samples is low. Therefore, the model is confident for clean samples and it becomes indecisive as the degree of interpolation increases. We show additional such plots and an alternative visualization of this phenomenon in Appendix F.

### 5.2.3 ANALYSING DIFFERENT FACTORS IN THE OBJECTIVE FUNCTION OF MIX-MAXENT

**The importance of mixing samples between classes** We empirically observe that not imposing the constraint $y_i \neq y_j$ does not produce significantly different performance (Table 13 in Appendix G). This is because of the fact that the probability of sampling pairs with the same class is extremely low (0.1 for CIFAR-10, 0.01 for CIFAR-100) compared to sampling pairs from different classes. However, as we enforce the constraint that the pairs must belong to the same class $y_i = y_j$, the performance degrades significantly (refer Table 14). Therefore, mixing samples between classes is crucial.

**The importance of having an interpolation factor close to 0.5** In Table 11, we show that Mix-MaxEnt performance relies on having interpolation factors close to 0.5. Mix-MaxEnt with $\alpha << 1$ (i.e. making $\lambda_0 \approx 0$ or $\approx 1$) degrades the OOD detection performance, the data-shift robustness, and the calibration with respect to the case $\alpha >> 1$ (i.e. making $\lambda_0 \approx 0.5$).

**The importance of maximising the entropy defined over the support of the interpolated points** Recall that Mix-MaxEnt (refer Eq (2)) maximises the entropy on the two interpolated classes. The rationale is that the input image will contain features from the two classes, not from any other classes. Therefore, the classifier should assign a non-zero probability only to these classes and there is no reason for it to assign a non-zero probability to other classes. We empirically check that this is a sensitive choice as maximising the entropy over all the classes significantly degrades the performance (refer Table 12). We also observed that, since $\alpha >> 1$ (and hence $\lambda_0 \approx 0.5$), the entropy regularizer can be replaced by the Mixup loss with no performance degradation.

## 6 CONCLUSION

We proposed Mix-MaxEnt, an extremely simple approach that regularizes a neural network to be uncertain in regions of the data manifold that are unknown during training. We conducted a wide range of experiments to show that Mix-MaxEnt significantly improves the reliability of uncertainty estimates of deep neural networks, while also providing a notable boost in the accuracy. A potential extension of our work regards the possibility of mixing features using more sophisticated methods. An interesting future direction would be to elaborate more sophisticated mixing techniques either based on data statistics (e.g. using PCA directions) or latent space geometry that would allow to map out-of-distribution points closer to the center of the embedding space, and to further improve the clustering for data-shift examples.

**Ethics Statement.** The fact that neural networks can be wrong with high confidence poses a potential threat to the deployment of deep learning systems into the real-world applications such as driverless cars and medical imaging. Therefore, finding fast and effective methods to train models that can provide reliable uncertainty is of utmost importance. Hence, we are not particularly concerned that the research presented in this paper can lead to harmful applications – on the contrary, we believe that it can help deploying safer machine learning applications.

**Reproducibility Statement.** In this paper, we provide an extremely simple approach that can be implemented in merely a few additional lines of code. We have provided all the details needed to reproduce our and baseline results. All the training hyperparameters and details about the dataset (we only used publicly available datasets) are provided in the manuscript. To facilitate further research on this topic, we will release the code to reproduce all the experiments.

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

## A    THE EFFECT OF BOUNDING LOCAL SMOOTHNESS ON MIX-MAXENT

Let $f_\theta = g \circ h(\mathbf{x})$ be the neural network where $h : \mathbf{x} \mapsto \phi(\mathbf{x})$ is the feature extractor and $g(.)$ the last linear layer. It is desirable that the predictive distribution is robust to input perturbations that do not alter the label of the image, so that the network is able to preferentially carry over the true signal in the data. In other words, we would like our model to be bi-Lipschitz so that, given an input perturbation, the change in the output is both upper and lower bounded. Of course, these bounds should be reasonable so that the network neither looses its capacity (is trainable), nor it is too sensitive, which otherwise would cost its generalization ability. It has also been shown that ResNet architectures, because of their skip connections, are by design bi-Lipschitz if all the linear operators are constrained to have a spectral norm of less than one (Behrmann et al., 2019). Recent work (Liu et al., 2020a) built on this finding and showed that applying Spectral Normalization (SN) (Miyato et al., 2018b) to a standard neural network does improve the quality of its uncertainty estimates.

As an alternative, in this work, we enforce bi-Lipschitzness via the application of Stable Rank Normalization (SRN) (Sanyal et al., 2020). Let $\mathbf{W}$ be a linear operator with $\{\sigma_1, \cdots, \sigma_k\}$ as its ordered set of non-zero singular values. Then, SRN effectively provides a new matrix whose singular values are $\{1, \eta\frac{\sigma_2}{\sigma_1} \cdots, \eta\frac{\sigma_k}{\sigma_1}\}$, where $\eta \in (0, 1]$. Note, if $\eta = 1$, SRN boils down to SN. Mathematically speaking, along with controlling the largest singular value of a linear operator, the SRN objective also allows us to minimize its stable rank (a softer version of the rank operator)[4].

We prefer SRN for the following two reasons. (1) It has been shown to control noise sensitivity, and provide better data-dependent empirical Lipschitz constant. (2) It encourages learning more compact embeddings in the feature space (we prove in Proposition A.1), a property that is shown to be important to obtain improved out-domain detection performance (Winkens et al., 2020).

**Proposition A.1.** *Stable Rank Normalization (Sanyal et al., 2020) (SRN) encourages low volume clusters compared to Spectral Normalization (SN).*

*Proof.* Let $\{\mathbf{v}_i \in \mathbb{R}^n\}_{i=1}^n$ be a set of zero-mean vectors with empirical covariance matrix as $\mathbf{S} = \frac{1}{n}\sum_{i=1}^n \mathbf{v}_i\mathbf{v}_i^\top$. Let $\mathbf{W}_0$ be a linear mapping with $\{\sigma_i\}_{i=1}^k$ as its ordered non-zero singular values ($\sigma_1$ being the max). Let the SN and SRN versions of $\mathbf{W}_0$ be $\mathbf{W}_1$ and $\mathbf{W}_2$, respectively. Then, the volume of the covarinace matrix when each vector $\mathbf{v}_i$ is mapped using $\mathbf{W}_0$ can be obained as $\mathtt{Vol}(\mathbf{S}_0) = \det(\frac{1}{n}\sum_{i=1}^n (\mathbf{W}_0\mathbf{v}_i)(\mathbf{W}_0\mathbf{v}_i)^\top) = \det(\mathbf{W}_0)\det(\mathbf{S})\det(\mathbf{W}_0) = \prod_{i=1}^k \sigma_i^2 \det(\mathbf{S})$ (ignoring the multiplicative constant as all the matrices are in the same space). Note, we abuse the notation where the determinant $\det(.)$ of a matrix is the product of non-zero singular values. Similarly, the volume of the projected points when the vectors are mapped via $\mathbf{W}_1$ and $\mathbf{W}_2$ can be obtained as $\mathtt{Vol}(\mathbf{S}_1) = \frac{1}{\sigma_1^{2k}}\prod_{i=1}^k \sigma_i^2 \det(\mathbf{S})$ and $\mathtt{Vol}(\mathbf{S}_s) = \frac{\eta^{2(k-1)}}{\sigma_1^{2k}}\prod_{i=1}^k \sigma_i^2 \det(\mathbf{S})$, respectively. Since $\eta \in (0, 1]$ for SRN, it is trivial to note that $\mathtt{Vol}(\mathbf{S}_2) \leq \mathtt{Vol}(\mathbf{S}_1) \leq \mathtt{Vol}(\mathbf{S}_0)$. Hence, the clusters obtained using SRN will have the lowest volume.                    $\square$

### A.1    CONSIDERING DIFFERENT TECHNIQUES TO BOUND LOCAL SMOOTHNESS

In our experiments, we considered two techniques to bound local smoothness: (1) using Spectral Normalization (SN) (Yoshida & Miyato, 2017), (2) using Stable Rank Normalization (SRN) (Sanyal et al., 2020). We also considered a variant of our method without these two components. We report the performance of these variants in Table 3. It is important to observe that, although in a few cases SN can outperform SRN, in most cases they do not exhibit significantly different behaviour. We also observe that, in the case of ResNet50, bounding local smoothness via SN/SRN does not produce improvements (as already visible in Tables 5 and 6). This probably can be attributed to the high capacity of the network and the (relatively) small quantity of training data in the datasets considered.

### A.2    THE IMPACT OF SRN ON DATA-SHIFT ROBUSTNESS

In this section we report additional plots showing the impact of SRN when applied on top of the maximum entropy regularizer. As it can be seen in Figures 4 and 5, for certain kinds of noises it significantly improves the robustness to data-shift.

---

[4]$\mathtt{srank}(\mathbf{W}) = \frac{\|\mathbf{W}\|_F^2}{\|\mathbf{W}\|_2^2}$, where $\|\mathbf{W}\|_2$ is the spectral norm.

| Methods | Clean | | | Corrupted | | | CIFAR | | SVHN | |
|---|---|---|---|---|---|---|---|---|---|---|
| | Accuracy (↑) | ECE (↓) | AdaECE (↓) | Accuracy (↑) | ECE (↓) | AdaECE (↓) | AUROC (↑) | AUPR (↑) | AUROC (↑) | AUPR (↑) |
| **C10 R50** | | | | | | | | | | |
| Mix-MaxEnt | 96.69 ± 0.17 | 0.65 ± 0.11 | 0.94 ± 0.21 | 81.16 ± 1.48 | 12.61 ± 1.77 | 12.52 ± 1.78 | 87.63 ± 0.67 | 85.85 ± 0.83 | 94.39 ± 0.72 | 96.31 ± 0.49 |
| Mix-MaxEnt-SN | 96.80 ± 0.19 | 0.70 ± 0.11 | 0.74 ± 0.22 | 79.91 ± 1.19 | 13.86 ± 0.85 | 13.78 ± 0.86 | 87.99 ± 0.88 | 86.09 ± 1.21 | 93.32 ± 1.83 | 95.90 ± 1.39 |
| Mix-MaxEnt-SRN | 96.74 ± 0.09 | 0.68 ± 0.06 | 0.79 ± 0.14 | 81.35 ± 1.31 | 12.21 ± 1.84 | 12.12 ± 1.84 | 87.25 ± 0.71 | 86.19 ± 0.69 | 94.18 ± 0.92 | 96.76 ± 0.67 |
| **C100 R50** | | | | | | | | | | |
| Mix-MaxEnt | 81.49 ± 0.31 | 1.57 ± 0.18 | 1.53 ± 0.21 | 57.62 ± 0.30 | 13.42 ± 0.93 | 13.39 ± 0.94 | 79.44 ± 0.33 | 75.80 ± 0.14 | 88.68 ± 0.69 | 93.48 ± 0.34 |
| Mix-MaxEnt-SN | 81.39 ± 0.12 | 1.66 ± 0.21 | 1.68 ± 0.15 | 57.64 ± 0.17 | 12.98 ± 0.28 | 12.95 ± 0.27 | 79.89 ± 0.64 | 76.09 ± 0.29 | 86.38 ± 1.04 | 92.46 ± 0.43 |
| Mix-MaxEnt-SRN | 81.24 ± 0.36 | 2.55 ± 0.40 | 2.33 ± 0.42 | 56.94 ± 1.09 | 14.06 ± 1.81 | 14.03 ± 1.80 | 79.27 ± 0.37 | 75.66 ± 0.25 | 85.31 ± 2.09 | 91.65 ± 1.39 |
| **C10 WRN** | | | | | | | | | | |
| Mix-MaxEnt | 97.44 ± 0.06 | 0.63 ± 0.08 | 0.50 ± 0.08 | 83.10 ± 1.48 | 10.13 ± 1.59 | 10.08 ± 1.59 | 89.13 ± 0.18 | 88.12 ± 0.37 | 96.22 ± 0.49 | 98.01 ± 0.41 |
| Mix-MaxEnt-SN | 97.40 ± 0.07 | 0.53 ± 0.11 | 0.64 ± 0.12 | 83.15 ± 1.33 | 8.46 ± 0.59 | 8.48 ± 0.58 | 89.59 ± 0.73 | 88.56 ± 1.11 | 96.72 ± 0.42 | 98.32 ± 0.29 |
| Mix-MaxEnt-SRN | 97.51 ± 0.06 | 0.53 ± 0.12 | 0.62 ± 0.15 | 84.49 ± 0.53 | 8.00 ± 0.86 | 8.02 ± 0.84 | 89.33 ± 0.60 | 88.40 ± 0.73 | 97.22 ± 0.76 | 98.67 ± 0.38 |
| **C100 WRN** | | | | | | | | | | |
| Mix-MaxEnt | 83.23 ± 0.22 | 1.67 ± 0.59 | 1.76 ± 0.62 | 59.39 ± 0.72 | 7.93 ± 0.84 | 7.93 ± 0.84 | 81.04 ± 0.48 | 77.28 ± 0.35 | 89.32 ± 1.61 | 94.45 ± 0.90 |
| Mix-MaxEnt-SN | 83.87 ± 0.18 | 1.48 ± 0.30 | 1.41 ± 0.32 | 61.06 ± 0.69 | 7.42 ± 0.20 | 7.39 ± 0.21 | 80.83 ± 0.16 | 77.32 ± 0.19 | 91.05 ± 2.77 | 95.59 ± 1.25 |
| Mix-MaxEnt-SRN | 83.96 ± 0.18 | 1.73 ± 0.22 | 1.66 ± 0.30 | 60.70 ± 0.91 | 8.43 ± 1.52 | 8.40 ± 1.52 | 81.74 ± 0.31 | 78.05 ± 0.13 | 90.38 ± 2.40 | 95.15 ± 1.20 |

Table 3: Evaluation of our method considering a version without any regulariser to bound local smoothness (Mix-MaxEnt), using Spectral Normalization (Mix-MaxEnt-SN) and using Stable Rank Normalization (Mix-MaxEnt-SRN). The hyperparameters are set as in Table 4.

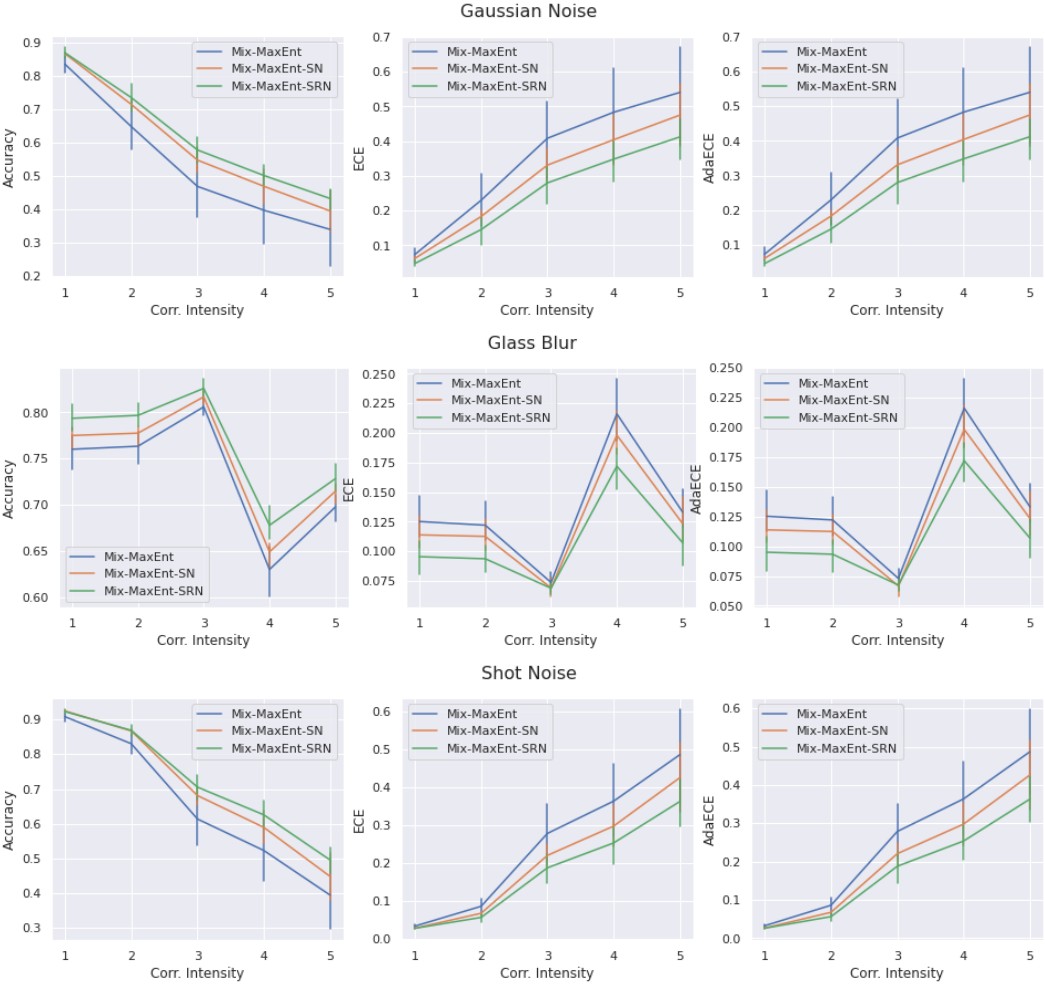

Figure 4: Accuracy, ECE and AdaECE for some corruptions that show the remarkable improvements of using SRN for CIFAR-10-C over not using it, architecture WideResNet28-10.

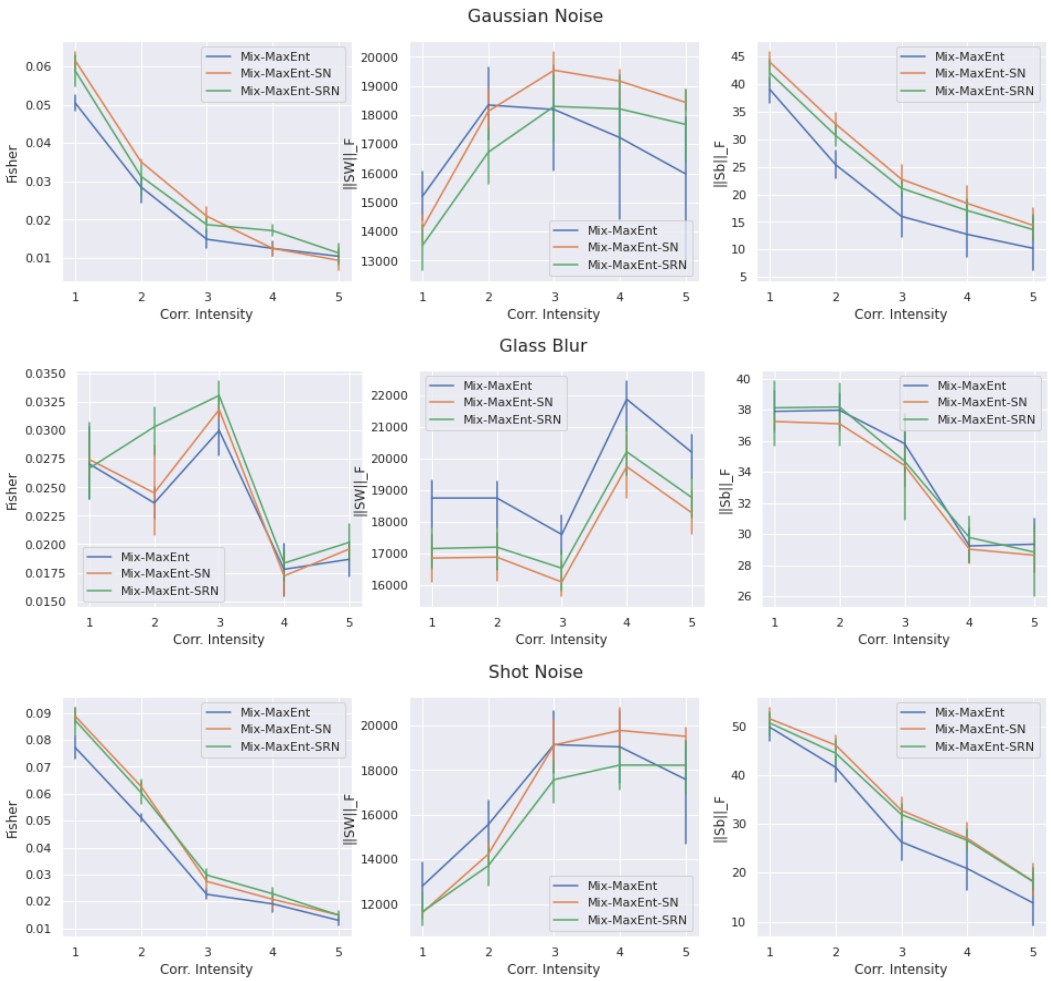

Figure 5: Fisher criterion (and the spectral norm of its factors) for some corruptions that show the remarkable improvements of using SRN for CIFAR-10-C over not using it, architecture WideResNet28-10.

# B    KRONECKER-FACTORED LAST LAYER LAPLACE APPROXIMATION

A structural problem of using MLE logistic regression is that the produced uncertainties depend on the decision boundary. On the other hand, replacing the MLE logistic regression with a Bayesian logistic regression and estimating the predictive posterior employing a Laplace approximation allows to produce better uncertainties (Kristiadi et al., 2020). However, a Bayesian training either requires a modification in the architecture (Liu et al., 2020a) or makes the inference procedure very expensive (Kingma et al., 2015; Gal & Ghahramani, 2016). Since the objective is to utilize the standard MLE training of neural networks, the idea of Kronecker-Factored Last Layer Laplace Approximation (Kristiadi et al., 2020) is making the network Bayesian at test time with almost no additional cost.

Let $\mathbf{w}$ be the parameters of the of the last layer of a neural network, then we seek to obtain the posterior only over $\mathbf{w}$. Let $p(\mathbf{w}|\mathbf{x})$ be the posterior, then the predictive distribution can be written as:

$$p(y = k|\mathbf{x}, \mathcal{D}) = \int \texttt{softmax}(\mathbf{s}_k) p(\mathbf{w}|\mathcal{D}) d\mathbf{w}, \tag{3}$$

where, $\mathbf{s}$ is the logit vector and $\texttt{softmax}(\mathbf{s}_k)$ is the $k$-th index of the $\texttt{softmax}$ output of the network.

The Laplace approximation assumes that the posterior $p(\mathbf{s}|\mathcal{D}) \sim \mathcal{N}(\mathbf{s}|\mu, \Sigma)$, where $\mu$ is a mode of the posterior $p(\mathbf{w}|\mathcal{D})$ (found via standard optimization algorithms for NNs) and $\Sigma$ is the inverse of the Hessian $\mathbf{H}^{-1} = -(\nabla^2 \log p(\mathbf{w}|\mathcal{D})|_\mu)^{-1}$. For the formulations and definitions, including the variants with the terms associated to the bias, we refer to (Kristiadi et al., 2020).

For our experiments, we obtain $\Sigma$ using the Kronecker-factored (KF) approximation (Ritter et al., 2018). Broadly speaking, the KF approximation allows to reduce the computational complexity of computing the Hessian by factorizing the inverse of the Hessian as $\mathbf{H}^{-1} \approx \mathbf{V}^{-1} \otimes \mathbf{U}^{-1}$, then the covariance of the posterior evaluated at a point $\mathbf{x}$ takes following form $\Sigma = (\phi(\mathbf{x})^T \mathbf{V} \phi(\mathbf{x})) \mathbf{U}$. This procedure can be easily implemented using the Backpack library (Dangel et al., 2020) to compute $\mathbf{V}$ and $\mathbf{U}$ by performing a single pass over the training set after the end of the training, as detailed in the Appendix of (Kristiadi et al., 2020) and clearly exemplified in the code-base of (Hobbhahn et al., 2021).

Let $\Sigma_k$ be the covariance matrix of the posterior over the last linear layer parameters for the k-th class obtained using the Laplace approximation around $\mu$, then, given an input $\mathbf{x}$, we obtain $\sigma_k = \phi(\mathbf{x})^\top \Sigma_k \phi(\mathbf{x})$ representing the variance of k-th logit $\mathbf{s}_k$. Once we obtain the covariance matrix, the Monte Carlo approximation of the predictive distribution (equation (3)) is obtained as:

$$\tilde{p} = \frac{1}{m} \sum_{i=1}^{m} \texttt{softmax}(\mathbf{s}(i)), \tag{4}$$

where, $m$ logit vectors $\mathbf{s}(i)$ are sampled from a distribution with mean $\mathbf{s}$ and a covariance matrix (depending on the approximation used). Lu et. al (Lu et al., 2020) showed that similar performance can be achieved via the mean-field approximation which provides an approximate closed form solution of the integration in equation (3) involving the re-scaling of the logits and then taking the softmax of the re-scaled logit. The re-scaling is defined as follows:

$$\tilde{\mathbf{s}}_k = \frac{\mathbf{s}_k}{\sqrt{1 + \lambda \sigma_k^2}} \tag{5}$$

Note, the scaling of the k-th logit depends on its variance (obtained using the Laplace approximation) and a hyperparameter $\lambda$. This approximation is efficient in the sense that it does not require multiple samples as required in the MC approximation (which can become expensive as the number of classes and samples grow). In our experiments, we use the MC approximation, since we could not find an obvious way to fine-tune $\lambda$. Additionally, we observe that the mean-field approximation imposes a trade-off between calibration and OOD detection performance. Increasing $\lambda$, indeed, flattens the softmax distribution and improves OOD detection scores; although, as a consequence, harms calibration by making the network underconfident.

## C   EXPERIMENT DETAILS

**Hyperparameters**   For all our experiments we set the batch size to $128^5$. At training time, we apply standard augmentation (random cropping and horizontal flipping, similarly to (Liu et al., 2020a)). The data is appropriately normalized before being fed to the network both at train and test time.

- for DNN-SN, DNN-SRN and Mix-MaxEnt the set of SN clamping factors we considered in our experiments are $c \in \{0.5, 0.75, 1.0\}$, the target of stable rank $r \in \{0.3, 0.5, 0.7, 0.9\}$ (as $r = 1$ for SRN is the same as applying SN with $c = 1.0$).

- for Mixup, we considered the Beta distribution hyperparameters to be $\alpha \in \{0.1, 0.2, 0.3, 0.4, 0.5\}$, as suggested in the literature (Thulasidasan et al., 2019).

- for Mix-MaxEnt we considered the Beta distribution hyperparameters to be $\alpha \in \{5, 10, 15, 20, 30\}$. In both cases we set $\alpha = \beta$ in $\texttt{Beta}(\alpha, \beta)$.

- for the KFAC-LLLA, we took 1000 samples. Although the number might seem quite high, we could not notice significant improvements using a lower number of samples. We tuned the prior variance $\sigma_0$ needed for the computation of the Laplace approximation minimising the ECE on the validation set. We also tried using the theoretical value $\sigma_0 = 1/\tau$ (Kristiadi et al., 2020), where $\tau$ represents the weight decay, but it produced inferior results with respect to our cross-validation procedure.

- for Deep Ensembles we use 5 members.

- when temperature scaling is applied, the temperature $T$ is tuned on the validation set, minimising the ECE (we considered values ranging from 0.1 to 10, with a step size of 0.01). For Deep Ensembles, we first compute the mean of the logits, then scale it by the temperature parameter before passing it through the softmax.

All the hyperparameters we chose are reported in Table 4. Such hyperparameters have been selected performing cross-validation with stratified-sampling on a 90/10 split of the training set to maximise Accuracy[6]. Indeed, it is important to observe that:

- optimising the hyperparameters based solely on the ECE can prefer models with lower Accuracy but better calibration. However, Accuracy is commonly considered of primary importance, and any method improving calibration should avoid degrading it.

- Optimising considering any of the corrupted experiments and OOD detection metrics would be equivalent to *overfitting the test set* (indeed, both corruptions and OOD datasets are assumed to be unknown at training time by all the considered methodologies, and so they should be during the hyperparameter selection procedure).

For all of our experiments we train the methods initialising the networks with 5 different seeds and report the average and standard deviation *in percentage* for all the metrics. For the hyperparameter search, we trained approximately 350 models. For the final tables (five seeds), we trained approximately 250 models.

**Code**   For the SNGP method we use the official code-base with the suggested hyperparameters and training procedures. The code diverges slightly from the procedure described in their paper, hence the slight differences in the performance. The only modification we performed to the official code-base was to make the inference procedure consistent with the one described in the paper: indeed, in their code they implement a mean-field approximation to estimate the predictive distribution (Lu et al., 2020), while in their paper they use Monte Carlo Integration with a number of samples equal to the number of members in the ensembles they use as a baseline, which provides better calibration. The rationale is that we could not find an obvious way to tune the mean-field approximation hyperparameters to improve at the same time both the calibration and OOD detection performance (indeed, the mean-field approximation imposes a trade-off between calibration and OOD detection performance). Additionally, since the standard KFAC-LLLA uses the same Monte Carlo Integration procedure,

---

[5]For SNGP and DUQ, we use the hyperparameters suggested in their original papers.

[6]Except for the $\sigma_0$ of the KFAC-LLLA, as we could not observe significant differences in Accuracy between hyperparameters optimising the Accuracy and ECE

| Training Set Architecture | Hyp | C10 | | C100 | |
|---|---|---|---|---|---|
| | | WRN | R50 | WRN | R50 |
| DNN | $T$ | 1.3 | 1.5 | 1.3 | 1.4 |
| DNN-SN | $c$ | 0.5 | 0.5 | 0.5 | 0.5 |
| | $T$ | 1.4 | 1.5 | 1.2 | 1.4 |
| DNN-SR | $r$ | 0.3 | 0.3 | 0.3 | 0.3 |
| | $T$ | 1.3 | 1.4 | 1.2 | 1.4 |
| DE | $T$ | 1.3 | 1.4 | 1.1 | 1.2 |
| SNGP | $T$ | 1.4 | - | 1.5 | - |
| Mixup | $\alpha$ | 0.3 | 0.3 | 0.3 | 0.3 |
| | $T$ | 0.7 | 0.8 | 1.09 | 1.2 |
| Mix-MaxEnt | $\alpha$ | 20 | 20 | 10 | 10 |
| | $T$ | 1.1 | 1.3 | 1.2 | 1.2 |
| Mix-MaxEnt-SN | $\alpha$ | 20 | 20 | 20 | 20 |
| | $T$ | 1.23 | 1.23 | 1.09 | 1.19 |
| | $c$ | 0.5 | 0.5 | 0.5 | 0.5 |
| Mix-MaxEnt-SRN | $\alpha$ | 20 | 30 | 20 | 10 |
| | $T$ | 1.2 | 1.3 | 1.09 | 1.09 |
| | $r$ | 0.9 | 0.9 | 0.9 | 0.9 |
| KFAC-LLLA | samples | 1000 | 1000 | 1000 | 1000 |
| | $\sigma_0$ | 1 | 0.6 | 4 | 0.1 |

Table 4: Hyperparameters selected via cross-validation (90/10 split with stratified sampling) for all the methods we trained from scratch. We tuned $T$ and $\sigma_0$ by minimising the ECE. All other hyperparameters have been tuned to maximise the Accuracy.

| Methods | Clean | | | Corrupted | | | CIFAR100 | | SVHN | |
|---|---|---|---|---|---|---|---|---|---|---|
| | Accuracy (↑) | ECE (↓) | AdaECE (↓) | Accuracy (↑) | ECE (↓) | AdaECE (↓) | AUROC (↑) | AUPR (↑) | AUROC (↑) | AUPR (↑) |
| DNN | 95.19 ± 0.23 | 1.38 ± 0.19 | 1.45 ± 0.19 | 75.18 ± 0.69 | 12.31 ± 0.84 | 12.29 ± 0.85 | 88.61 ± 0.66 | 88.05 ± 0.52 | 93.20 ± 1.98 | 96.43 ± 0.95 |
| DNN-SN | 95.20 ± 0.15 | 1.11 ± 0.09 | 1.27 ± 0.10 | 74.88 ± 0.96 | 11.75 ± 0.48 | 11.74 ± 0.49 | 88.19 ± 0.36 | 87.72 ± 0.32 | 93.46 ± 3.41 | 96.56 ± 1.87 |
| DNN-SRN | 95.39 ± 0.08 | 1.23 ± 0.08 | 1.27 ± 0.13 | 75.40 ± 0.67 | 12.22 ± 0.64 | 12.20 ± 0.64 | 88.82 ± 0.40 | 88.15 ± 0.31 | 93.54 ± 2.41 | 96.63 ± 1.27 |
| Deep Ensembles | 96.23 ± 0.05 | 1.27 ± 0.05 | 1.28 ± 0.03 | 77.63 ± 0.36 | 13.12 ± 0.32 | 12.68 ± 0.32 | **91.38 ± 0.21** | **90.75 ± 0.13** | 96.90 ± 0.07 | 98.27 ± 0.09 |
| KFAC-LLLA | 95.21 ± 0.26 | 0.79 ± 0.26 | **0.69 ± 0.24** | 75.18 ± 0.89 | 10.26 ± 0.97 | **10.23 ± 0.97** | 89.54 ± 0.41 | 88.30 ± 0.41 | 93.13 ± 1.01 | 96.25 ± 0.63 |
| Mixup | 96.05 ± 0.15 | **0.59 ± 0.39** | 2.17 ± 0.51 | 78.63 ± 0.72 | **10.17 ± 0.91** | 10.35 ± 0.95 | 84.24 ± 2.95 | 85.35 ± 1.76 | 89.40 ± 4.35 | 95.57 ± 1.41 |
| Mix-MaxEnt | **96.69 ± 0.17** | 0.65 ± 0.11 | 0.94 ± 0.21 | **81.16 ± 1.48** | 12.61 ± 1.77 | 12.52 ± 1.78 | 87.63 ± 0.67 | 85.85 ± 0.83 | **94.39 ± 0.72** | **96.31 ± 0.49** |

Table 5: ResNet50 trained on C10. The cross-validated hyperparameters are provided in Appendix C.

we opted for the latter for a fair comparison. For the KFAC-LLLA we leverage the official repository[7] (Hobbhahn et al., 2021) and the Backpack library (Dangel et al., 2020) for the computation of the Kronecker-Factored Hessian. For the SNGP ResNet50 experiments, we tried running the official implementation. The official implementation for ResNet50 is specifically fine-tuned for ImageNet, and has not been used for experiments on CIFAR. We could not make SNGP converge to SOTA accuracy values both on CIFAR-10 and CIFAR-100. All the other methods were implemented by us in PyTorch and the training, cross-validation and evaluation code will be made publicly available upon acceptance of the paper.

# D  ADDITIONAL RESULTS

## D.1  RESNET50 EXPERIMENTS

In Tables 5 and 6 we report the experimental results for ResNet50.

## D.2  CALIBRATION METRICS WITHOUT TEMPERATURE SCALING

For completeness, we report the calibration metrics over all the methods and considered datasets without temperature scaling (Guo et al., 2017) in Table 7. We can observe that temperature scaling

---

[7]https://github.com/19219181113/LB_for_BNNs

| Methods | Clean | | | Corrupted | | | CIFAR10 | | SVHN | |
|---|---|---|---|---|---|---|---|---|---|---|
| | Accuracy (↑) | ECE (↓) | AdaECE (↓) | Accuracy (↑) | ECE (↓) | AdaECE (↓) | AUPR (↑) | AUROC (↑) | AUPR (↑) | AUROC (↑) |
| DNN | 79.19 ± 0.44 | 3.05 ± 0.29 | 2.94 ± 0.31 | 50.62 ± 0.42 | 19.80 ± 0.31 | 19.76 ± 0.31 | 79.33 ± 0.70 | 75.20 ± 0.61 | 82.45 ± 3.21 | 89.92 ± 1.99 |
| DNN-SN | 79.27 ± 0.25 | 3.15 ± 0.12 | 3.13 ± 0.15 | 50.55 ± 0.39 | 12.19 ± 0.47 | 12.16 ± 0.47 | 79.20 ± 0.17 | 75.22 ± 0.13 | 80.78 ± 1.08 | 88.87 ± 0.84 |
| DNN-SRN | 78.96 ± 0.42 | 2.98 ± 0.24 | 2.95 ± 0.24 | 50.48 ± 0.37 | 12.62 ± 0.58 | 12.59 ± 0.58 | 78.77 ± 0.14 | 74.87 ± 0.15 | 82.39 ± 2.83 | 89.52 ± 1.75 |
| Deep Ensembles | **82.09 ± 0.33** | 3.15 ± 0.10 | 2.98 ± 0.19 | 53.91 ± 0.37 | 12.53 ± 0.31 | 12.36 ± 0.31 | **81.93 ± 0.28** | **77.65 ± 0.34** | 85.08 ± 1.60 | 91.49 ± 0.88 |
| KFAC-LLLA | 79.41 ± 0.44 | **1.30 ± 0.09** | **1.19 ± 0.24** | 50.85 ± 0.49 | **10.59 ± 0.56** | **10.57 ± 0.56** | 79.30 ± 0.41 | 75.27 ± 0.38 | 82.80 ± 3.84 | 90.38 ± 2.17 |
| Mixup | 80.12 ± 0.28 | 7.49 ± 0.32 | 7.47 ± 0.35 | 53.96 ± 0.21 | 13.57 ± 0.38 | 13.52 ± 0.38 | 77.02 ± 0.41 | 74.40 ± 0.43 | 76.86 ± 3.40 | 87.36 ± 1.62 |
| Mix-MaxEnt | 81.49 ± 0.31 | 1.57 ± 0.18 | 1.53 ± 0.21 | **57.62 ± 0.30** | 13.42 ± 0.93 | 13.39 ± 0.94 | 79.44 ± 0.33 | 75.80 ± 0.14 | **88.68 ± 0.69** | **93.48 ± 0.34** |

Table 6: ResNet50 trained on C100. The cross-validated hyperparameters are provided in Appendix C.

| Methods | Clean | | Corrupted | |
|---|---|---|---|---|
| | ECE (↓) | AdaECE (↓) | ECE (↓) | AdaECE (↓) |
| C10 R50 | | | | |
| DNN | 3.06 ± 0.20 | 3.02 ± 0.19 | 17.31 ± 0.73 | 17.30 ± 0.73 |
| DNN-SN | 2.92 ± 0.11 | 2.90 ± 0.11 | 17.41 ± 1.02 | 17.40 ± 1.02 |
| DNN-SRN | 2.85 ± 0.16 | 2.82 ± 0.16 | 17.18 ± 0.62 | 17.17 ± 0.62 |
| Deep Ensembles | 2.12 ± 0.01 | 2.10 ± 0.03 | 14.01 ± 0.32 | 13.99 ± 0.32 |
| KFAC-LLLA | 0.84 ± 0.26 | 0.76 ± 0.24 | 11.59 ± 0.62 | 11.52 ± 0.62 |
| Mixup | 2.84 ± 0.54 | 2.87 ± 0.46 | 11.17 ± 0.91 | 11.35 ± 0.95 |
| Mix-MaxEnt (**Ours**) | 1.64 ± 0.15 | 1.42 ± 0.13 | 12.21 ± 1.84 | 12.12 ± 1.84 |
| C100 R50 | | | | |
| DNN | 9.51 ± 0.58 | 9.47 ± 0.60 | 25.18 ± 1.46 | 25.17 ± 1.47 |
| DNN-SN | 9.47 ± 0.44 | 9.44 ± 0.46 | 25.06 ± 0.73 | 25.04 ± 0.73 |
| DNN-SRN | 9.63 ± 0.23 | 9.59 ± 0.23 | 25.51 ± 0.69 | 25.50 ± 0.69 |
| Deep Ensembles | 6.65 ± 0.12 | 6.50 ± 0.04 | 19.80 ± 0.31 | 19.76 ± 0.31 |
| KFAC-LLLA | 1.56 ± 0.09 | 1.49 ± 0.24 | 12.11 ± 1.12 | 12.18 ± 1.12 |
| Mixup | 7.49 ± 0.32 | 7.47 ± 0.35 | 21.53 ± 0.38 | 21.52 ± 0.38 |
| Mix-MaxEnt (**Ours**) | 4.33 ± 0.43 | 4.17 ± 0.39 | 14.06 ± 1.81 | 14.03 ± 1.80 |
| C10 WRN | | | | |
| DNN | 2.30 ± 0.11 | 2.27 ± 0.11 | 15.94 ± 0.66 | 15.92 ± 0.66 |
| DNN-SN | 2.25 ± 0.12 | 2.21 ± 0.13 | 15.55 ± 0.23 | 15.53 ± 0.23 |
| DNN-SRN | 2.25 ± 0.12 | 2.23 ± 0.13 | 15.13 ± 0.44 | 15.11 ± 0.44 |
| Deep Ensembles | 1.76 ± 0.02 | 1.74 ± 0.03 | 13.54 ± 0.19 | 13.52 ± 0.18 |
| SNGPGood | 1.62 ± 0.09 | 1.51 ± 0.06 | 11.36 ± 0.37 | 11.33 ± 0.36 |
| KFAC-LLLA | 1.06 ± 0.08 | 1.12 ± 0.07 | 11.69 ± 0.76 | 11.67 ± 0.76 |
| Mixup | 2.02 ± 0.72 | 2.23 ± 0.64 | 7.88 ± 0.94 | 7.93 ± 0.97 |
| Mix-MaxEnt (**Ours**) | 0.92 ± 0.06 | 0.71 ± 0.13 | 8.93 ± 0.70 | 8.87 ± 0.69 |
| C100 WRN | | | | |
| DNN | 5.34 ± 0.38 | 5.30 ± 0.42 | 17.43 ± 0.75 | 17.38 ± 0.75 |
| DNN-SN | 5.15 ± 0.25 | 4.97 ± 0.24 | 16.39 ± 0.44 | 16.35 ± 0.45 |
| DNN-SRN | 5.12 ± 0.17 | 5.05 ± 0.25 | 15.75 ± 0.85 | 15.71 ± 0.85 |
| Deep Ensembles | 4.03 ± 0.16 | 3.92 ± 0.16 | 13.51 ± 0.28 | 13.47 ± 0.29 |
| SNGPGood | 5.68 ± 0.26 | 5.65 ± 0.28 | 10.97 ± 1.52 | 10.89 ± 1.52 |
| KFAC-LLLA | 2.20 ± 0.31 | 2.30 ± 0.32 | 8.97 ± 0.21 | 8.99 ± 0.21 |
| Mixup | 3.44 ± 1.08 | 3.60 ± 0.99 | 16.53 ± 1.52 | 16.54 ± 1.49 |
| Mix-MaxEnt (**Ours**) | 2.66 ± 0.07 | 2.47 ± 0.09 | 10.54 ± 1.49 | 10.49 ± 1.49 |

Table 7: Calibration metrics for all networks and all datasets, without temperature scaling

leaves the ranking among methods mostly unchanged. Details about the cross-validation procedure used and the temperature $T$ used are in Section C.

# E  COMPARING DIFFERENT UNCERTAINTY MEASURES

In Section 5.1.3, we reported the OOD detection performance using the DS score, except when it damages the performance of a method (e.g. Mixup) or when it does not yield improvements (e.g. KFAC-LLLA). For completeness reasons, in Table 8 we report the performance with respect to some common uncertainty measures we defined in Section 5.1.3: (1) Maximum Probability Score (MPS), (2) Entropy (H), (3) Energy Score (equivalent to Dempster-Shafer) (DS).

**Feature Space Density Estimation scores**    Similar to (Winkens et al., 2020; Lee et al., 2018), we also use feature space density estimation based score as an estimate of uncertainty. Assuming that a Gaussian Mixture Model would estimate the density of the feature space accurately, the likelihood

| Methods | CIFAR MPS AUROC (↑) | AUPR (↑) | DS AUROC (↑) | AUPR (↑) | H AUROC (↑) | AUPR (↑) | SVHN MPS AUROC (↑) | AUPR (↑) | DS AUROC (↑) | AUPR (↑) | H AUROC (↑) | AUPR (↑)(↑) |
|---|---|---|---|---|---|---|---|---|---|---|---|---|
| **C10 R50** | | | | | | | | | | | | |
| DNN | 88.30 ± 0.43 | 88.30 ± 0.43 | 88.61 ± 0.66 | 88.05 ± 0.52 | 88.52 ± 0.45 | 86.97 ± 0.45 | 91.41 ± 1.75 | 91.41 ± 1.75 | 93.20 ± 1.98 | 96.43 ± 0.95 | 91.76 ± 1.78 | 95.51 ± 0.91 |
| DNN-SN | 87.94 ± 0.25 | 87.94 ± 0.25 | 88.19 ± 0.36 | 87.72 ± 0.32 | 88.18 ± 0.24 | 86.70 ± 0.34 | 92.33 ± 1.87 | 92.33 ± 1.87 | 93.46 ± 3.41 | 96.56 ± 1.87 | 92.72 ± 2.01 | 96.03 ± 1.18 |
| DNN-SRN | 88.34 ± 0.35 | 88.34 ± 0.35 | 88.82 ± 0.40 | 88.15 ± 0.31 | 88.58 ± 0.34 | 86.86 ± 0.38 | 91.55 ± 2.01 | 91.55 ± 2.01 | 93.54 ± 2.41 | 96.63 ± 1.27 | 91.95 ± 2.11 | 95.65 ± 1.09 |
| Deep Ensemble | 90.70 ± 0.17 | 90.70 ± 0.17 | 91.38 ± 0.21 | 90.75 ± 0.13 | 90.97 ± 0.18 | 89.62 ± 0.14 | 94.94 ± 0.06 | 94.94 ± 0.06 | 96.90 ± 0.07 | 98.27 ± 0.09 | 95.41 ± 0.06 | 97.47 ± 0.05 |
| Mixup | 84.31 ± 2.63 | 84.31 ± 2.63 | 80.95 ± 4.05 | 83.68 ± 2.63 | 84.24 ± 2.95 | 85.35 ± 1.76 | 89.59 ± 3.85 | 89.59 ± 3.85 | 84.87 ± 6.98 | 93.99 ± 2.35 | 89.40 ± 4.35 | 95.57 ± 1.41 |
| **(Ours)** | 88.22 ± 0.36 | 88.22 ± 0.36 | 87.63 ± 0.67 | 85.85 ± 0.83 | 88.30 ± 0.40 | 86.62 ± 0.38 | 93.35 ± 0.55 | 93.35 ± 0.55 | 94.39 ± 0.72 | 96.31 ± 0.49 | 93.62 ± 0.59 | 96.52 ± 0.32 |
| **C100 R50** | | | | | | | | | | | | |
| DNN | 78.46 ± 0.51 | 78.46 ± 0.51 | 79.33 ± 0.70 | 75.20 ± 0.61 | 79.18 ± 0.53 | 75.11 ± 0.52 | 79.58 ± 2.52 | 79.58 ± 2.52 | 82.45 ± 3.21 | 89.92 ± 1.99 | 80.90 ± 2.78 | 89.36 ± 1.65 |
| DNN-SN | 78.47 ± 0.05 | 78.47 ± 0.05 | 79.20 ± 0.17 | 75.22 ± 0.13 | 79.13 ± 0.10 | 75.16 ± 0.10 | 78.22 ± 0.50 | 78.22 ± 0.50 | 80.78 ± 1.08 | 88.87 ± 0.84 | 79.32 ± 0.56 | 88.39 ± 0.44 |
| DNN-SR | 78.20 ± 0.13 | 78.20 ± 0.13 | 78.77 ± 0.14 | 74.87 ± 0.15 | 78.90 ± 0.08 | 74.95 ± 0.09 | 80.50 ± 2.06 | 80.50 ± 2.06 | 82.39 ± 2.83 | 89.52 ± 1.75 | 81.58 ± 2.33 | 89.45 ± 1.49 |
| Deep Ensemble | 80.76 ± 0.20 | 80.76 ± 0.20 | 81.93 ± 0.28 | 77.65 ± 0.34 | 81.53 ± 0.21 | 77.38 ± 0.34 | 81.40 ± 1.46 | 81.40 ± 1.46 | 85.08 ± 1.60 | 91.49 ± 0.88 | 82.98 ± 1.61 | 90.62 ± 0.86 |
| Mixup | 77.60 ± 0.25 | 77.60 ± 0.25 | 77.02 ± 0.41 | 74.40 ± 0.43 | 78.25 ± 0.29 | 75.10 ± 0.29 | 78.04 ± 2.60 | 78.04 ± 2.60 | 76.86 ± 3.40 | 87.36 ± 1.62 | 78.88 ± 2.78 | 88.17 ± 1.41 |
| **(Ours)** | 79.52 ± 0.12 | 79.52 ± 0.12 | 79.44 ± 0.33 | 75.80 ± 0.14 | 80.23 ± 0.13 | 76.77 ± 0.11 | 83.75 ± 1.41 | 83.75 ± 1.41 | 88.68 ± 0.69 | 93.48 ± 0.34 | 85.05 ± 1.59 | 91.66 ± 1.01 |
| **C10 WRN** | | | | | | | | | | | | |
| DNN | 88.74 ± 0.24 | 88.74 ± 0.24 | 88.61 ± 0.34 | 88.91 ± 0.21 | 88.90 ± 0.26 | 88.38 ± 0.15 | 95.04 ± 1.00 | 95.04 ± 1.00 | 96.00 ± 1.10 | 98.08 ± 0.66 | 95.43 ± 1.05 | 97.66 ± 0.70 |
| DNN-SN | 88.80 ± 0.28 | 88.80 ± 0.28 | 88.56 ± 0.36 | 89.01 ± 0.34 | 88.95 ± 0.29 | 88.58 ± 0.33 | 95.02 ± 0.41 | 95.02 ± 0.41 | 95.59 ± 0.49 | 97.85 ± 0.22 | 95.34 ± 0.42 | 97.61 ± 0.20 |
| DNN-SRN | 88.76 ± 0.22 | 88.76 ± 0.22 | 88.46 ± 0.36 | 88.84 ± 0.37 | 88.90 ± 0.23 | 88.45 ± 0.26 | 95.47 ± 1.21 | 95.47 ± 1.21 | 96.12 ± 1.61 | 98.10 ± 0.81 | 95.82 ± 1.28 | 97.89 ± 0.67 |
| Deep Ensemble | 90.98 ± 0.12 | 90.98 ± 0.12 | 91.25 ± 0.14 | 91.12 ± 0.15 | 91.19 ± 0.11 | 90.44 ± 0.15 | 96.40 ± 0.61 | 96.40 ± 0.61 | 97.53 ± 0.69 | 98.84 ± 0.28 | 96.87 ± 0.64 | 98.46 ± 0.27 |
| SNGP | 90.62 ± 0.09 | 90.62 ± 0.09 | 90.61 ± 0.07 | 90.39 ± 0.12 | 90.85 ± 0.09 | 89.28 ± 0.22 | 94.17 ± 0.20 | 94.17 ± 0.20 | 95.25 ± 0.55 | 97.98 ± 0.18 | 94.57 ± 0.23 | 97.33 ± 0.13 |
| Mixup | 83.70 ± 0.77 | 83.70 ± 0.77 | 76.73 ± 1.14 | 81.20 ± 0.78 | 83.17 ± 0.87 | 85.47 ± 0.45 | 88.58 ± 5.37 | 88.58 ± 5.37 | 74.56 ± 8.17 | 89.77 ± 3.04 | 87.53 ± 6.07 | 95.08 ± 2.12 |
| **(Ours)** | 89.27 ± 0.48 | 89.27 ± 0.48 | 89.33 ± 0.60 | 88.40 ± 0.73 | 89.43 ± 0.50 | 88.35 ± 0.46 | 95.59 ± 0.92 | 95.59 ± 0.92 | 97.22 ± 0.76 | 98.67 ± 0.38 | 96.07 ± 0.91 | 98.09 ± 0.46 |
| **C100 WRN** | | | | | | | | | | | | |
| DNN | 80.68 ± 0.22 | 80.68 ± 0.22 | 81.06 ± 0.29 | 77.35 ± 0.39 | 81.17 ± 0.26 | 77.48 ± 0.31 | 78.84 ± 4.37 | 78.84 ± 4.37 | 79.69 ± 4.81 | 88.46 ± 2.53 | 79.50 ± 4.57 | 88.53 ± 2.39 |
| DNN-SN | 80.89 ± 0.22 | 80.89 ± 0.22 | 81.10 ± 0.35 | 77.34 ± 0.19 | 81.30 ± 0.27 | 77.56 ± 0.17 | 81.66 ± 2.90 | 81.66 ± 2.90 | 83.43 ± 3.63 | 91.01 ± 2.05 | 82.68 ± 3.19 | 90.62 ± 1.77 |
| DNN-SRN | 80.99 ± 0.15 | 80.99 ± 0.15 | 81.26 ± 0.18 | 77.36 ± 0.30 | 81.42 ± 0.16 | 77.52 ± 0.28 | 83.00 ± 0.87 | 83.00 ± 0.87 | 85.51 ± 1.18 | 91.84 ± 1.12 | 84.29 ± 1.01 | 91.18 ± 1.05 |
| Deep Ensemble | 82.79 ± 0.08 | 82.79 ± 0.08 | 83.26 ± 0.14 | 79.82 ± 0.27 | 83.28 ± 0.09 | 79.89 ± 0.23 | 83.11 ± 1.40 | 83.11 ± 1.40 | 85.07 ± 1.58 | 91.65 ± 0.97 | 84.16 ± 1.49 | 91.25 ± 0.90 |
| SNGP | 78.42 ± 0.30 | 78.42 ± 0.30 | 75.54 ± 0.44 | 72.96 ± 0.32 | 78.99 ± 0.29 | 75.01 ± 0.34 | 77.69 ± 1.76 | 77.69 ± 1.76 | 86.78 ± 1.90 | 93.30 ± 1.05 | 79.91 ± 1.90 | 89.95 ± 1.07 |
| Mixup | 78.42 ± 0.71 | 78.42 ± 0.71 | 77.16 ± 1.58 | 75.03 ± 0.99 | 78.37 ± 1.20 | 75.95 ± 0.56 | 77.34 ± 7.27 | 77.34 ± 7.27 | 74.55 ± 10.13 | 87.30 ± 4.03 | 78.68 ± 4.29 | 88.27 ± 1.89 |
| **(Ours)** | 81.48 ± 0.18 | 81.48 ± 0.18 | 81.74 ± 0.31 | 78.05 ± 0.13 | 82.07 ± 0.23 | 78.61 ± 0.17 | 87.23 ± 2.31 | 87.23 ± 2.31 | 90.38 ± 2.40 | 95.15 ± 1.20 | 88.82 ± 2.47 | 94.29 ± 1.33 |

Table 8: OOD detection performance considering Maximum Probability Score (MPS), Desmpter-Shafer/Energy (DS) and Entropy (H) as uncertainty metrics.

that a given sample $\mathbf{x}$ belongs to class $c$ can be computed as:

$$p(y = c|\mathbf{x}) = \frac{\exp(-(\phi(\mathbf{x}) - \mu_c)^T \Sigma_c^{-1}(\phi(\mathbf{x}) - \mu_c))}{\sqrt{(2\pi)^n \det \Sigma_c}}$$

where, $\mu_c$ and $\Sigma_c$ are the empirical mean and the covariance-matrix of training samples of c-th class. Using $p(y = c|x)$, following scores can be computed to estimate the uncertainty:

- *Maximum Log-likelihood*: $m(\mathbf{x}) = \max_c \log p(y = c|\mathbf{x})$. If the score $m(\mathbf{x})$ is low then the sample more likely belongs to OOD.

- *Belief*: This score indicates how likely is it that the sample $\mathbf{x}$ belongs to IND and is computed as $b(\mathbf{x}) = \sum_c p(y = c|\mathbf{x})$.

We consider two variants of both the scores mentioned above: (1) tied-covariance (**LDA assumption**) when all the covariance matrices are tied and there is one common covariance matrix for the entire training data-set; and (2) per-class covariance matrix (**QDA assumption**) when the covariance matrix is separately computed for each class. Note, in both the assumptions, the mean is always class-dependent.

Note, the scores defined above require inverting matrices, computing logarithms and exponentials. Practically, we observe that these computations introduce numerical instability[8] leading to unreliable uncertainty scores (although in a few cases they can achieve better performance). In particular, the main sources of instability are caused by the inversion of the covariance matrices and by the computation of their log-determinants (hence we evaluate the scores also without considering the log-normalization term). We apply two known procedures to make the process numerically stable:

- Diagonal perturbation by addition of $\epsilon \mathbb{I}$ for the computation of the inverse, where $\epsilon$ is a small positive constant.

- Low-Rank Pseudo-Inverse (LRP): preprocess the covariance matrices (see next), compute the precision as the pseudo-inverse of the pre-processed covariance (stabilised further by adding $\epsilon I$).

---

[8]The causes of such instability can depend on the high dimensionality of the latent space ($h = 640$ for WideResNet28-10, $h = 2048$ for ResNet50) and the few datapoints that can be leveraged for the estimation of the covariances.

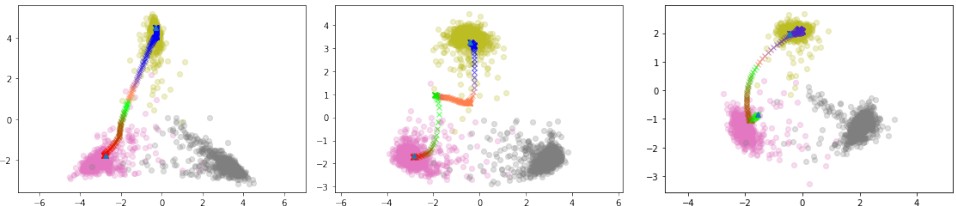

Figure 6: **DNN (Left), Ours (Center), Mixup (Right)**. Interpolation experiment on CIFAR10 to show embeddings of the linear interpolation of two randomly picked input samples from class 1 (purple) and class 2 (yellow). Red and green samples are classified as class 1, and orange and blue samples as class 2. As the color changes from red to green, the predictive entropy increases. Same for the color change from blue to orange. Note, DNN classifies interpolated points with very high confidence (low entropy) even if the samples shift drastically from the data. However, Mix-MaxEnt's entropy increases as we go away from the data. Details of this experiment is provided in Section 5.2. Observe Mixup scale is different from the one of Mix-MaxEnt and DNN.

We preprocess the covariance matrices with the following procedure. Let's assume $A$ is a positive semi-definite matrix, and call $\hat{A}$ an approximation of $A$ that exhibits better numerical stability to inversion and log-determinant operations. We obtain $\hat{A}$ as follows:

- compute the SVD decomposition of the $A$ as $A = U\Sigma V^T$.

- select the top-k SVD components of $A$ that explain the $\gamma\%$ of the variance. Let's call $U_k$, $\Sigma_k$ and $V_k$ the matrices obtained by considering only such components.

- produce an approximation $\hat{A}$ of $A$ as $\hat{A} = U_k \Sigma_k V_k^T$

As it can be seen from Tables 9 and 10, the metric can show outsandingly good performance in detecting OOD data in a few cases. For instance, in Table 9, consider DNN-SRN tested on SVHN: with the LRP stabilisation and $\epsilon = 1e - 08$, $\gamma = 0.95$ the the belief $b_c$ can achieve an AUROC of 97.84% and an AUPR of 99.06%, which are both superior to the AUROC of 96.12% and AUPR of 98.10% reported in Table 8 with the DS metric for the same method. However, as it is clearly visible, the behaviour of these metrics is highly inconsistent across different methods, training and test datasets and stabilisation procedures.

For instance, let's consider some examples from Table 9. It is evident how the $m_t$ is extremely sensitive to the choice of $\epsilon$ for the DNN method: the OOD performance on SVHN drops from 93.99% AUROC and 92.21% AUPR to 73.35% AUROC and 65.90% AUPR when using $\epsilon = 1e - 08$ instead of $\epsilon = 1e - 07$. On the other hand, for the same variation in $\epsilon$, the scores improve for DNN-SN on the same task (increasing the AUROC from 98.34% to 98.40% and the AUPR from 97.05% to 97.12%). Another example of such extreme sensitivity and inconsistency of the metric across various stabilisation techniques and methods can be observed when comparing the $b_c$, $m_c$ and $m_t$ performance using the diagonal perturbation with $\epsilon = 1e - 07$ or the LRP with same $\epsilon$ and $\gamma = 0.99$ for DNN and DNN-SN when tested on SVHN. For DNN, all the scores increase when adding the LRP. For DNN-SN, the $b_c$ increases dramatically (from 76.89% AUROC to 95.35% AUROC, from 88.44% AUPR to 97.89% AUPR) but at the same time a significant drop is observed in $m_c$ and $m_t$ (e.g. $m_t$ drops from 98.34% AUROC to 94.99%, and from 97.05% AUPR to 92.95%).

The fact the considered stabilisation procedures can produce inconsistent improvements and degradations across various methodologies and metrics, and that such variations are extremely sensitive to the choice of the stabilisation hyperparameters makes these metrics unreliable. Indeed, there is no stabilisation procedure with a specific set of hyperparameters that dominates the others in all the cases, and there is no obvious way to blindly (i.e. without resorting to OOD data at training time) select such procedure and hyperparameters.

| Method | Stabilisation | $b_c$ AUROC (↑) | AUPR(↑) | $b_t$ AUROC (↑) | AUPR(↑) | $m_c$ AUROC (↑) | AUPR(↑) | $m_t$ AUROC (↑) | AUPR(↑) |
|---|---|---|---|---|---|---|---|---|---|
| **IND: C10, OOD: SVHN** | | | | | | | | | |
| DNN | $\epsilon$ =1e-07 | 75.78 ± 0.93 | 89.30 ± 1.06 | 40.93 ± 6.32 | 70.61 ± 6.24 | 90.51 ± 15.17 | 84.75 ± 25.13 | 93.99 ± 9.79 | 92.21 ± 11.05 |
| | $\epsilon$ =1e-07 LRP $\gamma$ =0.99 | 92.20 ± 1.89 | 96.57 ± 0.66 | 27.75 ± 4.06 | 60.10 ± 2.38 | 94.13 ± 1.44 | 92.81 ± 1.21 | 95.46 ± 0.61 | 93.64 ± 0.69 |
| | $\epsilon$ =1e-07 LRP $\gamma$ =0.95 | 94.42 ± 1.43 | 97.59 ± 0.58 | 45.57 ± 3.61 | 70.54 ± 2.43 | 93.47 ± 1.23 | 91.82 ± 1.26 | 93.01 ± 0.88 | 91.00 ± 0.93 |
| | $\epsilon$ =1e-07 LRP $\gamma$ =0.85 | 91.21 ± 2.70 | 96.31 ± 1.16 | 71.07 ± 6.71 | 79.87 ± 5.39 | 94.23 ± 1.79 | 92.33 ± 1.96 | 87.53 ± 0.95 | 86.00 ± 0.81 |
| | $\epsilon$ =1e-08 | 68.21 ± 7.28 | 80.20 ± 8.31 | 45.09 ± 5.31 | 68.42 ± 2.99 | 91.13 ± 6.21 | 84.40 ± 11.62 | 73.35 ± 25.14 | 65.90 ± 31.38 |
| | $\epsilon$ =1e-08 LRP $\gamma$ =0.99 | 90.34 ± 3.76 | 95.01 ± 2.90 | 27.90 ± 3.94 | 60.12 ± 2.23 | 86.94 ± 8.97 | 70.10 ± 17.71 | 95.39 ± 0.64 | 93.58 ± 0.69 |
| | $\epsilon$ =1e-08 LRP $\gamma$ =0.95 | 94.68 ± 1.45 | 97.67 ± 0.64 | 45.59 ± 4.26 | 70.52 ± 2.56 | 78.93 ± 27.54 | 76.41 ± 29.15 | 93.09 ± 0.89 | 91.07 ± 0.89 |
| | $\epsilon$ =1e-08 LRP $\gamma$ =0.85 | 90.98 ± 1.88 | 96.20 ± 0.86 | 71.62 ± 6.76 | 80.19 ± 5.40 | 93.38 ± 2.38 | 91.15 ± 2.59 | 87.57 ± 0.79 | 86.03 ± 0.67 |
| DNN-SN | $\epsilon$ =1e-07 | 76.89 ± 12.75 | 88.44 ± 7.08 | 51.82 ± 12.39 | 75.57 ± 6.23 | 96.68 ± 0.58 | 95.45 ± 0.61 | 98.34 ± 0.35 | 97.05 ± 0.49 |
| | $\epsilon$ =1e-07 LRP $\gamma$ =0.99 | 95.35 ± 0.91 | 97.89 ± 0.44 | 36.48 ± 7.91 | 64.47 ± 4.09 | 94.60 ± 0.33 | 93.23 ± 0.43 | 94.99 ± 1.12 | 92.95 ± 1.24 |
| | $\epsilon$ =1e-07 LRP $\gamma$ =0.95 | 96.16 ± 1.50 | 98.31 ± 0.71 | 56.81 ± 7.58 | 76.33 ± 5.00 | 93.64 ± 0.86 | 92.07 ± 0.63 | 93.10 ± 1.11 | 91.14 ± 1.20 |
| | $\epsilon$ =1e-07 LRP $\gamma$ =0.85 | 95.62 ± 2.04 | 98.02 ± 1.00 | 75.70 ± 4.13 | 83.36 ± 4.46 | 93.70 ± 1.22 | 92.22 ± 0.93 | 88.12 ± 1.39 | 86.61 ± 1.50 |
| | $\epsilon$ =1e-08 | 76.66 ± 12.77 | 88.31 ± 7.16 | 49.64 ± 11.90 | 74.38 ± 5.97 | 96.72 ± 0.57 | 95.49 ± 0.61 | 98.40 ± 0.34 | 97.12 ± 0.49 |
| | $\epsilon$ =1e-08 LRP $\gamma$ =0.99 | 94.29 ± 0.17 | 97.37 ± 0.07 | 30.13 ± 6.13 | 61.33 ± 3.46 | 91.74 ± 1.67 | 74.47 ± 5.84 | 95.74 ± 0.98 | 93.74 ± 1.19 |
| | $\epsilon$ =1e-08 LRP $\gamma$ =0.95 | 95.60 ± 1.81 | 98.02 ± 0.84 | 57.19 ± 9.67 | 76.74 ± 6.14 | 70.59 ± 23.63 | 52.27 ± 25.02 | 91.82 ± 2.32 | 88.07 ± 5.19 |
| | $\epsilon$ =1e-08 LRP $\gamma$ =0.85 | 93.50 ± 1.12 | 96.93 ± 0.62 | 77.34 ± 5.18 | 85.19 ± 5.12 | 93.69 ± 0.10 | 87.87 ± 4.19 | 74.43 ± 14.78 | 58.78 ± 28.95 |
| DNN-SRN | $\epsilon$ =1e-07 | 78.16 ± 15.07 | 89.11 ± 8.23 | 57.36 ± 20.24 | 77.96 ± 10.57 | 96.79 ± 0.67 | 95.61 ± 0.59 | 98.24 ± 0.37 | 96.96 ± 0.53 |
| | $\epsilon$ =1e-07 LRP $\gamma$ =0.99 | 94.24 ± 4.77 | 97.17 ± 2.58 | 33.66 ± 16.61 | 62.69 ± 7.51 | 94.76 ± 1.08 | 93.58 ± 0.76 | 95.78 ± 1.97 | 94.10 ± 2.00 |
| | $\epsilon$ =1e-07 LRP $\gamma$ =0.95 | 96.22 ± 2.28 | 98.33 ± 1.13 | 52.62 ± 19.75 | 73.62 ± 10.63 | 93.90 ± 0.79 | 92.69 ± 0.59 | 93.35 ± 2.32 | 91.87 ± 2.02 |
| | $\epsilon$ =1e-07 LRP $\gamma$ =0.85 | 96.07 ± 2.50 | 98.26 ± 1.21 | 71.84 ± 8.46 | 80.62 ± 6.50 | 93.66 ± 0.88 | 92.16 ± 0.91 | 88.44 ± 3.31 | 87.36 ± 2.95 |
| | $\epsilon$ =1e-08 | 77.87 ± 13.54 | 89.37 ± 7.35 | 57.64 ± 18.29 | 78.49 ± 9.72 | 96.64 ± 0.68 | 95.31 ± 0.81 | 98.08 ± 0.50 | 96.74 ± 0.72 |
| | $\epsilon$ =1e-08 LRP $\gamma$ =0.99 | 96.38 ± 0.06 | 98.30 ± 0.12 | 30.70 ± 9.92 | 60.80 ± 3.94 | 93.60 ± 0.24 | 87.80 ± 4.66 | 96.14 ± 0.54 | 94.41 ± 0.24 |
| | $\epsilon$ =1e-08 LRP $\gamma$ =0.95 | 97.83 ± 0.04 | 99.06 ± 0.03 | 50.86 ± 9.50 | 71.66 ± 4.79 | 57.85 ± 35.58 | 55.77 ± 36.66 | 93.02 ± 1.44 | 91.41 ± 1.15 |
| | $\epsilon$ =1e-08 LRP $\gamma$ =0.85 | 96.10 ± 2.38 | 98.23 ± 1.18 | 73.52 ± 6.05 | 81.33 ± 5.65 | 92.93 ± 1.25 | 90.93 ± 1.46 | 85.95 ± 5.92 | 79.31 ± 14.72 |
| **IND: C10, OOD: C100** | | | | | | | | | |
| DNN | $\epsilon$ =1e-07 | 75.98 ± 2.38 | 78.04 ± 1.75 | 66.19 ± 10.03 | 68.82 ± 11.22 | 85.85 ± 11.68 | 84.83 ± 14.96 | 85.97 ± 10.02 | 85.95 ± 9.94 |
| | $\epsilon$ =1e-07 LRP $\gamma$ =0.99 | 83.00 ± 1.55 | 83.79 ± 1.24 | 46.57 ± 1.79 | 50.66 ± 1.53 | 89.57 ± 0.13 | 89.74 ± 0.12 | 86.93 ± 0.40 | 87.32 ± 0.36 |
| | $\epsilon$ =1e-07 LRP $\gamma$ =0.95 | 82.64 ± 1.17 | 83.12 ± 1.12 | 54.27 ± 2.43 | 57.65 ± 1.85 | 89.69 ± 0.12 | 89.97 ± 0.18 | 86.43 ± 0.42 | 87.80 ± 0.41 |
| | $\epsilon$ =1e-07 LRP $\gamma$ =0.85 | 83.42 ± 0.67 | 84.18 ± 0.55 | 63.92 ± 2.76 | 59.78 ± 3.42 | 90.03 ± 0.16 | 90.27 ± 0.18 | 86.08 ± 0.70 | 87.35 ± 0.51 |
| | $\epsilon$ =1e-08 | 71.17 ± 6.84 | 67.49 ± 12.00 | 60.85 ± 11.16 | 60.44 ± 14.17 | 76.14 ± 14.87 | 76.48 ± 15.00 | 74.64 ± 15.73 | 75.60 ± 14.60 |
| | $\epsilon$ =1e-08 LRP $\gamma$ =0.99 | 81.32 ± 3.47 | 80.70 ± 5.54 | 46.52 ± 1.58 | 50.60 ± 1.30 | 82.37 ± 6.42 | 76.12 ± 8.07 | 86.87 ± 0.44 | 87.29 ± 0.40 |
| | $\epsilon$ =1e-08 LRP $\gamma$ =0.95 | 82.54 ± 1.39 | 83.01 ± 1.24 | 54.37 ± 2.35 | 57.71 ± 1.86 | 83.21 ± 12.66 | 81.74 ± 15.98 | 86.37 ± 0.48 | 87.74 ± 0.46 |
| | $\epsilon$ =1e-08 LRP $\gamma$ =0.85 | 83.11 ± 0.83 | 83.86 ± 0.66 | 63.96 ± 2.72 | 59.71 ± 3.52 | 89.41 ± 0.48 | 88.30 ± 1.60 | 86.10 ± 0.70 | 87.38 ± 0.52 |
| DNN-SN | $\epsilon$ =1e-07 | 75.67 ± 1.26 | 77.11 ± 1.32 | 74.67 ± 0.80 | 76.36 ± 0.62 | 91.19 ± 0.17 | 91.53 ± 0.23 | 90.51 ± 0.26 | 90.22 ± 0.34 |
| | $\epsilon$ =1e-07 LRP $\gamma$ =0.99 | 82.86 ± 0.93 | 83.95 ± 0.57 | 50.25 ± 1.22 | 53.14 ± 0.79 | 89.85 ± 0.09 | 90.03 ± 0.13 | 86.77 ± 0.26 | 87.10 ± 0.38 |
| | $\epsilon$ =1e-07 LRP $0\gamma$ =0.95 | 81.96 ± 1.14 | 82.42 ± 1.12 | 57.20 ± 2.77 | 59.69 ± 2.94 | 90.08 ± 0.14 | 90.24 ± 0.19 | 85.81 ± 0.48 | 87.25 ± 0.36 |
| | $\epsilon$ =1e-07 LRP $\gamma$ =0.85 | 83.61 ± 0.96 | 84.33 ± 0.83 | 61.40 ± 1.13 | 58.30 ± 1.50 | 90.38 ± 0.13 | 90.69 ± 0.12 | 86.31 ± 0.85 | 87.54 ± 0.64 |
| | $\epsilon$ =1e-08 | 75.65 ± 1.48 | 77.14 ± 1.58 | 74.35 ± 0.95 | 76.11 ± 0.69 | 91.19 ± 0.16 | 91.54 ± 0.20 | 90.50 ± 0.27 | 90.23 ± 0.34 |
| | $\epsilon$ =1e-08 LRP $\gamma$ =0.99 | 82.83 ± 0.28 | 83.55 ± 0.03 | 50.42 ± 0.37 | 53.94 ± 0.06 | 86.55 ± 0.90 | 79.31 ± 2.19 | 86.81 ± 0.10 | 87.36 ± 0.16 |
| | $\epsilon$ =1e-08 LRP $\gamma$ =0.95 | 81.23 ± 1.39 | 81.30 ± 2.13 | 57.63 ± 3.14 | 59.82 ± 3.31 | 77.65 ± 13.26 | 71.43 ± 14.06 | 84.79 ± 1.50 | 85.23 ± 3.30 |
| | $\epsilon$ =1e-08 LRP $\gamma$ =0.85 | 83.14 ± 2.19 | 83.46 ± 2.35 | 61.56 ± 0.48 | 58.01 ± 0.54 | 88.00 ± 2.43 | 83.78 ± 6.67 | 81.57 ± 4.87 | 77.12 ± 10.71 |
| DNN-SRN | $\epsilon$ =1e-07 | 74.93 ± 1.80 | 76.78 ± 1.79 | 74.06 ± 1.66 | 76.03 ± 1.30 | 91.11 ± 0.16 | 91.40 ± 0.22 | 90.46 ± 0.15 | 90.24 ± 0.31 |
| | $\epsilon$ =1e-07 LRP $\gamma$ =0.99 | 83.49 ± 0.28 | 84.11 ± 0.45 | 48.97 ± 3.59 | 52.81 ± 2.65 | 89.68 ± 0.22 | 89.75 ± 0.30 | 87.08 ± 0.37 | 87.31 ± 0.30 |
| | $\epsilon$ =1e-07 LRP $\gamma$ =0.95 | 82.01 ± 0.82 | 82.25 ± 1.02 | 55.67 ± 3.07 | 59.04 ± 2.67 | 90.17 ± 0.31 | 90.38 ± 0.35 | 86.20 ± 0.52 | 87.59 ± 0.47 |
| | $\epsilon$ =1e-07 LRP $\gamma$ =0.85 | 83.86 ± 0.77 | 84.41 ± 0.88 | 61.29 ± 5.99 | 57.99 ± 5.04 | 90.45 ± 0.13 | 90.75 ± 0.16 | 86.78 ± 0.07 | 87.88 ± 0.31 |
| | $\epsilon$ =1e-08 | 75.88 ± 2.11 | 77.73 ± 2.15 | 74.86 ± 2.70 | 76.78 ± 2.41 | 91.10 ± 0.14 | 91.41 ± 0.20 | 90.52 ± 0.18 | 90.39 ± 0.36 |
| | $\epsilon$ =1e-08 LRP $\gamma$ =0.99 | 83.29 ± 0.23 | 84.08 ± 0.57 | 48.11 ± 1.96 | 52.09 ± 0.94 | 88.70 ± 0.40 | 86.37 ± 2.86 | 87.09 ± 0.12 | 87.47 ± 0.22 |
| | $\epsilon$ =1e-08 LRP $\gamma$ =0.95 | 80.11 ± 2.99 | 80.57 ± 3.23 | 52.96 ± 2.49 | 56.08 ± 1.48 | 76.80 ± 13.19 | 74.44 ± 15.77 | 85.99 ± 0.44 | 87.46 ± 0.51 |
| | $\epsilon$ =1e-08 LRP $\gamma$ =0.85 | 83.97 ± 0.80 | 84.62 ± 0.95 | 61.59 ± 5.35 | 57.84 ± 4.92 | 89.96 ± 0.47 | 89.73 ± 1.29 | 85.03 ± 2.95 | 84.44 ± 6.10 |

Table 9: OOD detection performance when using Feature Density-based estimation methods. The results are reported for WideResNet28-10, trained on CIFAR10. Stabilisation legend: $\epsilon$ alone denotes only the addition of $\epsilon I$, LRP denotes Low-Rank Pseudoinverse, $\gamma$ is the fraction of the explained variance.

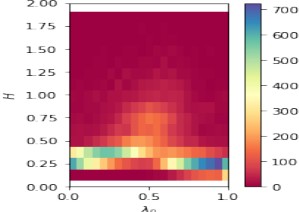

Figure 7: **Left figure**: $(\lambda, H)$-heat map for Mixup ($\alpha = 0.3$) trained using WideResNet28-10 on CIFAR-10.

| Method | Stabilisation | $b_c$ | | $b_t$ | | $m_c$ | | $m_t$ | |
|---|---|---|---|---|---|---|---|---|---|
| | | AUROC (↑) | AUPR(↑) | AUROC (↑) | AUPR(↑) | AUROC (↑) | AUPR(↑) | AUROC (↑) | AUPR(↑) |
| **IND: C100, OOD: SVHN** | | | | | | | | | |
| DNN | $\epsilon$ = 1e-07 | 74.92 ± 4.45 | 84.06 ± 2.83 | 67.19 ± 4.76 | 78.96 ± 2.64 | 89.05 ± 0.97 | 80.60 ± 3.26 | 83.59 ± 3.25 | 73.00 ± 6.28 |
| | $\epsilon$ = 1e-07 LRP $\gamma$ =0.99 | 74.13 ± 4.40 | 84.09 ± 2.38 | 68.71 ± 2.62 | 79.03 ± 1.61 | 79.42 ± 5.43 | 58.66 ± 10.07 | 83.23 ± 3.16 | 73.98 ± 5.20 |
| | $\epsilon$ = 1e-07 LRP $\gamma$ =0.95 | 71.62 ± 4.28 | 82.73 ± 2.46 | 72.52 ± 3.67 | 82.23 ± 2.34 | 64.59 ± 5.39 | 41.18 ± 6.91 | 81.16 ± 4.13 | 71.38 ± 5.92 |
| | $\epsilon$ = 1e-07 LRP $\gamma$ =0.85 | 71.94 ± 5.91 | 83.18 ± 3.35 | 77.36 ± 3.46 | 85.39 ± 2.18 | 63.64 ± 18.10 | 41.83 ± 17.87 | 79.45 ± 3.86 | 69.01 ± 6.76 |
| | $\epsilon$ = 1e-08 | 74.97 ± 4.48 | 84.10 ± 2.86 | 66.78 ± 4.67 | 78.67 ± 2.55 | 89.08 ± 0.94 | 80.64 ± 3.21 | 83.78 ± 3.29 | 73.33 ± 6.25 |
| | $\epsilon$ = 1e-08 LRP 0$\gamma$ = .99 | 76.43 ± 4.31 | 85.50 ± 2.60 | 64.57 ± 2.21 | 77.23 ± 1.84 | 81.17 ± 9.25 | 67.49 ± 15.20 | 63.07 ± 13.63 | 42.73 ± 15.93 |
| | $\epsilon$ = 1e-08 LRP $\gamma$ =0.95 | 73.93 ± 5.65 | 84.22 ± 3.39 | 63.61 ± 12.43 | 76.71 ± 7.55 | 86.16 ± 3.15 | 78.33 ± 4.27 | 63.63 ± 15.63 | 48.36 ± 15.70 |
| | $\epsilon$ = 1e-08 LRP $\gamma$ =0.85 | 72.48 ± 6.25 | 83.49 ± 3.72 | 77.39 ± 2.25 | 85.61 ± 1.32 | 86.06 ± 1.56 | 78.00 ± 2.24 | 77.97 ± 2.59 | 65.06 ± 4.51 |
| DNN-SN | $\epsilon$ = e-07 | 78.39 ± 7.09 | 87.57 ± 4.41 | 70.78 ± 9.13 | 82.65 ± 5.35 | 87.17 ± 3.64 | 80.46 ± 4.07 | 85.47 ± 3.16 | 77.06 ± 5.06 |
| | $\epsilon$ = 1e-07 LRP $\gamma$ =0.99 | 80.04 ± 5.92 | 88.38 ± 3.77 | 72.15 ± 8.68 | 82.09 ± 5.19 | 75.21 ± 8.75 | 60.33 ± 13.59 | 81.08 ± 4.80 | 73.59 ± 5.78 |
| | $\epsilon$ = 1e-07 LRP $\gamma$ =0.95 | 78.26 ± 7.84 | 87.25 ± 5.45 | 78.73 ± 6.55 | 86.95 ± 4.61 | 67.24 ± 9.36 | 43.96 ± 9.70 | 77.90 ± 4.02 | 69.75 ± 5.77 |
| | $\epsilon$ = 1e-07 LRP $\gamma$ =0.85 | 79.08 ± 6.69 | 88.35 ± 4.02 | 81.27 ± 5.18 | 88.68 ± 3.60 | 75.03 ± 9.64 | 56.05 ± 13.94 | 78.68 ± 3.61 | 69.64 ± 5.34 |
| | $\epsilon$ = 1e-08 | 78.43 ± 7.10 | 87.59 ± 4.39 | 70.61 ± 9.09 | 82.51 ± 5.27 | 86.84 ± 3.83 | 80.14 ± 4.26 | 85.67 ± 3.12 | 77.34 ± 5.04 |
| | $\epsilon$ = 1e-08 LRP $\gamma$ =0.99 | 80.94 ± 6.24 | 88.95 ± 4.01 | 70.13 ± 8.56 | 80.54 ± 4.90 | 82.95 ± 5.01 | 75.80 ± 5.56 | 68.23 ± 16.51 | 51.03 ± 22.15 |
| | $\epsilon$ = 1e-08 LRP $\gamma$ =0.95 | 79.52 ± 6.86 | 88.36 ± 4.25 | 77.62 ± 6.66 | 86.46 ± 4.64 | 83.20 ± 4.37 | 75.69 ± 5.06 | 76.18 ± 5.85 | 66.05 ± 8.43 |
| | $\epsilon$ = 1e-08 LRP $\gamma$ =0.85 | 79.35 ± 6.87 | 88.49 ± 4.17 | 81.01 ± 5.52 | 88.68 ± 3.91 | 82.02 ± 5.39 | 72.97 ± 7.56 | 75.98 ± 5.58 | 64.94 ± 7.76 |
| DNN-SRN | $\epsilon$ = 1e-07 | 83.40 ± 3.27 | 90.28 ± 2.31 | 75.35 ± 4.21 | 84.87 ± 2.74 | 85.47 ± 2.90 | 78.86 ± 3.14 | 85.54 ± 2.38 | 77.61 ± 3.32 |
| | $\epsilon$ = 1e-07 LRP $\gamma$ =0.99 | 83.79 ± 2.12 | 90.52 ± 1.72 | 77.51 ± 1.62 | 85.27 ± 1.31 | 74.79 ± 10.68 | 57.64 ± 17.23 | 78.86 ± 1.72 | 71.90 ± 1.78 |
| | $\epsilon$ = 1e-07 LRP $\gamma$ =0.95 | 83.62 ± 2.49 | 90.62 ± 2.01 | 81.47 ± 2.59 | 88.56 ± 1.94 | 65.29 ± 14.76 | 47.76 ± 18.71 | 77.10 ± 1.37 | 70.28 ± 1.69 |
| | $\epsilon$ = 1e-07 LRP $\gamma$ =0.85 | 83.35 ± 2.80 | 90.51 ± 2.22 | 83.46 ± 2.20 | 89.95 ± 1.49 | 64.23 ± 20.69 | 49.43 ± 18.12 | 77.31 ± 1.80 | 69.36 ± 1.78 |
| | $\epsilon$ = 1e-08 | 83.40 ± 3.20 | 90.29 ± 2.26 | 75.06 ± 4.12 | 84.63 ± 2.63 | 84.94 ± 2.98 | 78.26 ± 3.20 | 85.86 ± 2.26 | 78.02 ± 3.17 |
| | $\epsilon$ = 1e-08 LRP $\gamma$ =0.99 | 84.85 ± 1.85 | 91.25 ± 1.48 | 73.59 ± 3.26 | 81.88 ± 3.14 | 82.66 ± 1.99 | 75.12 ± 1.82 | 55.40 ± 15.66 | 44.30 ± 18.25 |
| | $\epsilon$ = 1e-08 LRP $\gamma$ =0.95 | 83.11 ± 0.44 | 89.82 ± 0.33 | 72.14 ± 0.83 | 80.99 ± 0.97 | 84.16 ± 0.13 | 75.27 ± 0.15 | 41.96 ± 3.81 | 27.90 ± 4.82 |
| | $\epsilon$ = 1e-08 LRP $\gamma$ =0.85 | 83.14 ± 0.68 | 90.01 ± 0.45 | 81.96 ± 3.31 | 88.10 ± 2.17 | 84.21 ± 0.62 | 75.88 ± 0.54 | 77.80 ± 5.28 | 67.80 ± 3.03 |
| **IND: C100, OOD: C10** | | | | | | | | | |
| DNN | $\epsilon$ = 1e-07 | 75.87 ± 0.30 | 72.63 ± 0.08 | 76.66 ± 0.43 | 73.63 ± 0.11 | 74.60 ± 0.07 | 78.22 ± 0.13 | 79.68 ± 0.16 | 79.13 ± 0.32 |
| | $\epsilon$ = 1e-07 LRP $\gamma$ =0.99 | 78.02 ± 0.41 | 74.56 ± 0.49 | 73.99 ± 0.75 | 70.55 ± 0.63 | 65.66 ± 5.16 | 62.37 ± 3.87 | 75.28 ± 0.58 | 77.02 ± 0.63 |
| | $\epsilon$ = 1e-07 LRP $\gamma$ =0.95 | 77.87 ± 1.22 | 73.98 ± 1.38 | 73.35 ± 0.78 | 70.68 ± 0.19 | 59.26 ± 4.39 | 57.64 ± 5.76 | 78.76 ± 0.16 | 79.77 ± 0.22 |
| | $\epsilon$ = 1e-07 LRP 0$\gamma$ = .85 | 78.03 ± 0.52 | 74.42 ± 0.58 | 74.47 ± 0.46 | 71.63 ± 0.32 | 68.85 ± 4.10 | 65.08 ± 5.02 | 79.48 ± 0.48 | 80.31 ± 0.46 |
| | $\epsilon$ = 1e-08 | 75.90 ± 0.32 | 72.65 ± 0.08 | 76.70 ± 0.45 | 73.66 ± 0.12 | 74.51 ± 0.10 | 78.14 ± 0.11 | 79.62 ± 0.17 | 79.04 ± 0.34 |
| | $\epsilon$ = e-08 LRP $\gamma$ =0.99 | 77.33 ± 0.61 | 73.80 ± 0.69 | 70.75 ± 1.22 | 66.03 ± 2.57 | 68.85 ± 2.35 | 69.14 ± 4.43 | 57.93 ± 10.05 | 57.55 ± 7.90 |
| | $\epsilon$ = 1e-08 LRP $\gamma$ =0.95 | 77.51 ± 0.66 | 73.87 ± 0.79 | 62.18 ± 10.65 | 58.97 ± 10.98 | 73.25 ± 0.54 | 76.76 ± 0.78 | 65.94 ± 9.03 | 65.91 ± 8.36 |
| | $\epsilon$ = 1e-08 LRP $\gamma$ =0.85 | 77.41 ± 0.75 | 73.72 ± 0.86 | 73.52 ± 1.70 | 70.37 ± 1.21 | 73.90 ± 0.75 | 76.95 ± 1.35 | 77.26 ± 1.70 | 77.73 ± 2.34 |
| DNN-SN | $\epsilon$ = 1e-07 | 75.50 ± 1.04 | 72.32 ± 0.91 | 76.24 ± 0.89 | 73.16 ± 0.84 | 73.02 ± 0.55 | 76.98 ± 0.41 | 79.43 ± 0.27 | 79.38 ± 0.53 |
| | $\epsilon$ = 1e-07 LRP $\gamma$ =0.99 | 78.16 ± 0.68 | 74.97 ± 0.67 | 73.50 ± 1.75 | 70.33 ± 1.17 | 66.45 ± 8.70 | 66.48 ± 9.41 | 74.46 ± 1.04 | 77.09 ± 1.07 |
| | $\epsilon$ = 1e-07 LRP $\gamma$ =0.95 | 78.53 ± 0.44 | 74.12 ± 1.82 | 72.47 ± 1.30 | 70.50 ± 1.41 | 59.81 ± 6.43 | 57.16 ± 5.65 | 78.10 ± 1.13 | 80.18 ± 0.88 |
| | $\epsilon$ = 1e-07 LRP $\gamma$ =0.85 | 78.23 ± 0.61 | 74.86 ± 0.49 | 74.47 ± 1.36 | 71.72 ± 1.57 | 68.25 ± 3.73 | 64.42 ± 6.23 | 79.84 ± 0.39 | 81.20 ± 0.44 |
| | $\epsilon$ = 1e-08 | 75.49 ± 1.07 | 72.27 ± 0.94 | 76.24 ± 0.89 | 73.15 ± 0.86 | 72.26 ± 0.52 | 76.34 ± 0.41 | 79.41 ± 0.27 | 79.34 ± 0.51 |
| | $\epsilon$ = 1e-08 LRP $\gamma$ =0.99 | 77.79 ± 0.49 | 74.61 ± 0.57 | 72.51 ± 1.81 | 68.01 ± 1.96 | 72.17 ± 0.43 | 76.05 ± 0.17 | 59.90 ± 10.30 | 59.94 ± 11.11 |
| | $\epsilon$ = 1e-08 LRP $\gamma$ =0.95 | 78.07 ± 0.35 | 74.81 ± 0.42 | 71.01 ± 1.14 | 68.92 ± 1.48 | 73.01 ± 0.24 | 76.62 ± 0.69 | 76.81 ± 0.67 | 78.65 ± 0.99 |
| | $\epsilon$ = 1e-08 LRP $\gamma$ =0.85 | 77.88 ± 0.30 | 74.59 ± 0.41 | 73.82 ± 1.09 | 70.91 ± 1.14 | 72.62 ± 0.98 | 74.34 ± 2.95 | 78.46 ± 0.81 | 79.88 ± 1.16 |
| DNN-SRN | $\epsilon$ = 1e-07 | 76.60 ± 0.84 | 72.96 ± 0.81 | 77.18 ± 0.58 | 73.78 ± 0.59 | 72.56 ± 0.37 | 76.57 ± 0.29 | 79.41 ± 0.27 | 79.14 ± 0.52 |
| | $\epsilon$ = 1e-07 LRP $\gamma$ =0.99 | 78.68 ± 0.40 | 75.10 ± 0.45 | 74.75 ± 0.95 | 71.45 ± 0.68 | 67.65 ± 3.26 | 63.10 ± 3.73 | 74.82 ± 0.42 | 77.03 ± 0.68 |
| | $\epsilon$ = 1e-07 LRP $\gamma$ =0.95 | 79.14 ± 0.52 | 75.45 ± 0.58 | 73.60 ± 0.87 | 70.95 ± 0.53 | 64.27 ± 3.92 | 60.71 ± 3.34 | 78.54 ± 0.47 | 80.21 ± 0.60 |
| | $\epsilon$ = 1e-07 LRP $\gamma$ =0.85 | 79.24 ± 0.44 | 75.52 ± 0.49 | 75.42 ± 0.56 | 72.23 ± 0.42 | 58.33 ± 10.11 | 56.41 ± 7.85 | 79.78 ± 0.54 | 80.90 ± 0.57 |
| | $\epsilon$ = 1e-08 | 76.60 ± 0.85 | 72.94 ± 0.83 | 77.20 ± 0.58 | 73.81 ± 0.58 | 71.84 ± 0.35 | 75.95 ± 0.29 | 79.36 ± 0.27 | 79.08 ± 0.51 |
| | $\epsilon$ = 1e-08 LRP $\gamma$ =0.99 | 78.58 ± 0.54 | 74.96 ± 0.62 | 69.78 ± 2.67 | 64.10 ± 4.41 | 71.89 ± 0.32 | 75.53 ± 0.45 | 52.72 ± 13.14 | 56.92 ± 12.20 |
| | $\epsilon$ = 1e-08 LRP $\gamma$ =0.95 | 78.72 ± 0.65 | 75.15 ± 0.92 | 69.90 ± 1.69 | 64.89 ± 1.95 | 72.71 ± 0.35 | 76.26 ± 0.36 | 52.74 ± 5.53 | 54.01 ± 1.80 |
| | $\epsilon$ = $e$ 1e-08 LRP $\gamma$ =0.85 | 78.78 ± 0.74 | 75.19 ± 1.00 | 75.21 ± 0.46 | 71.54 ± 0.83 | 72.76 ± 0.42 | 76.19 ± 0.58 | 75.00 ± 3.20 | 73.91 ± 5.78 |

Table 10: OOD detection performance when using Feature Density-based estimation methods. The results are reported for WideResNet28-10, trained on CIFAR100. Stabilisation legend: $\epsilon$ alone denotes only the addition of $\epsilon I$, LRP denotes Low-Rank Pseudoinverse, $\gamma$ is the fraction of the explained variance.

| Methods | Accuracy (↑) | ECE (↓) | AdaECE (↓) | C-Accuracy (↑) | C-ECE (↓) | C-AdaECE (↓) | CIFAR | | SVHN | |
|---|---|---|---|---|---|---|---|---|---|---|
| | | | | | | | AUROC (↑) | AUPR (↑) | AUROC (↑) | AUPR (↑) |
| **C10 WRN** | | | | | | | | | | |
| $\alpha$ =0.1 | 96.36 | 2.00 | 2.02 | 80.00 | 13.01 | 12.99 | 82.70 | 85.32 | 88.57 | 95.66 |
| $\alpha$ =0.2 | 96.54 | 1.01 | 1.48 | 80.41 | 11.58 | 11.66 | 80.39 | 83.25 | 91.42 | 96.83 |
| $\alpha$ =0.3 | 96.92 | 0.70 | 0.94 | 81.91 | 10.09 | 10.11 | 77.67 | 80.37 | 82.81 | 92.95 |
| $\alpha$ =0.4 | 96.77 | 0.75 | 0.79 | 82.54 | 9.82 | 9.86 | 75.66 | 76.64 | 56.72 | 79.17 |
| $\alpha$ =0.5 | 97.04 | 0.86 | 1.14 | 82.40 | 10.77 | 10.80 | 78.52 | 81.58 | 78.57 | 90.80 |
| $\alpha$ =0.6 | 97.28 | 0.87 | 0.91 | 82.57 | 9.60 | 9.64 | 76.29 | 78.95 | 81.40 | 91.91 |
| $\alpha$ =0.7 | 97.10 | 0.99 | 0.91 | 82.69 | 9.14 | 9.15 | 75.00 | 75.81 | 65.30 | 81.58 |
| $\alpha$ =1 | 97.47 | 0.48 | 0.59 | 83.33 | 8.60 | 8.56 | 76.87 | 75.62 | 78.18 | 87.60 |
| $\lambda \in [0.25, 0.75]$ | 97.49 | 0.69 | 0.56 | 85.18 | 7.44 | 7.38 | 87.50 | 86.09 | 94.09 | 96.39 |
| $\lambda \in [0, 0.25] \cup [0.75, 1]$ | 96.61 | 1.08 | 1.49 | 77.62 | 14.14 | 14.24 | 72.88 | 75.30 | 74.55 | 88.41 |
| **C100 WRN** | | | | | | | | | | |
| $\alpha$ =0.1 | 82.78 | 4.75 | 4.68 | 56.00 | 16.14 | 16.11 | 80.00 | 76.49 | 80.04 | 88.36 |
| $\alpha$ =0.2 | 82.82 | 5.17 | 5.08 | 56.55 | 17.85 | 17.78 | 79.30 | 76.02 | 81.52 | 90.01 |
| $\alpha$ =0.3 | 83.28 | 2.77 | 2.75 | 57.31 | 15.44 | 15.40 | 78.34 | 75.95 | 83.72 | 90.87 |
| $\alpha$ =0.4 | 83.38 | 4.00 | 3.77 | 57.87 | 16.51 | 16.46 | 76.47 | 74.37 | 69.74 | 83.85 |
| $\alpha$ =0.5 | 83.44 | 3.09 | 3.10 | 57.85 | 15.72 | 15.68 | 76.79 | 74.17 | 85.14 | 92.03 |
| $\alpha$ =0.6 | 83.31 | 3.86 | 3.64 | 57.92 | 16.54 | 16.49 | 76.81 | 74.12 | 85.05 | 92.00 |
| $\alpha$ =0.7 | 83.52 | 2.63 | 2.36 | 58.14 | 14.38 | 14.34 | 77.78 | 74.73 | 85.72 | 92.77 |
| $\alpha$ =1 | 83.38 | 3.92 | 3.80 | 58.56 | 15.51 | 15.46 | 77.58 | 74.92 | 83.97 | 91.28 |
| $\lambda \in [0.25, 0.75]$ | 83.26 | 4.23 | 4.14 | 61.01 | 12.05 | 12.03 | 80.25 | 76.63 | 91.82 | 95.97 |
| $\lambda \in [0, 0.25] \cup [0.75, 1]$ | 81.46 | 5.44 | 5.31 | 52.84 | 20.88 | 20.84 | 76.31 | 74.80 | 68.34 | 82.50 |

Table 11: WideResNet28-10, Mix-MaxEnt with low $\alpha$ or controlled interpolation factor

# F    ADDITIONAL VISUALIZATION OF THE MAXIMUM ENTROPY EFFECT OF MIX-MAXENT

In Figure 7 we report plots similar to the ones in Section 5.2.2. As already explained, Mixup makes the network less confident both close and far away from the training data, for this reason it results in better calibration (as it alleviates the overconfidence of DNN) but worse OOD detection performance (because both IND and OOD samples have higher entropy, hence are more difficult to distinguish).

An alternative visualization of the phenomenon can be observed with the following method. We take samples from the first three classes of C10, embed them into the feature space, and visualize them using the approach used in (Müller et al., 2019) (presented in Appendix H for completeness). Then, we randomly select two samples $\mathbf{x}_1$ and $\mathbf{x}_2$ from two different classes ('purple' and 'yellow') and create a set of interpolated samples $\{\bar{\mathbf{x}}_i = \lambda\mathbf{x}_1 + (1 - \lambda)\mathbf{x}_2\}$, as $\lambda$ varies from 0 to 1 with an interval of 0.005. Projecting these samples into the embedding space (Figure 6) shows that DNN projects most of the samples to either class 1 or class 2, and very few to the region of high uncertainty. However, Mix-MaxEnt projects most of the interpolated points with $\lambda$ close to 0.5 in high uncertainty regions. Therefore, this plot qualitatively verifies that Mix-MaxEnt creates a high entropy barrier between classes.

# G    RESULTS OF ABLATIONS IN SECTION 5.2.3

In this Section, we report the results that could not be included in Section 5.2.3 for reason of space. In particular, we report Tables 14, 13, 11 and 12.

We also discuss why Mix-MaxEnt interpolates input images and not their features. We did experiment with interpolating IND embeddings from different class clusters and introducing a maximum entropy regularizer on such samples. However, we observed this approach to be ineffective. A reason could be that neural classifiers are notoriously not bijective and interpolations in the embedding space do not necessarily correspond to inputs that even partially resemble data coming from neither $\mathcal{X}_I$ nor $\mathcal{X}_O$ (restricting $\mathcal{X}_O$ to the manifold of natural images). The interpolated images, instead, are bound to contain features from both classes, and hence it is reasonable for the network to be unsure about which of the two classes is represented in them.

| Methods | Accuracy (↑) | ECE (↓) | AdaECE (↓) | C-Accuracy (↑) | C-ECE (↓) | C-AdaECE (↓) | CIFAR | | SVHN | |
|---|---|---|---|---|---|---|---|---|---|---|
| | | | | | | | AUROC (↑) | AUPR (↑) | AUROC (↑) | AUPR (↑) |
| C10 WRN | | | | | | | | | | |
| $\alpha$ =15 | 95.88 | 3.15 | 2.79 | 75.13 | 15.50 | 15.32 | 85.88 | 81.51 | 87.17 | 89.97 |
| $\alpha$ =20 | 95.90 | 3.26 | 2.99 | 73.84 | 15.91 | 15.74 | 85.51 | 81.60 | 91.38 | 91.90 |
| $\alpha$ =30 | 95.99 | 2.79 | 2.05 | 73.59 | 16.20 | 15.9 | 86.18 | 82.96 | 92.55 | 93.47 |
| C100 WRN | | | | | | | | | | |
| $\alpha$ =15 | 80.57 | 7.23 | 7.02 | 50.08 | 18.21 | 18.18 | 76.68 | 69.38 | 72.96 | 80.91 |
| $\alpha$ =20 | 80.24 | 7.71 | 7.41 | 49.99 | 17.00 | 16.93 | 76.69 | 69.71 | 75.60 | 82.09 |
| $\alpha$ =30 | 80.62 | 6.99 | 7.06 | 50.47 | 17.56 | 17.51 | 77.34 | 70.46 | 77.61 | 84.39 |

Table 12: WideResNet28-10, loss: $CE(p_I, y) - \bar{H}(p_O)$ where $\bar{H}(p_O)$ is the entropy over all the labels.

| Methods | Accuracy (↑) | ECE (↓) | AdaECE (↓) | C-Accuracy (↑) | C-ECE (↓) | C-AdaECE (↓) | CIFAR | | SVHN | |
|---|---|---|---|---|---|---|---|---|---|---|
| | | | | | | | AUROC (↑) | AUPR (↑) | AUROC (↑) | AUPR (↑) |
| C10 WRN | | | | | | | | | | |
| $\alpha$ =5 | 97.55 | 0.90 | 0.72 | 83.05 | 9.58 | 9.53 | 87.96 | 86.40 | 94.37 | 96.93 |
| $\alpha$ =10 | 97.19 | 1.16 | 1.07 | 84.29 | 7.35 | 7.28 | 89.90 | 89.28 | 96.58 | 98.38 |
| $\alpha$ =15 | 97.32 | 0.95 | 0.84 | 83.79 | 8.00 | 7.94 | 89.39 | 88.72 | 96.81 | 98.44 |
| $\alpha$ =20 | 97.31 | 0.87 | 0.84 | 83.41 | 9.89 | 9.84 | 88.21 | 87.54 | 96.30 | 98.14 |
| $\alpha$ =30 | 97.46 | 1.12 | 0.85 | 85.65 | 7.90 | 7.85 | 89.27 | 88.98 | 96.89 | 98.55 |
| $\alpha$ =5,$r$ = 0.9 | 97.26 | 0.78 | 0.72 | 84.66 | 8.50 | 8.43 | 87.58 | 85.99 | 94.00 | 96.74 |
| $\alpha$ =10,$r$ = 0.9 | 97.52 | 0.72 | 0.54 | 85.00 | 8.06 | 8.00 | 89.17 | 87.73 | 95.75 | 97.78 |
| $\alpha$ =15,$r$ = 0.9 | 97.56 | 0.78 | 0.59 | 82.24 | 10.66 | 10.61 | 89.31 | 88.54 | 94.30 | 97.41 |
| $\alpha$ =20,$r$ = 0.9 | 97.33 | 1.04 | 0.97 | 83.70 | 8.69 | 8.64 | 89.53 | 89.37 | 97.13 | 98.62 |
| $\alpha$ =30,$r$ = 0.9 | 97.44 | 0.84 | 0.87 | 84.45 | 8.66 | 8.61 | 90.18 | 89.51 | 97.81 | 98.97 |
| C100 WRN | | | | | | | | | | |
| $\alpha$ =5 | 83.15 | 3.76 | 3.69 | 58.83 | 12.98 | 12.96 | 80.68 | 76.86 | 85.68 | 92.63 |
| $\alpha$ =10 | 83.22 | 4.07 | 3.85 | 60.55 | 12.24 | 12.21 | 81.02 | 77.63 | 90.69 | 95.08 |
| $\alpha$ =15 | 82.92 | 4.08 | 4.00 | 59.13 | 11.48 | 11.43 | 81.98 | 78.09 | 83.95 | 90.79 |
| $\alpha$ =20 | 82.92 | 4.28 | 4.13 | 60.30 | 12.80 | 12.77 | 82.36 | 78.47 | 89.48 | 94.25 |
| $\alpha$ =30 | 82.92 | 4.70 | 4.64 | 60.09 | 12.62 | 12.58 | 81.46 | 77.71 | 90.42 | 95.04 |
| $\alpha$ =5,$r$ = 0.9 | 83.81 | 2.85 | 2.51 | 61.46 | 8.61 | 8.57 | 81.07 | 77.39 | 88.15 | 93.17 |
| $\alpha$ =10,$r$ = 0.9 | 83.89 | 3.06 | 2.89 | 60.72 | 10.97 | 10.92 | 82.04 | 78.44 | 93.67 | 96.88 |
| $\alpha$ =15,$r$ = 0.9 | 84.11 | 2.45 | 2.30 | 61.12 | 9.40 | 9.35 | 81.74 | 78.35 | 89.53 | 94.77 |
| $\alpha$ =20,$r$ = 0.9 | 83.81 | 2.55 | 2.23 | 60.77 | 8.95 | 8.88 | 81.55 | 77.81 | 90.97 | 95.42 |
| $\alpha$ =30,$r$ = 0.9 | 83.82 | 2.98 | 2.81 | 61.28 | 9.00 | 8.94 | 80.49 | 77.04 | 88.02 | 93.30 |

Table 13: WideResNet28-10, Mix-MaxEnt (+SRN) mixing samples both within and between classes

| Methods | Accuracy (↑) | ECE (↓) | AdaECE (↓) | C-Accuracy (↑) | C-ECE (↓) | C-AdaECE (↓) | CIFAR | | SVHN | |
|---|---|---|---|---|---|---|---|---|---|---|
| | | | | | | | AUROC (↑) | AUPR (↑) | AUROC (↑) | AUPR (↑) |
| C10 WRN | | | | | | | | | | |
| $\alpha$ =0.1 | 96.49 | 1.88 | 1.89 | 78.45 | 13.57 | 13.55 | 89.39 | 89.64 | 96.87 | 98.43 |
| $\alpha$ =0.2 | 96.21 | 2.08 | 2.05 | 77.63 | 14.38 | 14.36 | 89.76 | 89.72 | 96.51 | 98.11 |
| $\alpha$ =0.3 | 96.69 | 1.67 | 1.63 | 77.91 | 13.41 | 13.39 | 90.47 | 90.38 | 98.26 | 99.19 |
| $\alpha$ =0.4 | 96.59 | 1.74 | 1.74 | 78.23 | 13.14 | 13.12 | 90.07 | 90.03 | 97.47 | 98.84 |
| $\alpha$ =15 | 96.59 | 1.78 | 1.76 | 75.85 | 16.60 | 16.58 | 89.15 | 89.47 | 95.96 | 98.13 |
| $\alpha$ =20 | 96.51 | 1.82 | 1.82 | 77.33 | 14.55 | 14.53 | 89.64 | 89.90 | 94.55 | 97.37 |
| $\alpha$ =30 | 96.67 | 1.78 | 1.74 | 76.53 | 15.79 | 15.78 | 89.48 | 89.80 | 97.84 | 98.86 |
| C100 WRN | | | | | | | | | | |
| $\alpha$ =0.3 | 82.01 | 5.58 | 5.48 | 53.11 | 19.72 | 19.68 | 81.04 | 77.20 | 84.51 | 91.43 |
| $\alpha$ =0.4 | 82.12 | 5.73 | 5.61 | 54.00 | 17.08 | 17.04 | 80.45 | 77.14 | 88.57 | 93.99 |
| $\alpha$ =5 | 82.32 | 4.83 | 4.79 | 52.13 | 18.01 | 17.97 | 80.68 | 76.98 | 87.11 | 92.89 |
| $\alpha$ =15 | 82.06 | 5.02 | 4.89 | 51.97 | 16.96 | 16.91 | 81.85 | 78.40 | 82.17 | 90.41 |
| $\alpha$ =20 | 81.73 | 5.54 | 5.37 | 52.65 | 19.96 | 19.92 | 82.05 | 78.29 | 79.10 | 86.44 |
| $\alpha$ =30 | 82.05 | 5.16 | 5.09 | 52.47 | 18.14 | 18.09 | 80.17 | 76.81 | 82.91 | 89.73 |

Table 14: WideResNet28-10, Mix-MaxEnt with only within-class mixing. The missing $\alpha$ hyperparameters are still training.

# H ALGORITHMS

In this section, we report the algorithm box for a single iteration of the training procedure of Mix-MaxEnt (Algorithm 1) and the visualization technique used to project the embeddings in some of the figures of the paper (Müller et al., 2019) (Algorithm 2)

**Mix-MaxEnt iteration**    Given a batch $B$, a $\lambda_0$ is sampled from $\mathtt{Beta}(\alpha, \alpha)$ and for each sample $(\mathbf{x}_1, y_1)$ our algorithm randomly selects another sample $(\mathbf{x}_2, y_2) \in B$ such that $y_1 \neq y_2$ and adds this sample to a batch $\bar{B}$. Both batches are passed through the network (with a single forward pass, with batch size $2|B|$). The batch $B$ is passed through the usual cross-entropy loss, the batch $\bar{B}$ is passed through our regularizer. The two terms are summed, and one backpropagation pass is performed.

**Visualization Projection Procedure**    Given three classes $1, 2, 3$ and their prototypes $\mathbf{w}_1, \mathbf{w}_2, \mathbf{w}_3$ (e.g. the means of the embeddings of the points of each class), first find an orthonormal basis of the plane passing through the three prototypes by using the standard Gram-Schmidt procedure. Build a projection matrix $P \in \mathbb{R}^{2 \times h}$ by using the vectors of the orthogonal basis (where $h$ is the dimensionality of the embeddings). Use the projection matrix to project any embedding $\phi(\mathbf{x}) \in \mathbb{R}^h$ to $\mathbf{p}_\mathbf{x} \in \mathbb{R}^2$.

---

**Algorithm 1:** Iteration of the Mix-MaxEnt training procedure on a batch $B$ using the network $f$

---

$\bar{B} \leftarrow \emptyset$;
$\lambda_0 \sim \mathtt{Beta}(\alpha, \alpha)$;
**foreach** $(\mathbf{x}_1, y_1) \in B$ **do**
    Randomly select $(\mathbf{x}_2, y_2) \in B | y_1 \neq y_2$;
    $\bar{B} \leftarrow \bar{B} \cup (\lambda_0 \mathbf{x}_1 + (1 - \lambda_0)\mathbf{x}_2, \lambda_0 y_1 + (1 - \lambda_0)y_2)$;
**end**
```
/* Implemented with a single forward pass with double the batch
   size                                                       */
```
$p(y|\mathbf{x} \in B; \theta) \leftarrow f(B)$;
$p(.|\mathbf{x} \in \bar{B}; \theta) \leftarrow f(\bar{B})$;
$\mathcal{L} = -\log p(y|\mathbf{x} \in B; \theta) - \mathcal{H}_{\bar{y}}(p(.|\mathbf{x} \in \bar{B}; \theta))$;
Update $\theta$ backpropagating the gradient of $\mathcal{L}$;

---

---

**Algorithm 2:** Computing the projection of point $\phi(\mathbf{x})$ given three class prototypes $\mathbf{w}_1, \mathbf{w}_2, \mathbf{w}_3$

---

**Result:** $\mathbf{p}_\mathbf{x} \in \mathbb{R}^2$ projection of $\phi(\mathbf{x}) \in \mathbb{R}^h$
```
/* Find an orthonormal basis of the plane passing through the
   three prototypes                                           */
```
$\mathbf{p}_1 \leftarrow (\mathbf{w}_2 - \mathbf{w}_1)/\|\mathbf{w}_2 - \mathbf{w}_1\|$;
$\mathbf{p}_2 \leftarrow \mathbf{w}_3 - \mathbf{w}_2$;
$\mathbf{p}_2 \leftarrow \mathbf{p}_2 - (\mathbf{p}_2^T p_1)\mathbf{p}_1$;
$\mathbf{p}_2 \leftarrow \mathbf{p}_2/\|\mathbf{p}_2\|$;
```
/* Compute the projection matrix                              */
```
$P \leftarrow [\mathbf{p}_1; \mathbf{p}_2]^T$;
```
/* Project the embeddings in 2D                               */
```
$\mathbf{p}_\mathbf{x} = P\phi(\mathbf{x})$;

---

# I EMBEDDING SPACE RELIABILITY PLOTS

A typical technique used to analyse the calibration of a model is to inspect the reliability plot (Guo et al., 2017), i.e. a histogram that shows the relationship between accuracy and confidence. Drawing inspiration from the visualization technique of (Müller et al., 2019) (that we described in Algorithm 2), we design a new visualization technique to analyse the calibration of a neural network. Indeed, one of the main disadvantages of the reliability plots is they do not show which are the regions of the

embedding space in which the model is overconfident/underconfident. The visualization technique we develop, that we call Embedding Space Reliability Plots, leverages Algorithm 2 to compute the 2D projections of the embeddings of a test set. For each test point $\mathbf{x}$, we compute the corresponding accuracy $A(\mathbf{x})$ and confidence $C(\mathbf{x})$, and for each of them we compute the quantity $A(\mathbf{x}) - C(\mathbf{x})$. We associate to each projected point the corresponding difference between the accuracy and confidence, and plot a histogram of this difference. The histogram will reveal in which regions of the latent space covered by the test set the model is overconfident ($A(\mathbf{x}) - C(\mathbf{x}) << 0$) or underconfident ($A(\mathbf{x}) - C(\mathbf{x}) >> 0$) or well calibrated ($A(\mathbf{x}) - C(\mathbf{x}) \approx 0$). Interesting insights can be extracted from these plots.

We report a few visualizations of this kind of plots in Figure 8. We apply it to the test points of CIFAR-10-C for some kinds of corruptions. In the figures, we compare DNN versus our method. The red regions of the space represent areas of overconfidence ($A(\mathbf{x}) - C(\mathbf{x}) < 0$), while green regions of the space represent areas of underconfidence ($A(\mathbf{x}) - C(\mathbf{x}) > 0$). Yellow areas can be either calibrated or contain no data-point. From these plots we can observe that DNN has wider over-confidence regions in the space between the three classes, and shows to be underconfident in some of the regions close to the means of the class clusters. Our method alleviates this issue, making the network overall more calibrated both close and far away from the cluster centers.

We believe this to be a valuable analysis tool that can be further leveraged by other researchers to better understand the calibration behaviour of their models and to design new algorithms.

## J   DETAILED ANALYSIS OF THE CORRUPTION EXPERIMENTS

In this section, we report detailed plots that break down the aggregated metric reported in Table 2 for CIFAR-10-C and WideResNet28-10. Specifically, in Figures 9, 10 and 11 we show how the Accuracy, ECE and AdaECE vary across all the corruption types and intensity values (horizontal axis) over 5 seeds. As it can be seen, in most cases our method achieves better Accuracy than any other method. In terms of calibration, our method is sometimes outperformed by Mixup, but we observe that Mixup exhibits lower Accuracy whenever this happens. As the intensity of the corruption increases, all the metrics deteriorate for all the methods (as expected). However, our method still achieves superior performance in most of the cases also in these circumstances.

To support our claim that better clustering behaviour induces better classification performance, we report the Fisher criterion, $||S_W||_F$ and $||S_B||_F$ (we defined in Section 5.2.1) plots for all corruptions and all intensity levels in Figures 12, 13 and 14.

The observed pattern is similar for all considered architectures and corrupted datasets, hence we report these plots only for WideResNet28-10 on CIFAR-10-C.

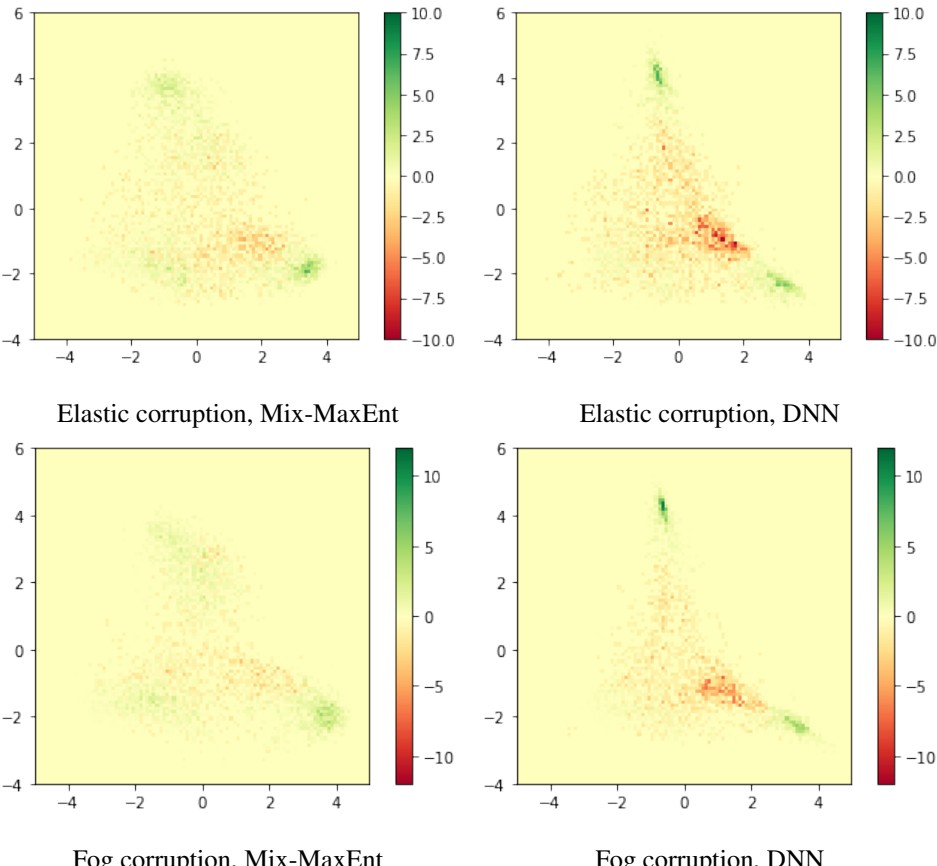

Elastic corruption, Mix-MaxEnt                    Elastic corruption, DNN

Fog corruption, Mix-MaxEnt                        Fog corruption, DNN

Figure 8: Embedding Space Reliability Plots: a simple way to visualize the confidence/overconfidence regions in the latent space for a specific test set. Red points represent histogram bins on which the network is overconfident. Green points represent histogram bins on which the network is underconfident. Yellow histogram bins can either represent well-calibrated histogram bins or areas of the projection space that contain no embedding. It is clearly visible that Mix-MaxEnt produces better-calibrated predictions by reducing the overconfidence far away from the centers of the clusters and by increasing the confidence close to the cluster centers. On the other hand, DNN shows large and sharp overconfidence/underconfidence regions.

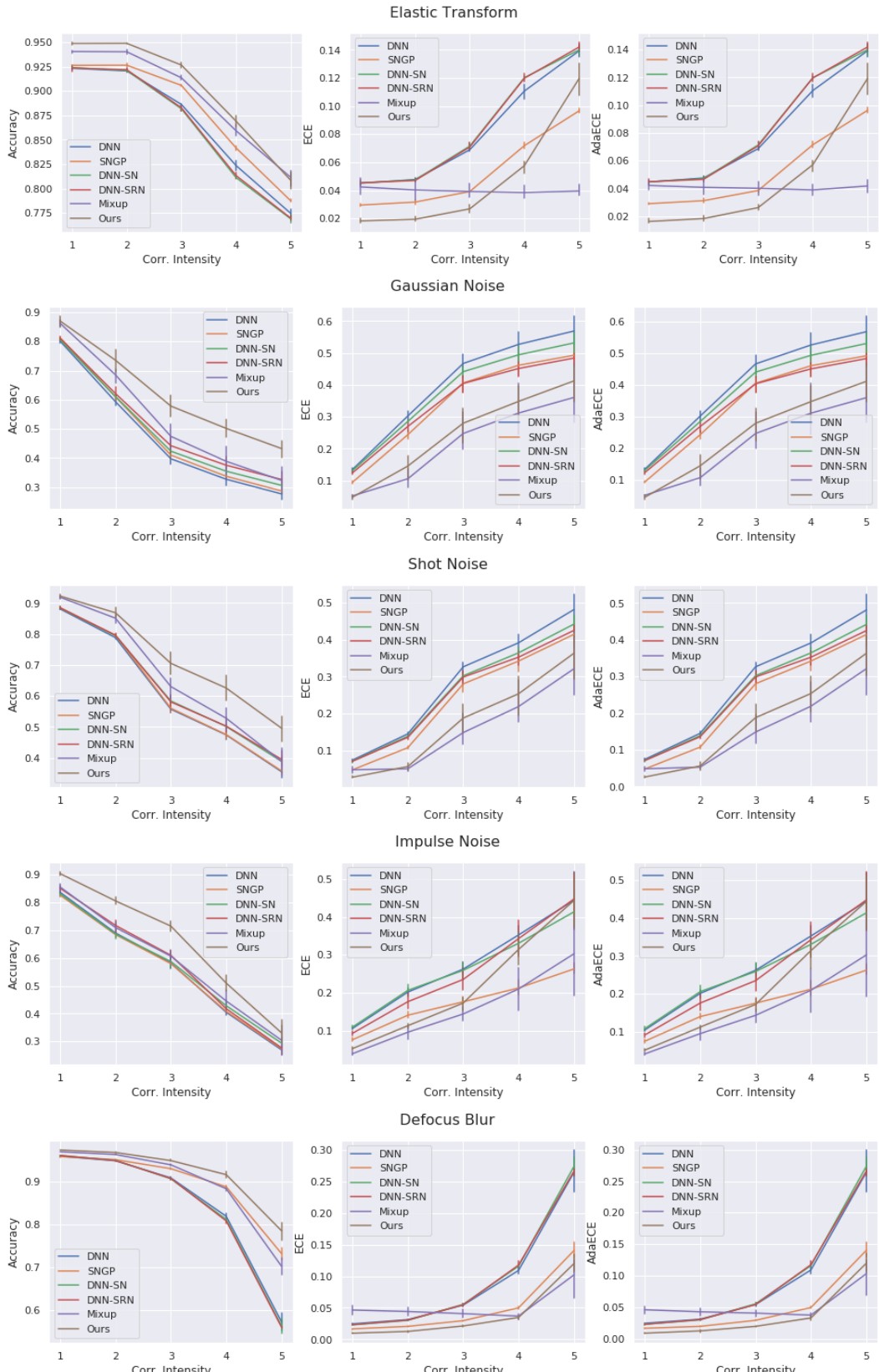

Figure 9: (Part 1 of 3) Accuracy, ECE and AdaECE for all corruptions and intensity values of CIFAR-10-C, architecture WideResNet28-10. A similar pattern can be observed in all other cases.

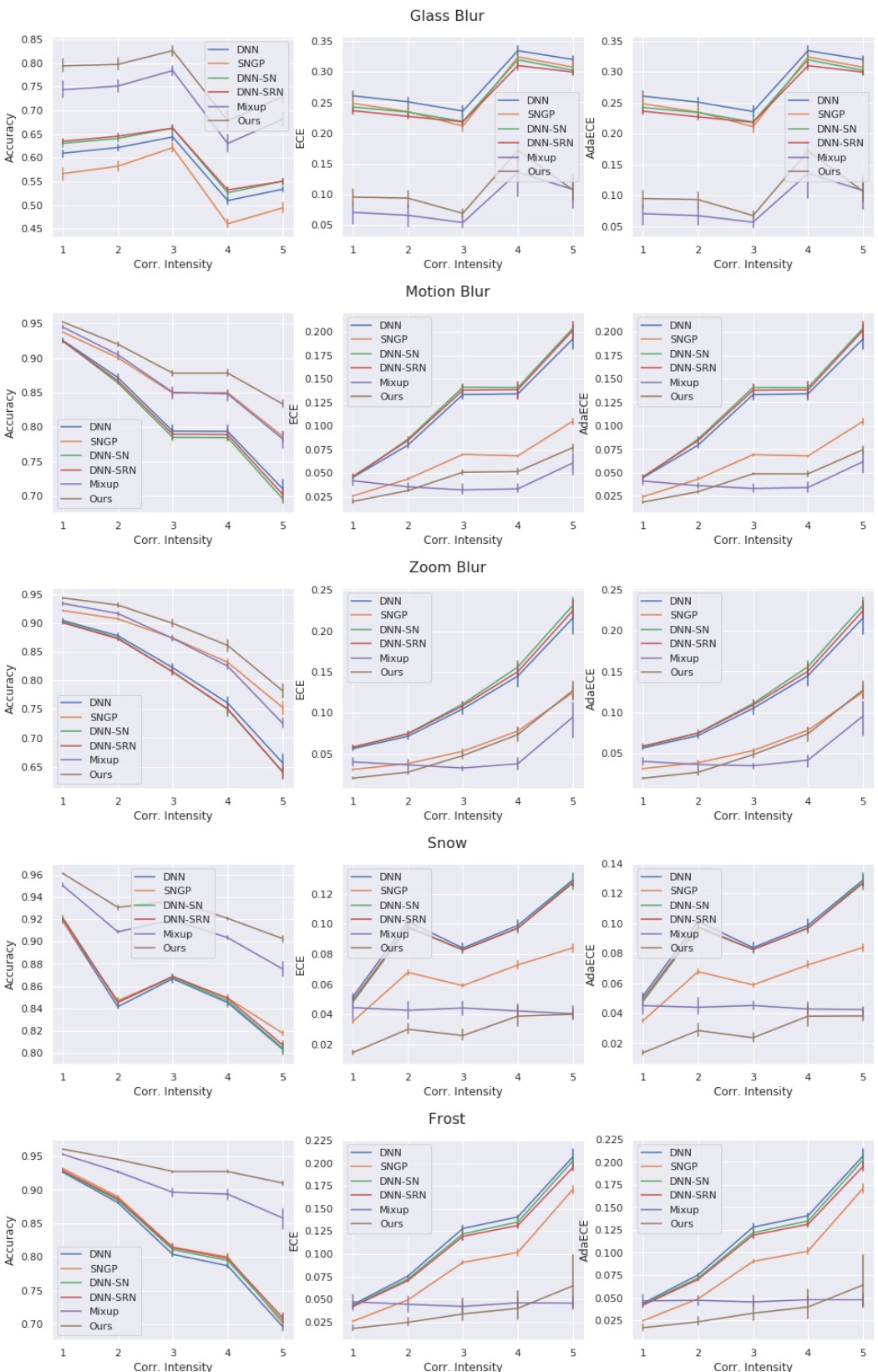

Figure 10: (Part 2 of 3) Accuracy, ECE and AdaECE for all corruptions and intensity values of CIFAR-10-C, architecture WideResNet28-10.

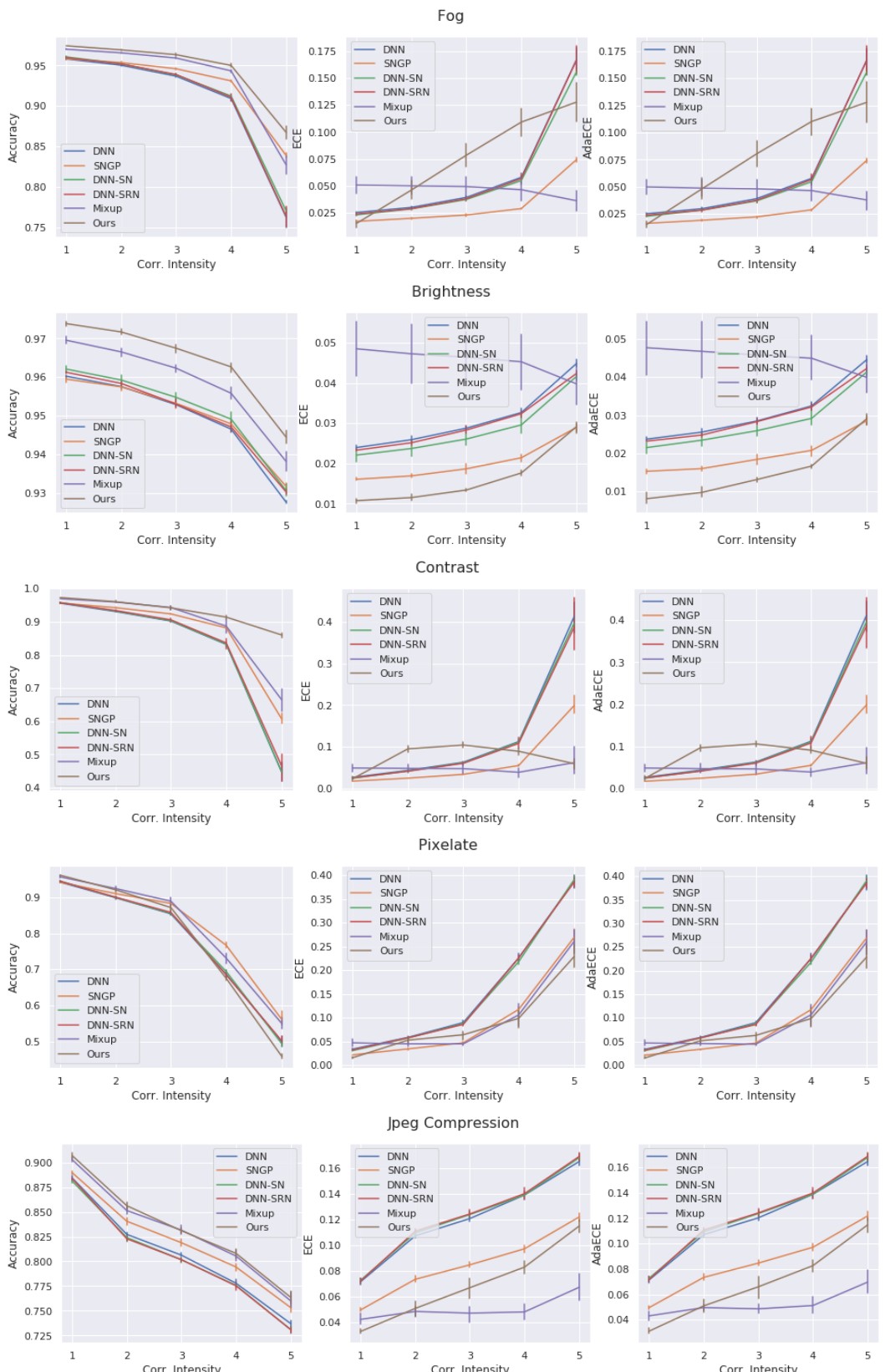

Figure 11: (Part 3 of 3) Accuracy, ECE and AdaECE for all corruptions and intensity values of CIFAR-10-C, architecture WideResNet28-10.

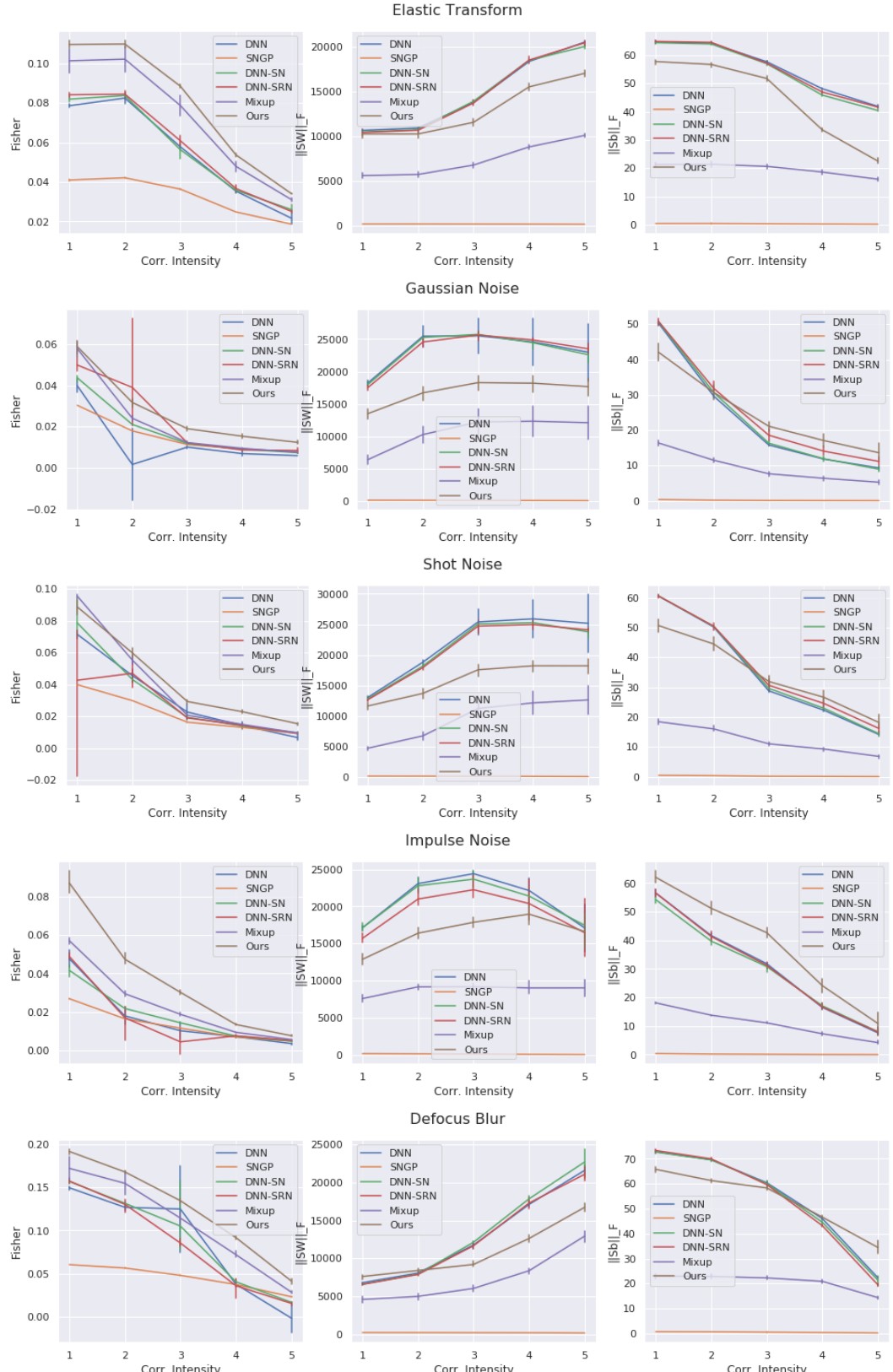

Figure 12: (Part 1 of 3) Fisher criterion, $||S_W||_F$ and $||S_B||_F$ for all corruptions and intensity values of CIFAR-10-C, architecture WideResNet28-10. A similar pattern can be observed in all other cases.

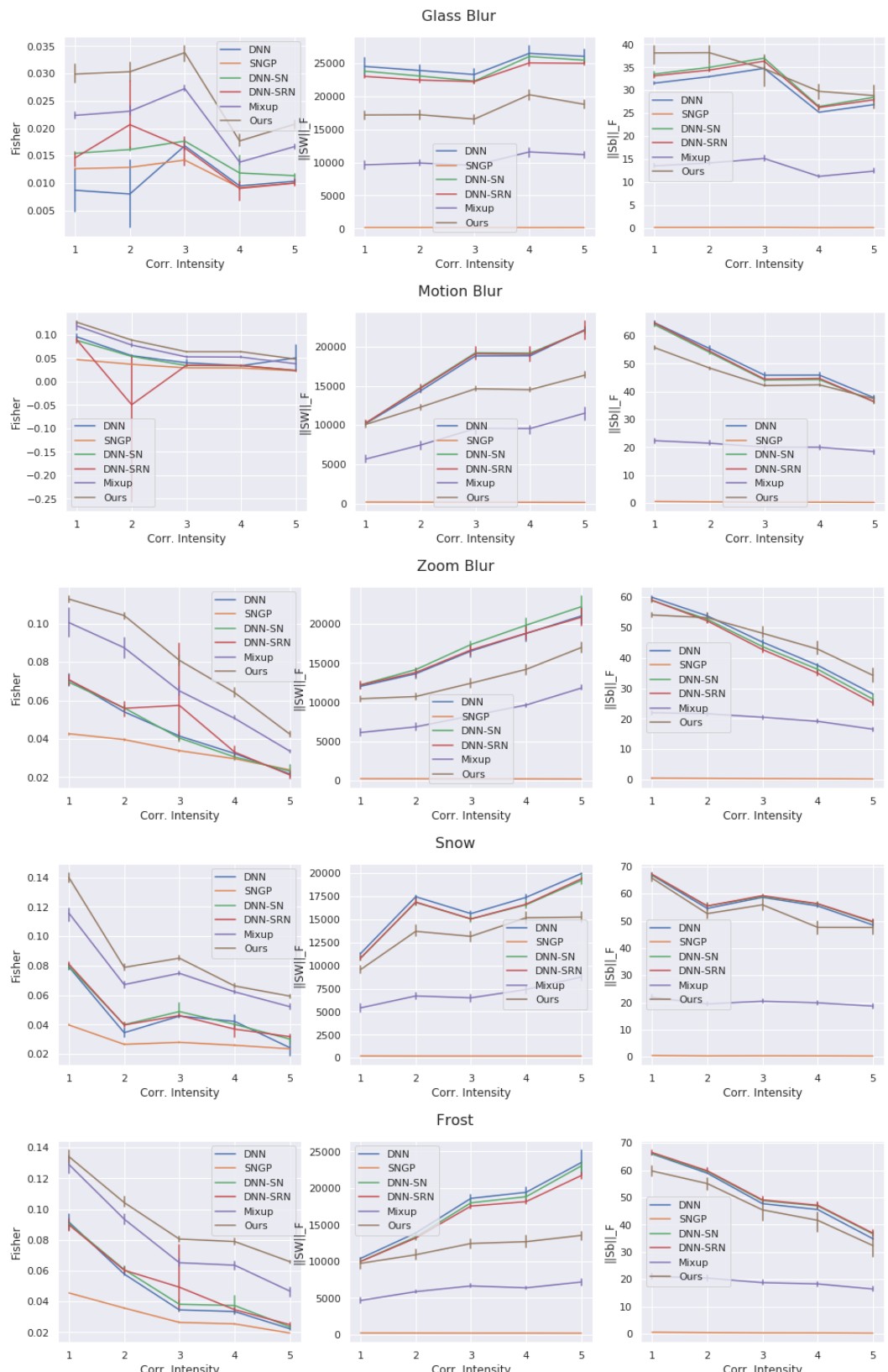

Figure 13: (Part 2 of 3) Fisher criterion, $||S_W||_F$ and $||S_B||_F$ for all corruptions and intensity values of CIFAR-10-C, architecture WideResNet28-10.

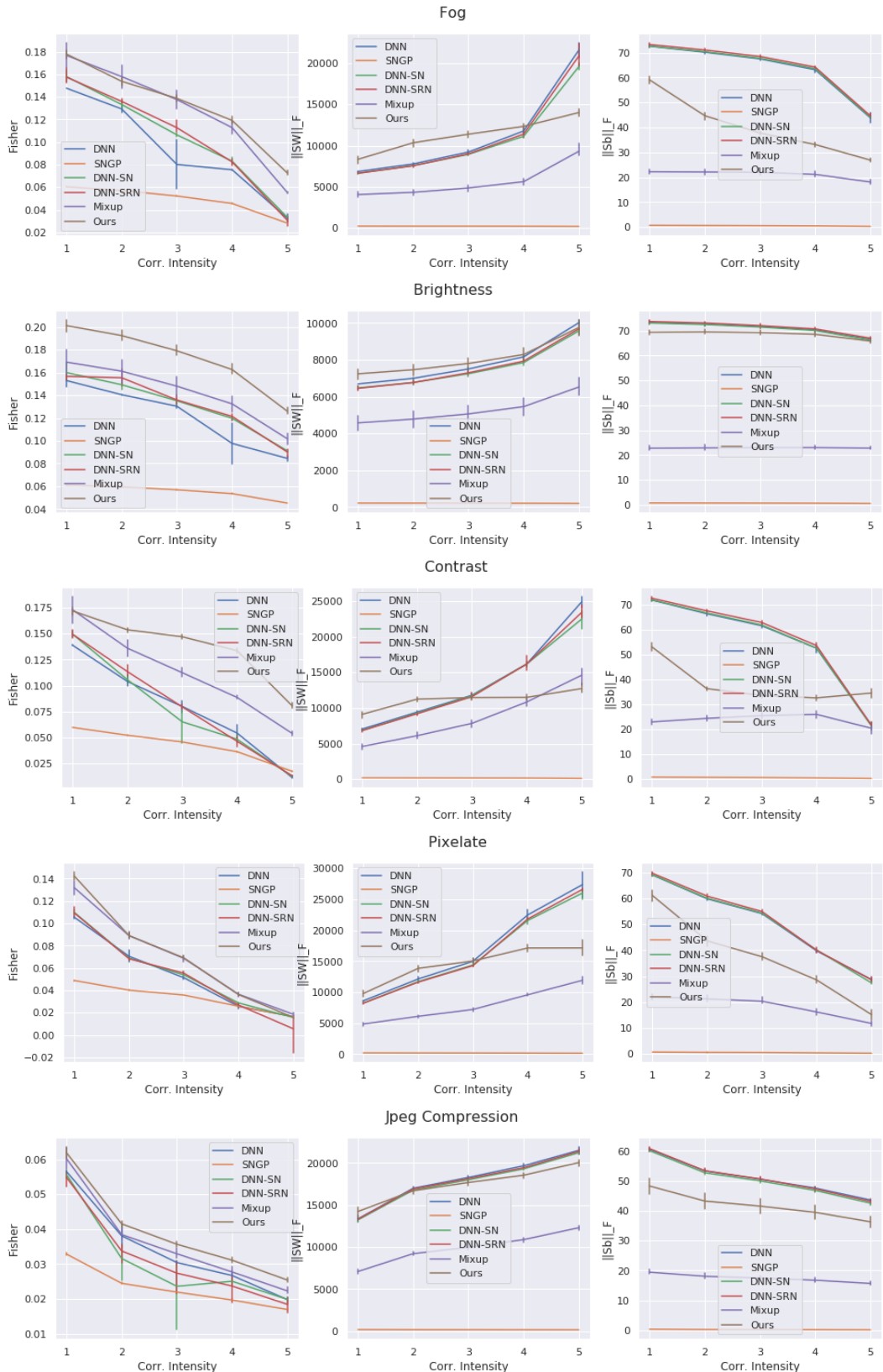

Figure 14: (Part 3 of 3) Fisher criterion, $||S_W||_F$ and $||S_B||_F$ for all corruptions and intensity values of CIFAR-10-C, architecture WideResNet28-10.

