# OpenReview forum: "Mix-MaxEnt: Creating High Entropy Barriers To Improve Accuracy and Uncertainty Estimates of Deterministic Neural Networks"
_ICLR.cc/2022/Conference — ICLR 2022 Submitted_

### Official Review · Reviewer_R5hm · 2021-10-18

**Correctness:** 3
**Technical Novelty And Significance:** 2
**Empirical Novelty And Significance:** 2
**Recommendation:** 6
**Confidence:** 4

**Main Review:**

My impression is that the proposed penalty is a slight variation over mixup, which is the major concern I have about the novelty of this paper. The authors explained the differences on page 5, but I found this paragraph hard to phrase and somewhat ambiguous. Below, I entail the comments I have after reading the paper.

Analysis and observations: Figure 2 (right) displays the histograms of the distances of the embeddings that correspond to CIFAR10 (for which the model was fitted) and CIFAR100. As illustrated, there is an overlap between the two histograms, where the one that corresponds to CIFAR100 is slightly shifted to the left. Now, I do not understand the intuition that the model “will produce stronger activations if the similarity between the features learned on the IND data and the ones present in the input is high.” This should be better explained, including a demonstration. Also, a table that summarizes the statistics of interest (evaluated on the entire test data) would be informative.

I believe that Figure 3 better illustrates this point. However, following Figure 7, mixup has an effect that is similar to the one presented in the right panel of Figure 3, although the “high entropy barrier” obtained by mixup is smoother compared to Min-MaxEnt.

While the data sets used are commonly used in this literature, the distribution shift considered is due to various corruptions in the input image, and therefore reflects the robustness of the fitted model to such deviations. I would suggest studying other types of domain-shifts, such as the one proposed in [1].

[1] Recht, B., Roelofs, R., Schmidt, L., Shankar, V.. (2019). Do ImageNet Classifiers Generalize to ImageNet? Proceedings of the 36th International Conference on Machine Learning.

Also, the OOD experiments are fairly easy, since CIFAR10 and SVHN are very different data sets. It will be great to explore other cases for which detecting OOD samples is more challenging. For example, using synthetic data can be helpful and informative [2]. In addition, it will be illuminating to compare the proposed method to existing techniques, such as [2].

[2] Liu, J. Z., Lin, Z., Padhy, S., Tran, D., Bedrax-Weiss, T., & Lakshminarayanan, B. (2020). Simple and principled uncertainty estimation with deterministic deep learning via distance awareness. arXiv preprint arXiv:2006.10108.

**Summary Of The Paper:**

This paper presents a new penalty function that can be augmented to the common cross-entropy loss, which was shown to improve the accuracy and calibration of deep net classifiers, applied to in-distribution data. This new loss also encourages the model to generate better estimates under domain-shift and better identify out-of-distribution samples.

The proposed regularizer is tightly connected to mixup. The latter suggests a data augmentation routine that creates new semi-synthetic samples by interpolating two images and their (possibly different) labels. In this paper, the idea is to interpolate between two images from different classes and then maximize the entropy on these samples during training. In a series of experiments, the authors show the advantage of this method over baseline approaches.


**Summary Of The Review:**

The paper offers an interesting approach to improve robustness to domain-shift and better detect OOD data. The major concern I have is about the novelty of the proposal and its similarity to mixup, especially because the latter was already studied in this context (calibration, OOD, and adversarial robustness). Here, I believe that the authors should do a better job in distinguishing between the two methods (and other existing extensions of mixup).

---

> ### Author Response · Authors · 2021-11-19
> **[Part 1/2] Thank you for the review, we address your concerns in two comments (references in second comment)**
>
> We thank the reviewer for appreciating the fact our model **improves accuracy and calibration on both in-distribution and data-shifted inputs**, and helps to **better identify out-of-distribution samples**. Below we provide answers to the questions asked by the reviewers.
>
> **Q1: My impression is that the proposed approach is a slight variation over mixup, which is the major concern I have about the novelty of this paper. The authors explained the differences on page 5, but I found this paragraph hard to phrase and somewhat ambiguous.**
>
> **Ans:** In this response we will clarify potential misunderstandings about our method and regarding the comparison of our technique with Mixup.
>
> **Objective function**: Here we show the difference between the objective functions of Mixup and Mix-MaxEnt (indicating with $y$ and $x$ the original dataset points, and $\bar{y}$ and $\bar{x}$ the interpolated points).
> - Mixup: == $\min_\theta - \log p(\bar{y}|\bar{x})$
> - MixMaxEnt == $\min_\theta - \log p(y|x) - H_{\bar{y}}(p(.|\bar{x})$
>
> As it’s clear from the objective functions, *mixup* is a *vicinal risk minimization* approach that, in effect, slightly perturbs the input and uses a small amount of label smoothing to **improve the performance in a small neighborhood** of the training set. Note, mixup was designed to avoid memorization.
>
> However, **Mix-MaxEnt** is a **regularization** based approach with (1) log-likelihood term over the clean samples; and (2) high-entropy regularizer over the interpolated samples. Note, there is **no use of clean samples in mixup objectives**.
>
> **Empirical behaviour**: Because of the fundamental differences between the objective functions of mixup and Mix-MaxEnt, their empirical behaviour also differs significantly.
> - Clean and data-shifted data **accuracy** of MixMaxEnt is higher than mixup
> - **Uncertainty estimates** of Mix-MaxEnt are more reliable
> - The best performing Beta hyperparameter of mixup and Mix-MaxEnt are very different. In the case of mixup, it is 0.3 as a higher value would mean a highly interpolated sample during training, implying degradation in clean data accuracy as the test data will be very different from the highly interpolated train data. However, the optimal Beta hyperparameter for Mix-MaxEnt is 30/20 as the interploated samples are used to regularize the log-likelihood that is being trained using clean samples.
>
> Note, the **only similarity** Mix-MaxEnt shares with Mixup is the **interpolation technique** (but we use a different distribution shape for the interpolation factor and choose the interpolation samples differently).
> At this link we provide an infographic summarising the comparison with mixup: https://tinyurl.com/bsfyccrw
>
> **Q2. Following Figure 7, mixup has an effect that is similar to the one presented in the right panel of Figure 3, although the “high entropy barrier” obtained by mixup is smoother compared to Mix-MaxEnt.**
>
> **Ans:** Note that the fact the entropy barrier of mixup is “smoother” is undesirable because it implies:
> - **mixup uniformly increases the entropy everywhere** (because it indiscriminately applies label smoothing both close and away from the training data). This explains the **worse OOD detection** performance of mixup (even with respect to DNN). Literature also already explained the improved calibration induced by Mixup as an effect of counterbalancing the typical overconfidence of NNs with **underconfidence** [1]; our previous observation confirms this behaviour.
> - on the other hand, **Mix-MaxEnt only increases the entropy away from the in-distribution data** hence the **better OOD detection** performance.
>
> These arguments show the methods are very different both in their motivation, concept and experimental performance: Mix-MaxEnt produces **better accuracies, calibration and OOD detection** with respect to Mixup in the vast majority of cases. There is no way we could tune the hyperparameters of Mixup to achieve the same performance of Mix-MaxEnt. We tried our best to do that.
>
> **Q3: “Missing comparison with [2]”**
>
> **Ans:** Note that the reference [2] that you mentioned is the **SNGP paper** that we have very well **cited and compared with** our method in the paper. Please refer to **Tables 1 and 2** in the paper. Our SNGP results are on WideResNet (similar to the SNGP original paper) as in the case of ResNet-50 (**Appendix C, Code section**), we could not manage to train SNGP to achieve SOTA accuracy,  and we preferred to omit poor results for SNGP. Note, the official code base does not support ResNet50 experiments on CIFAR-10/100 and the authors did not present results for ResNet50 in their paper.

---

> > ### Author Response · Authors · 2021-11-19
> > **[Part 2/2] Thank you for the review, we address your concerns in two comments (references in second comment)**
> >
> > **Q3: “Data-shift experiments focus on corruptions, you should consider other data shift types, such as the one proposed in [1].”**
> >
> > **Ans:** Thank you for this suggestion. Since performing ImageNet was not feasible given the limited time available, we performed additional experiments on **CIFAR10.1 and CIFAR10.2** [2,3] datasets that contain real-world-type data shifted examples for CIFAR-10. These experiments are in addition to the data shift experiments we performed in the main paper that cover 15 diverse types of corruptions at 5 intensity levels.
> >
> > The **results** on this dataset using WideResNet are presented at this link: https://tinyurl.com/mynhp2bx
> >
> > Note that **Mix-MaxEnt** on these experiments as well **outperform** extremely competitive baselines such as Deep Ensembles and Mixup not just in terms of accuracy, but also in providing more calibrated models (notice the variance of mixup in the case of CIFAR10.2 experiments).
> >
> > We could not find any other CIFAR data-shift datasets, but we will be more than **glad to execute any data-shift experiment the reviewer might suggest**, in case they are aware of any other data-shift test set for CIFAR-10/100.
> >
> > **Q4: I do not understand the intuition that the model “will produce stronger activations if the similarity between the features learned on the IND data and the ones present in the input is high”.**
> >
> > **Ans:** We believe this misunderstanding to be caused by a **typo**. In that paragraph we meant to say **kernels (filters), not features**. To be perfectly clear we **rephrase** our argument as follows:
> >
> > The application of a convolutional kernel on a part of the input produces a similarity score that represents how similar the input region and the kernel are. Kernels are trained to detect features contained in the training set. Since CIFAR-10 and SVHN look very different, it is very likely that the kernels needed to classify objects in CIFAR-10 are very different from the ones needed to classify objects in SVHN (hence, producing lower activations in the feature maps, and hence producing embeddings (i.e. flattened feature maps of the last convolutional layer) closer to 0). CIFAR-10 and CIFAR-100 are much more similar datasets, with some classes which are not distinguishable even by humans [4]. For this reason the OOD detection task is harder, and the similarity between the learned kernels and the input regions is stronger. For this reason, the activations are on average higher and the embeddings distance from the center can be similar to that of CIFAR-10 samples.
> >
> > To further support our statements, we report a histogram with CIFAR-10/100, Tiny-ImageNet (with several overlapping classes with CIFAR-10/100), FakeData (a dataset of gaussian noise) and CelebA. We also report the **boxplots** to provide some **statistics** information as requested. The boxplots confirm that OOD data is more shifted towards the center of the embedding space):
> >
> > https://tinyurl.com/4b7jrd3k
> >
> > https://tinyurl.com/ws8eadc
> >
> > **Q5: “Show synthetic examples experiments”**
> >
> > **Ans:** Synthetic low dimensional examples do not reflect the complexity of real-world data, and **solutions that fit perfectly synthetic data do not generalise to real-world data**. An **example** would be **SNGP** synthetic experiments where almost perfect performance is shown, yet it produces far from perfect OOD detection on real-world datasets. Having said that, we will definitely consider your suggestion of performing additional synthetic experiments to further highlight the effectiveness of our approach.
> >
> > We hope the reviewer will consider our response useful and exhaustive. Please let us know if anything is unclear or needs further elaboration.
> >
> > [1] Combining Ensembles and Data Augmentation Can Harm Your Calibration Yeming Wen, Ghassen Jerfel, Rafael Muller, Michael W Dusenberry, Jasper Snoek, Balaji Lakshminarayanan, Dustin Tran
> >
> > [2] Do CIFAR-10 Classifiers Generalize to CIFAR-10?, Benjamin Recht and Rebecca Roelofs and Ludwig Schmidt and Vaishaal Shankar
> >
> > [3] Harder or Different? A Closer Look at Distribution Shift in Dataset Reproduction" by Shangyun Lu, Bradley Nott, Aaron Olson, Alberto Todeschini, Puya Vahabi, Yair Carmon and Ludwig Schmidt
> >
> > [4] Exploring the Limits of Out-of-Distribution Detection, Stanislav Fort, Jie Ren, Balaji Lakshminarayanan

---

> > > ### Comment · Reviewer_R5hm · 2021-11-25
> > > **Follow-up**
> > >
> > > Thank you for the detailed response, especially for (i) clarifying the differences between mixup and the proposed regularizer; and (ii) conducting experiments with CIFAR10.1 and CIFAR10.2 data sets. The authors addressed the major concerns I've had in the initial review. Therefore, I lean towards accepting the paper.

---

> > > > ### Author Response · Authors · 2021-11-25
> > > > **Your reviews indeed were helpful. Thank you for your time.**
> > > >
> > > > Thank you for reading our replies and appreciating our efforts. Your reviews indeed have helped us  improve our paper. Thank you!
> > > >
> > > > We will include the additional experiments performed in the final draft. Please let us know if you have any questions. We will do our best to address them

---

> ### Author Response · Authors · 2021-11-23
> **Improved Mixup vs Mix-MaxEnt paragraph and fixed typo in revised version**
>
> We have updated our draft to make the mixup vs mix-maxent paragraph more clear, we hope the reviewer will find the new phrasing more readable. We also fixed the kernel/features typo, and are willing to further improve that paragraph in case it is not enough to make our statements clear enough. Upon acceptance of the paper, we will include the new data-shift experiments in further tables.

---

### Official Review · Reviewer_ESue · 2021-10-20

**Correctness:** 3
**Technical Novelty And Significance:** 3
**Empirical Novelty And Significance:** 3
**Recommendation:** 3
**Confidence:** 5

**Main Review:**

Increasing entropy to improve OOD detection performance is not very novel. For example, the Agnostophobia [1] and Outlier Exposure [2] approaches start from a similar idea. The advantage of the proposed method is the fact that it avoids collecting additional data as required by the Agnostophobia and Outlier Exposure approaches.

However, the Mixup [3] approach also uses data augmentation to improve robustness. Similar to the proposed approach, it also does not require collecting extra data. Despite both approaches being based on data augmentation, the authors present some differences between the proposed approach and Mixup. We partially agree with the authors in this case. This is not our main concern regarding the paper.

The IsoMax loss [4,5,6] and the Enhanced IsoMax (IsoMax+) loss [7] significantly also uses the strategy of improving the OOD detection performance by maximizing the entropy with the advantage of avoiding data augmentation and also massive hyperparameter tuning. For example, ten-fold cross-validation used to search five hyperparameters values requires training the same model 50 times. Rather than a regularization term, the IsoMax loss and the IsoMax+ loss are full losses used to seamlessly and entirely replace the SoftMax loss (combination of the cross-entropy loss, the SoftMax activation, and the last linear layer). Therefore, considering that the paper is proposing a loss regularization term to improve entropy in some cases, it must cite the IsoMax loss and IsoMax+ loss papers. The IsoMax losses work by simply replacing the SoftMax loss.

Additionally, it needs to change the following sentence because it is no longer valid after the advent of the IsoMax loss and the IsoMax+ loss: _"While the literature suggests that classifiers whose predictive distributions increase their entropy the further away the test input gets from the training data are desirable, the implementation of models satisfying such a property is not scalable, and can only occur at the cost of performing crude approximations and by non-trivial modifications to the architecture of the neural network (Liu et al., 2020a; Kristiadi et al., 2020). Such modifications sometimes lead to degraded accuracy."_

The SoftMax loss is an extremely easy baseline for the OOD detection task. Hence, the proposed method should combine the proposed method with IsoMax and Enhanced IsoMax (IsoMaxPlus or IsoMax+). This could be done simply by adding the proposed regularization term that was added to the SoftMax loss. By using the code provided with the IsoMax papers, the suggested experiments could be performed by changing two lines of code. Indeed, combining the proposed term with the IsoMax loss variants (i.e., IsoMax and IsoMax+) would be appropriate to allow us to compare with a much-improved OOD detection baseline.

Additionally, we would evaluate whether IsoMax variants are better than SoftMax loss when combined with the proposed approach from an uncertainty estimation point of view. Adding these additional experiments using an improved baseline loss rather than the SoftMax loss appears essential. One final reason to include IsoMax and IsoMax+ in the experiments is the fact that they are distance-based losses, which the community is starting to agree is the best approach for OOD detection. Hence, it would be great to add one line called "IsoMaxPlus" and other "IsoMaxPlus+Min-MaxEnt" to Tables 1, 2, 5, and 6.

Finally, considering that the Tables showing experiments with ResNet50 are as important as experiments using WideResNet28-10, to give a better notion of the real generalization power of the approach, we recommend moving Tables 5 and 6 to the main paper rather than being presented in the appendix. Additionally, section 5.1.4 should also consider the results obtained for the ResNet50 model. Considering the ResNet50 results, we believe the solution is attractive but lacks some generalization power (see ResNet50 results in the appendix). Furthermore, it is extremely costly, as we need to train the same model too many times to perform hyperparameter tuning.

The approach needs to improve the baseline performance of the IsoMax and IsoMax+ losses when combined with them to be valuable. Otherwise, we could simply use the IsoMax+ and improve the OOD detection performance training just once by avoiding the costly hyperparameter tuning.

The authors could be a bit more explicit about the fact the proposed approach requires increased training times because we need to add augmented data to the training process.

_"For all the methods, the ECE and AdaECE are computed after performing temperature scaling (Guo et al., 2017) with a **cross-validated temperature parameter**"_ and _"when temperature scaling is applied, **the temperature T is tuned on the validation set**, minimising the ECE (we considered values ranging from 0.1 to 10, with a step size of 0.01)"_ seem contradictory. Considering _"Such hyperparameters have been selected performing cross-validation with stratified-sampling on a 90/10 split of the training set to maximise
Accuracy"_, we believe the second sentence is true. Please, could you clarify?

[1] Reducing Network Agnostophobia: https://arxiv.org/abs/1811.04110

[2] Deep Anomaly Detection with Outlier Exposure: https://arxiv.org/abs/1812.04606

[3] On Mixup Training: Improved Calibration and Predictive Uncertainty for Deep Neural Networks: https://arxiv.org/abs/1905.11001

[4] IsoMax loss (first preprint, 2019): https://arxiv.org/abs/1908.05569v1

[5] IsoMax loss: https://arxiv.org/abs/1908.05569

[6] IsoMax loss (journal version): https://arxiv.org/abs/2006.04005

[7] IsoMax+ loss: https://arxiv.org/abs/2105.14399

David Macêdo


###########################################################################################
###########################################################################################
###########################################################################################
###########################################################################################


===== FINAL RECOMMENDATION POST-REBUTTAL ========

NOVELTY

The paper presents some level of novelty. It is similar to Mixup, but the differences are relevant. As we said before, this is not our main concern regarding the paper.

**Hence, overall, the novelty is regular.**

EXPERIMENTS

The paper presents many types of experiments such as uncertainty estimation, domain-shift, and out-of-distribution detection. However, the OOD data initially considered were only CIFAR and SVHN. We did not find results for the other methods for the new OOD data.

**Hence, overall, the experimentation design is ok, but it may be improved regarding OOD detection.**

PERFORMANCE

Considering WideResNet, the proposed approach generally outperforms other approaches regarding uncertainty estimation and domain-shift. **However, these results does not appear consistent if we consider the ResNet results presented in the appendix. Moreover, it usually performs similar to others regarding OOD detection and commonly does not even outperform regular neural networks in WideResNet.**

**Hence, overall, the performance of the approach is not so great, and the results are not very consistent for different types of models.**

CLARITY

**ResNets are much more largely used and relevant than WideResNets. Unfortunately, the results of the proposed approach are worse on ResNets than WideResNets. We see no reason to present the ResNet results in the appendix.**

We complained about this in our review. However, the authors did not answer.

SCALABILITY

**The proposed approach has a severe limitation: it appears too expensive to be used in real-world projects.**

To tune hyperparameters, the authors did 10-fold cross-validation and searched for five values. This means they need to train the same network using the same data (with just a different hyperparameter) 50 times. Considering that they use the same amount of augmented data as the number of training examples, the method also doubles the GPU memory requirements or training time.

**Therefore, overall, the proposed solution is about one hundred times more expensive than training a regular neural network.**

Hence, the proposed approach is viable for toy datasets like CIFAR. However, for real-world images such as ImageNet, it is essentially inviable. For example, let's say the cost to train a given big Transformer using a large amount of data is about 1 million dollars. If we plan to use the proposed approach, the total cost will be 100 million dollars. Same for CO2 emissions.

**All this for not very clear or consistent performance improvements.**

We complained about this in our review. However, the authors did not answer.

RECOMMENDATION

**Considering the above, we recommend REJECTION.**

FUTURE

The main idea is interesting, but we suggest that the authors remove the hyperparameter dependence and improve the OOD detection performance. We believe that combining the proposed approach with the IsoMax+ may help in this investigation.


###########################################################################################
###########################################################################################
###########################################################################################
###########################################################################################



**Summary Of The Paper:**

The paper proposes to add a regularization term to the loss used to train the neural network. The mentioned term imposes high entropy on synthetically generated examples to improve accuracy, uncertainty estimates, and OOD detection performance. In other words, it proposes an "entropy maximization regularizer." The paper also has the strength of performing a significant amount of experiments that includes uncertainty estimation, domain-shift, and out-of-distribution detection.

**Summary Of The Review:**

The proposed method is interesting, but the novelty is regular as it is similar to Mixup and the performance is not too impressive when we also consider ResNet50 results. Moreover, the paper completely ignores the IsoMax loss and the IsoMax+ loss that are significantly related solutions. Hence, considering that the IsoMax loss variants already significantly increase the OOD detection performance without requiring costly hyperparameter tuning, this proposed regularization term would be more valuable if the authors show it also increases the ECE, Domain-Shift and OOD detection performance of the IsoMax loss and IsoMax+ loss when combined with them. Finally, it is possible that combining IsoMaxPlus with Min-MaxEnt may produce state-of-the-art results and also make the proposed approach work better (i.e., producing more stable results) when dealing with the ResNet50 model.

---

> ### Author Response · Authors · 2021-11-21
> **[Part 1] Clarifications about our contributions and differences with mixup**
>
> First of all, thank you for taking the time to read our paper and provide comments. Before we provide our reply to the questions asked by the reviewer, we would like to **reiterate our contributions**:
>
> - We do **not claim any novelty in the concept of maximum entropy**, our **novelty** lies in the **way we regularize** log-likelihood solutions to prefer a solution that is uncertain as we go outside the data-distribution. And we do this without using any out-of-domain data points. This fact should not be ignored.
> - We provide **extensive experiments** using new **strong baselines** and also using recently published OOD works.
> - We compare various OOD metrics so that we provide the best result using each of the baselines we compare with.
> - We also analyze using Fisher metric to show that our approach provides **more compact and separated clusters**.
> - Our approach does **not require any architectural changes and is extremely easy to code**.
> - It provides extremely **good accuracy** (often better than Deep Ensembles) and extremely good or competitive **calibration and OOD performance**. Note, most OOD approaches underperform in the accuracy aspect.
>
> It looks like the **reviewer’s main concern is the fact that we did not refer or compare with the IsoMax paper**. One minor comment from the reviewer was regarding the **similarity of our approach with mixup**. Though this comment was minor, below we would like to provide a proper description of how our approach differs from mixup, and then we will provide a detailed reply regarding other comments.
>
> Our work **differs from mixup** in:
>
> **Objective function**: Here we show the difference between the objective functions of Mixup and Mix-MaxEnt (indicating with $y$ and $x$ the original dataset points, and $\bar{y}$ and $\bar{x}$ the interpolated points).
> - Mixup: == $\min_\theta - \log p(\bar{y}|\bar{x})$
> - MixMaxEnt == $\min_\theta - \log p(y|x) - H_{\bar{y}}(p(.|\bar{x})$
>
> As it’s clear from the objective functions, *mixup* is a *vicinal risk minimization* approach that, in effect, slightly perturbs the input and uses a small amount of label smoothing to **improve the performance in a small neighborhood** of the training set. Note, mixup was designed to avoid memorization.
>
> However, **Mix-MaxEnt** is a **regularization** based approach with (1) log-likelihood term over the clean samples; and (2) high-entropy regularizer over the interpolated samples. Note, there is **no use of clean samples in mixup objectives**.
>
> **Empirical behaviour**: Because of the fundamental differences between the objective functions of mixup and Mix-MaxEnt, their empirical behaviour also differs significantly.
> - Clean and data-shifted data **accuracy** of MixMaxEnt is higher than mixup
> - **Uncertainty estimates** of Mix-MaxEnt are more reliable
> - The best performing Beta hyperparameter of mixup and Mix-MaxEnt are very different. In the case of mixup, it is 0.3 as a higher value would mean a highly interpolated sample during training, implying degradation in clean data accuracy as the test data will be very different from the highly interpolated train data. However, the optimal Beta hyperparameter for Mix-MaxEnt is 30/20 as the interploated samples are used to regularize the log-likelihood that is being trained using clean samples.
>
> Note, the **only similarity** Mix-MaxEnt shares with Mixup is the **interpolation technique** (but we use a different distribution shape for the interpolation factor and choose the interpolation samples differently).
> At this link we provide an infographic summarising the comparison with mixup: https://tinyurl.com/bsfyccrw

---

> > ### Author Response · Authors · 2021-11-21
> > **[Part 2] Our take on IsoMax and why we do not consider it as a baseline**
> >
> > Since one of the main concerns of the reviewer is that **we did not mention IsoMax(+)** and did **not compare** our approach with it, here we provide a detailed reply to the reviewer about it. We mention **what we think about the paper**, and then provide **additional experimental results using IsoMax+**.
> >
> > **We did not come across IsoMax or IsoMax+ and we think we have a couple of reasons for that.**
> >
> > Though the reviewer provided four different entries of the same work on arxiv (with some modifications),  we would like to mention that we did not come across this work perhaps because it was **never published in any top venue known to us**, or perhaps because it was **not extensively cited** in the research literature we analysed.
> > It looks like it was **very recently accepted to a journal** and the arxiv version of the journal has been posted in August 2021. Honestly, because of the lack of time, we could not read all the many different versions and variants of the paper to figure out which version did what at what point in time. In any case, we would like to thank the reviewer for pointing out this paper to us. Below you will find our views about the paper.
> >
> > **Our take on the technical aspect of Isomax**
> >
> > Since there are many versions of this paper, we had a look at the one that was very recently published in “IEEE TRANSACTIONS ON NEURAL NETWORKS AND LEARNING SYSTEMS: SPECIAL ISSUE ON DEEP LEARNING FOR ANOMALY DETECTION”. This is what we think about the paper:
> >
> > - The arguments presented in the paper are mostly **handwavy**. There is **no justification behind many statements**. For example:
> > -- Even the first line of the abstract that says “poor OOD performance of NNs is mainly due to softmax loss” is **not supported with any theoretical insight or proper experimental analysis**. And it can just be wrong because there are so many other factors such as overparameterization, choice of optimizer, architecture, kind of nonlinearities, various inductive biases etc. playing important roles towards NNs performance (good or bad).
> > -- Replacing the last linear layer (affine) with Euclidean distance in eq (2) does not seem that compelling. The linear layer can be seen as a proxy to the perpendicular distance of a sample from each of the hyperplanes, and the Euclidean distance is also very similar (expand the equation). The only difference would be some scaling because of the norm of each prototype. I don’t see much fundamental difference here and the paper also does not explain it well enough.
> > -- Just replacing affine layer with Euclidean distance does**not necessarily make the uncertainty distance aware (as the literature agrees to be the gold standard)**. We **could not find any theoretical or empirical study in the paper that supports that the uncertainty of IsoMax is distance-aware**.
> > -- The paper says “We use the principle of maximum entropy as a theoretical motivation for constructing high entropy (low confidence) posterior probabilities” -- **which posterior probability is it talking about? We don’t see any Bayesian formulation justifying such statements**. And, for **maximum entropy**, a scalar is being multiplied throughout during training (eq 6) to be removed at test time. This is an extremely **ad-hoc trick**. We don’t see any principle here as claimed earlier. Also, obviously, if one multiplies everything with a positive scalar before applying softmax, depending on its value, it will increase or decrease the entropy.
> > -- Entropic score -- not sure why this paper is calling eq (11) a “negative entropy”? Also, using entropy for uncertainty estimation is extremely common so I believe there is no claim of any sort of contribution here.
> >
> > Also in terms of experiments, the paper is lacking important experiments:
> > - There is **no** report of **calibration** experiments, neither on **in-distribution** data nor on **data-shifted** inputs, and also the OOD performance is **not compared against any recent approaches**.
> >
> > **New experiments using Isomax+**
> >
> > To satisfy the curiosity of the reviewer, we ran experiments with **IsoMax+** and the results are at the following link:
> >
> > https://tinyurl.com/72b739ah
> >
> > We observed that while the **OOD performance** of IsoMax+ is reasonable in **some** cases, the **calibration** of IsoMax+ is **always extremely poor**. Additionally, on **data-shift** experiments, it performed **very poorly too**. On top of this, IsoMax+ also suffered a **drop in accuracy**. Because of this, **we do not consider IsoMax to be a strong baseline for reliable uncertainty estimation**.
> >
> > We would like to mention to the reviewer that **just performing well on OOD is not good enough**. The model must perform very well on in-distribution calibration, accuracy, and domain-shift experiments as well for it to find a place among approaches providing reliable uncertainty estimation. A method that achieves good OOD detection but completely **destroys calibration** makes the model’s predictions completely unreliable.

---

> > > ### Author Response · Authors · 2021-11-21
> > > **[Part 3] Additional replies to the reviewer**
> > >
> > > **Clarity regarding “The Softmax loss”**
> > >
> > > We are assuming that by softmax loss the reviewer means the cross-entropy loss (standard DNN) as the softmax is not a loss function. The IsoMax paper also uses the term SoftMax loss which is a bit confusing to us.
> > >
> > > **C1: “Softmax loss is an extremely easy baseline ..”**
> > >
> > > **Ans:** Please note that we do not only compare against the standard DNN. We try to improve the performance of DNN with low Lipschtiz regularizers (spectral norm and stable rank normalization) and also compare with **Deep Ensembles** (which is an ensemble of DNNs and is one of the **strongest baselines for OOD**) along with other **recent OOD approaches**.
> > >
> > > **C2:  “For all methods, the ECE and AdECE are computed … with a cross-validated temperature parameter and …, minimising the ECE.. Considering Such .. cross-validation with stratified sampling… to maximise accuracy”**
> > >
> > > **Ans:** **Temperature scaling** does not change the prediction, so it **cannot change the accuracy**. For this reason it cannot be cross-validated on the accuracy. It’s common practice to cross-validate that hyperparameter using the **ECE or the NLL**. All other parameters are cross-validated on accuracy. We hope it is clear now.
> > >
> > > **C3: “Additionally, it needs to change the following sentence because it is no longer valid after the advent of the IsoMax loss and IsoMax+ loss: While the literature suggests … “**
> > >
> > > **Ans:** We would like to request the reviewer to point out **where in the IsoMax or its variants does the paper show theoretically or experimentally that as the input gets further away from the data the entropy increases**. We could not find this in the paper. Merely changing the loss from inner product to Euclidean distance (as done in IsoMax) does not guarantee anything related to the distance awareness of the uncertainties.
> > >
> > > **C4: One final reason to include IsoMax and IsoMax+ in the experiments is the fact that they are distance-based losses, *which the community is starting to agree* is the best approach for OOD detection.**
> > >
> > > **Ans:** We respectfully disagree with the reviewer as there is a whole lot of work that people have been doing for decades towards developing distance aware methods as they are fundamentally very important for reliable uncertainty estimation (e.g., Gaussian processes for outlier detection). The **recent trend is not to realise that distance awareness is important, it is more to make it feasible by designing plausible approximations**. Therefore, indicating that the community recently started to agree that distance-based approach is important for OOD would be an attempt to overpraise IsoMax.
> > >
> > > **C5: “it would be great to add one line called "IsoMaxPlus" and other "IsoMaxPlus+Min-MaxEnt" to Tables 1, 2, 5, and 6.”**
> > >
> > > **Ans:** Again, we respectfully disagree with the reviewer. **IsoMax** does **not** seem to be a **theoretically motivated and experimentally very well tested** approach for reliable uncertainty estimations. Also, based on our experiments, the approach turns out to be **extremely miscalibrated**. Therefore, we do not believe that we should include it in our work at this stage and compare our work with this approach and create new unnecessarily complicated variants such as “IsoMaxPlus+Min-MaxEnt”.

---

> > > > ### Comment · Reviewer_ESue · 2021-11-21
> > > > **Some brief comments**
> > > >
> > > > **The authors did not appear to have read the IsoMax papers carefully.**
> > > >
> > > > If they plan to review those papers, they should have read them carefully, as we did with their paper.
> > > >
> > > > For example, regarding "the SoftMax loss" terminology, all IsoMax papers made it very clear that it was created many years before in the paper "Large-margin softmax loss for convolutional neural networks." International Conference on Machine Learning, 2016. See the Figure 1.
> > > >
> > > > Other example, the Figure 1 of the IsoMax+ paper shows very clearly that entropies increase when the distances to prototypes increase.
> > > >
> > > > Just one more, it is clear in the papers that temperature validation is necessary to calibrate the IsoMax trained models. Similar to SoftMax loss trained networks, it is a very easy and fast procedure that significantly increase the calibration of such models.
> > > >
> > > > **Therefore, considering that the authors did not carefully read the mentioned papers (and it is not our job here (nor theirs) to review the IsoMax articles), we will not lose our time responding to the authors' criticism of the cited papers.**
> > > >
> > > > We will focus on updating our review of their paper, considering their responses.

---

> > > ### Comment · Reviewer_ESue · 2021-11-21
> > > **Some brief comments**
> > >
> > > _Regarding the IsoMax papers:_
> > >
> > > **The theoretical insight is the principle of maximum entropy, which is directly related to Bayesian statistics. There are many references in the paper connecting these two ideas. The experimental evidence is overwhelming, as we have three papers and a set of experiments repeatedly done in very different situations.**
> > >
> > > _Regarding the completely nonsense experiments with IsoMax+:_
> > >
> > > IsoMax+ does not produce classification accuracy drop. **The problem is that the authors are comparing very different things.** The first thing they should know is that IsoMax+ loss is a replacement to the Softmax loss with focus on OOD detection. Moreover, they should also understand that IsoMax+ must be combined with other techniques to tackle uncertainty and distribution shift.
> > >
> > > The problem is the authors added data augmentation to their technique to increase the accuracy of their approach. Add Mixup (or combine with deep ensembles) to IsoMax+, and the small gap in classification accuracy to IsoMax+ will be gone. The same thing to distribution shift, add Mixup (or combine with deep ensembles), and the gap will also be gone.
> > >
> > > Same thing regarding calibration. The authors trained their model ***50 (FIFTY) times*** but did not validate the temperature of the IsoMax+ to calibrate it! Temperature calibration is a much faster procedure than training the same network using the same data ***50 (FIFTY) times*** to find optimal hyperparameters!
> > >
> > > ########################################################
> > > ########################################################
> > >
> > > **In summary, the results presented by the authors regarding IsoMax+ are _completely meanless and entirely misleading_ because they were obtained using extremely wrong assumptions and very poor understanding of the IsoMax+ loss purpose. The only exception is the OOD detection results, in which IsoMax+ OUTPERFORMS their solution training the neural network just once rather than 50 (FIFTY) times.**
> > >
> > > ########################################################
> > > ########################################################
> > >
> > > ___
> > >
> > > _**Therefore, if you only need in your application OOD detection, we get better results training the network once (rather than 50 (FIFTY) times) using IsoMax+. This is really a remarkable result in favor of the IsoMax+.**_
> > >
> > > _**If you need calibrated results, fast calibrate the temperature of the IsoMax+ loss (or combine with deep ensembles or Mixup) and avoid training the network 50 (FIFTY) times. The same thing (combine with deep ensembles or Mixup) if you need to improve distribution shift accuracy and want to avoid training the network 50 (FIFTY) times.**_
> > >
> > > ___
> > >
> > > ***Unfortunately, the authors did not understand that the IsoMax losses are ENTIRELY COMPLEMENTARY to their approach, as they are replacements to the SoftMax loss, while Mix-MaxEnt is a regularization term that may be applied to the SoftMax loss or the IsoMax losses. This is the reason why we suggest combining the IsoMax losses with their approach. We hope someday they finally realize this point.***

---

> > > ### Comment · Reviewer_ESue · 2021-11-24
> > > **Temperature calibration significantly improves IsoMax+ uncertainty estimation.**
> > >
> > > While the authors reported 82.69%, **after a fast temperature calibration**, we obtained ECE below 1% for clean data using IsoMax+ loss training a WideResNet28-10 on CIFAR10.
> > >
> > > We needed to make this point clear to **vehemently refute** the authors' allegation that IsoMax+ "destroys calibration."
> > >
> > > The OOD detection performance of IsoMax+ is not affected because all distances (which are the logits of IsoMax+ and work as score) are equally divided by the calibrated temperature.

---

> ### Comment · Reviewer_ESue · 2021-11-24
> **===== FINAL RECOMMENDATION POST-REBUTTAL ========**
>
> We edit our review to include our final recommendation post-rebuttal.

---

> ### Author Response · Authors · 2021-11-25
> **We respectfully disagree with your assessment of our paper**
>
> Thank you for your time.
>
> We **respectfully disagree** with many points you made regarding our paper. Perhaps these points reflect some misunderstanding of our work. Please find below few points regarding your comments:
> - **K-fold cross-validation** for hyper-parameter tuning is not an integrated component of our approach. Performing it is **not a drawback specific to Mix-MaxEnt**, it is an experimental methodology with the purpose of **avoiding overfitting the choice of the hyperparameters on the test set**.
> - We did **not perform 10-fold cross-validation, we never claimed we did**. We do *one* stratified 90/10 split of the train data for cross-validation. Instead of cross-validating the hyper-parameter on the test set, we use the 10% split for this. We use the same 90/10 split for all the experiments, including for post-training techniques (e.g. temperature scaling, KFAC-LLLA etc.)
> - Our approach has **only one hyper-parameter just like most other approaches**. In fact **many SOTA methods have more than one**.
> -- Our hyperparameter is very **easy to cross-validate** since we already have a fairly **good idea of the range it should be searched for**. Specifically, $\alpha > 1$ so that $\lambda_0 \approx 1/ 2$. The ablations also suggest that while the main contributions are given for $\lambda_0 \approx 1/2$, the tails of the beta distribution can be useful to further improve the performance, and therefore setting $\alpha$ too high might be suboptimal.
> -- In our preliminary experiments, we had observed our method was relatively stable, and **choosing $\alpha \approx 20$ most of the time produced significant improvements**. To carry out a scientifically valid analysis, we performed **cross-validation** to follow adequate evaluation procedures and **not inflate our performance by overfitting the hyperparameter to the test set**.
> - The claim that our approach would be 50/100 times slower than the baselines is evidently false. Training a Mix-MaxEnt model is nearly **1.5x slower than vanilla DNN at training time, with only 2x memory requirements. However, inference is as fast as the vanilla DNN**. We do not profile all the baselines but obviously one of the main competitors, **Deep Ensembles, is 5x slower or requires 5x the memory** (if **5 members** of the ensembles are chosen) both at training and inference.
> - The models we consider for our experiments can commonly be trained on ImageNet too. **Our experiments** are **extensive** and discuss the accuracy and uncertainty behaviour of our model in a **variety of settings and to an extent that is not common in literature**. We also extensively analyse the evaluation metrics themselves where appropriate. We do not only apply these analyses to our methodology, but to all the baselines (where possible). This already required a **huge effort**, especially considering many popular methods we evaluated (and discarded as baselines) do not even scale to CIFAR-100.
> - Regarding **ResNet experiments**, we would like to clarify that WideResNet architecture is extremely common in OOD/data-shift papers and most of them consider only one architecture (e.g., the SNGP paper to mention one), whereas we evaluate on both to provide a better view. Our method **still attains improvements** similar or superior to most of the **single-model baselines**, in a few cases it underperforms, but the small gap in a few tasks still makes it competitive across the many tasks for which improvements are observed.
>
> Finally, regarding **IsoMax(+)**, we stand with our earlier evaluation of the paper. The **experiments** we carried out **did not convince us to perform further experiments** and dig deeper. The technical novelty is also very limited. Therefore we do not believe the paper is worth citing yet. **The reviewer has agreed that IsoMax per se does not yield any data-shift improvement and that causes accuracy drops** on in-distribution data, but it is **not our responsibility to modify it** to fix this behaviour, especially if we are **requested to do it by combining it with other methods (which would mask these flaws)**. This is **not even the appropriate context** for this kind of analyses: the **authors of the IsoMax** papers perhaps **should take care** of these issues and **update their papers with adequate experiments, improved writing, and proper theoretical justifications.** We might consider citing these papers after these updates.
>
> Thank you once again for your time.

---

> > ### Comment · Reviewer_ESue · 2021-11-26
> > **Some additional clarifications**
> >
> > **We never acknowledge that IsoMax+ produces an accuracy drop regarding training using a regular cross-entropy (SoftMax) loss.**
> >
> > In the opposite direction, what we said is that its accuracy may even be increased if combined with Mixup, as we all know by the Mixup paper that it clearly improves accuracy **relative to training without it.** This is why we see no novelty in increasing the accuracy by the similar approach proposed by the authors. The data augmentation component is responsible for improving the accuracy in both cases.
> >
> > **_Unlike the proposed approach, training using IsoMax+ loss is as fast and as easy as training using the usual cross-entropy (SoftMax) loss._**
> >
> > **Therefore, IsoMax+ combined with Mixup would be a fair comparison with the proposed approach, as the proposed approach is essentially a variation of Mixup and presents similar additional computational cost and complexity. Hence, it is completely misleading saying that IsoMax+ presents classification accuracy drop relative to the proposed approach.**
> >
> > Additionally, we remember that **IsoMax+ easily outperforms the proposed approach in OOD detection, which unfortunately is not in fact even consistently better than a regular neural network in this regard (see Table 1, 2, 5, and 6).**
> >
> > _**By the way, Deep Ensembles should be bold rather than your approach in Table 5 regarding OOD for SVHN. Additionally, Table 5 and 6 show that Deep Ensembles outperform the proposed approach (almost) everywhere for ResNets. Why those tables are in the Appendix?**_
> >
> > Regarding calibration, the authors were extremely irresponsible in presenting results without calibrating IsoMax+ while they did this for all other approaches. **In five minutes, we calibrated the temperature of the IsoMax+ to achieve similar or better uncertainty estimations than the presented by the proposed approach.**
> >
> > Besides all this, IsoMax+ is **hyperparameter-free** and **much easier to implement and use than the proposed approach.** You do not waste your time searching for hyperparameters, and the training does not require increased memory resources or time to complete. Finally, **while the WideResNet results of the proposed approach are mixed, the ResNet results are even less convincing, and they should not be presented in the appendix.**
> >
> > **Moreover, considering that you did not show experiments using ImageNet, we can not be assured that the hyperparameters range you looked for using CIFAR will also work fine for ImageNet.**
> >
> > ############################################################
> >
> > ############################################################
> >
> > _**In summary, if your application requires OOD detection and calibration, we recommend simply training the network once using IsoMax+ followed by a fast temperature calibration post-training** rather than performing hyperparameter search and all ad-hoc complexity of the proposed approach. Additionally, **you obtain much better OOD detection performance** and calibration than using the proposed approach, **training your neural network only once!** That is it._
> >
> > _**If you also desire increased accuracy relative to a regular trained neural network and improved domain-shift, add Mixup to your setting (IsoMaxPlus+Mixup). It is just as simple as that**. Therefore, you will have **the best of all worlds**: improved accuracy (relative to a neural network trained without a specialized data argumentation like Mixup), domain shift, uncertainty estimation, and **much better** OOD detection performance. **This is a fair comparison with the proposed method in terms of computational complexity and cost that presents much better OOD detection performance and similar results in the other metrics.**_
> >
> > ############################################################
> >
> > ############################################################
> >
> > **Hence, we do not see good reasons to accept the paper.**
> >
> > #############################
> >
> > #############################
> >
> > _**One last word: we only suggested combining IsoMax+ with your approach because if this worked better than IsoMax+ regarding OOD detection performance, this would show true value to your work (remember, the OOD detection performance of your work does not appear to be consistently better than the one presented by a regular neural network), and we would possibly recommend accepting it. This is the reason. We tried to give you an opportunity to convincing us that your work deserve to be published. Considering this was not done, we see no substantial value in your work in the current form, as we firmly believe that Mixup combined with IsoMax+ is a much better solution (mainly considering the OOD detection performance).**_
> >
> > #############################
> >
> > #############################

---

> > ### Comment · Reviewer_ESue · 2021-11-26
> > **The pretty evident fatal flaw of the proposed approach: it does not improve the OOD detection performance regarding a trivially trained neural network!**
> >
> > **Tables 1, 2, 5, and 6 clearly show that the proposed approach does not increase the OOD detection when compared with a regular neural network (DNN).** In the four cases, we see two ties, the proposed approach outperforms DNN once, and in the other, it is outperformed by a trivially trained neural network. **This fact is incontestably evident based on what we see in those tables.**
> >
> > **The proposed approach is a regularization added to the usual SoftMax loss. The IsoMax+ loss is an alternative to the SoftMax loss (the log cross-entropy term) that clearly significantly increases the OOD detection performance in all cases when replaces the SoftMax loss (the log cross-entropy term).**
> >
> > **Therefore, it is pretty obvious that adding the proposed regularization term to the IsoMax+ loss would probably BENEFIT THE PROPOSED APPROACH BY FIXING ITS FATAL FLAW: no improvement in the OOD detection performance.** This obvious thing is what we suggested to the authors **TO HELP THEIR APPROACH ON ITS FATAL FLAW!** Rather than see the obvious, they lost their time trying to write lots of nonsense and misleading information about the IsoMax+ loss (read the papers and take your own conclusions).
> >
> > **_For the last time: The proposed approach and the IsoMax+ loss are complementary rather them competing approaches!_**
> >
> > _The IsoMax+ may be combined with the Mixup as well as it may replace the SoftMax loss (the log cross-entropy term) in the proposed approach to be combined with Mix-MaxEnt. This is so clearly obvious! This is why we recommended this in the first place. We even mention that this could provide state-of-the-art results across the board._
> >
> > In summary, we did not suggest combining both approaches to help IsoMax+ loss, as **it may be combined with Mixup regularization to great results across the board (i.e., improved accuracy, domain shift, uncertainty estimation, and OOD detection performance).** We suggested to combining the Mix-MaxEnt (a regularization term) to the IsoMax+ (a loss that replaces the cross-entropy term that is recognized for producing poor OOD detection performance) **TO SAVE THE PROPOSED APPROACH FROM BEING COMPLETELY INEFFECTIVE IN TERMS OF IMPROVING THE OOD DETECTION PERFORMANCE RELATIVE TO A TRIVIALLY TRAINED NEURAL NETWORK!**

---

### Official Review · Reviewer_L8Yz · 2021-11-03

**Correctness:** 4
**Technical Novelty And Significance:** 2
**Empirical Novelty And Significance:** 3
**Recommendation:** 5
**Confidence:** 3

**Main Review:**

The main advantage of this approach is simpleness. The proposed algorithm can be easily used to modify the training. My main concern is the significance of the empirical result.

1. Seems in the compared baselines, mixup tends to be the strongest one, which is quite surprising as mixup is not an algorithm specially designed for robustness and uncertainty. I am not an expert in this area and I am concerned about whether the current SOTA approaches are compared.

2. Cifar-100 is still a kind of a small dataset. For this empirical paper, I would recommend conducting larger-scale experiment on Imagenet with some more challenging natural distribution shifts.

Minor:
1. What if we mix images from 3 or even more classes? Why do we need to use beta distribution other than others to decide the weight?




**Summary Of The Paper:**

This paper proposes a simple approach called mix-maxent to improve the robustness of neural networks. The main approach is to regularize the network to make an uncertain prediction when the data is a interploation of two images with different classes. Experiment shows the good empirical results.

**Summary Of The Review:**

Overall, the main advantage is the simpleness but my concern on the significance of the result making me feel this paper is slightly under the acceptance bar.

---

> ### Author Response · Authors · 2021-11-18
> **[Part 1/2] Thanks for your comments, we address your perplexities in two comments (references in second comment)**
>
> We thank the reviewer for appreciating the **simplicity** and empirical **effectiveness** of our method. Below we provide answers to all the questions asked by the reviewer.
>
> **Q1: “Mixup tends to be the strongest baseline, which is quite surprising as it is not an algorithm specifically designed for robustness and uncertainty. I am not an expert in this area and I am concerned about whether the current SOTA approaches are compared”**
>
> **Ans:** Please note that we do compare our approach with several most recent SOTA approaches (mentioned below) and perform very **extensive and thorough experiments**.
> Though **recent papers on OOD have largely ignored mixup** as one of their baselines, mixup has been shown to **improve robustness** [1,2,3] and improved **calibration** as a result of the counterbalance between the usual overconfidence of neural networks and the ability of mixup to reduce the confidence [3]. We use mixup as an additional **strong baseline**.
>
> To summarize, while comparing our approach with the recent OOD papers published at top venues, we also **create new strong baselines** for OOD such as:
> - mixup [4];
> - DNN-SN [5] and DNN-SRN [6], following the observations in SNGP that showed normalisation techniques to improve uncertainty estimation properties
>
> The recent OOD works we compare our approach with are:
> - **SNGP** [7], often considered one of the most effective single-model approaches.
> - **DUQ** [8], as much as possible as we found it hard to make DUQ work on large experiments mostly because of training instabilities and centroid collapses.
> - **Deep Ensemble** [9] (often used as the gold standard);
> - **KFAC-LLLA** [10] (an incredibly effective Bayesian method).
>
> We mention most of these observations in **Section 5 and Appendix C, Code section**.
> We **omitted** methods simple to implement (like MC-Dropout) that are known to have **suboptimal performance** or that **assume knowledge of the OOD data** at training time.
>
> We would gently ask the reviewer to pick any of the above OOD papers (and these other papers [8, 11,12,13,14,15, 16] ) to realise the choice and amounts of baselines is either comparable or **superior** to most of the papers cited. Differently from most of these papers, we also compare using multiple uncertainty estimation tasks using several datasets and architectures. We trained each of the baselines from scratch (where possible) and cross-validated all the hyperparameters to allow them to produce the **best performance possible**.
>
> **Q2: CIFAR-100 is still a kind of small dataset. For this empirical paper, I would recommend conducting large-scale experiment on Imagent with some challenging natural distribution shifts.**
>
> **Ans:** We would like to highlight that **most recent approaches do not even perform experiments on CIFAR10/CIFAR100** on both **ResNet and WideResNet** [4,7,8,10]. Few of them **do not even scale properly to CIFAR100**. Most of the papers only report results for **one architecture type**, without showing state-of-the-art accuracy nor performing evaluations on **calibration, data-shift or out-of-distribution detection**.
> There are several recent papers where only **small scale** (such as MNIST, Fashion MNIST, and SVHN, or only on subsets of classes of CIFAR-10) experiments are used [8, 11,12,13,14,15, 16] (just to mention a few!)
>
> In this work, we train on CIFAR 10 and CIFAR 100 from scratch and cross-validate all the hyperparameters fairly for all our baselines in order to make them perform to their best and to evaluate all the models for accuracy, calibration, data-shift and OOD detection. This already required **extreme amounts of computation** (as reported in **Appendix C**). For our submission we did not have enough resources to do the same at ImageNet scale, but we will consider it for future experiments. However, please note that it **might not be possible to scale all the existing published works to ImageNet as we found it difficult to make them work even on CIFAR100**.
>
> **Q3: Additional experiments with more natural distribution shifts**
>
> **Ans:** Though the experiments presented in the paper contain **15 diverse types of corruptions at 5 intensity levels** (CIFAR-C datasets), we perform **additional experiments on CIFAR10.1 and CIFAR10.2** datasets [17,18] using WideResNet. Note, this dataset contains **real-world distribution shifts** and in this one as well, **our approach outperforms** extremely competitive baselines. See:
> https://tinyurl.com/mynhp2bx
>
> We could not find any other CIFAR data-shift datasets, but we will be more than **glad to execute any data-shift experiment the reviewer might suggest**, in case they are aware of any other data-shift test set for CIFAR-10/100.

---

> > ### Author Response · Authors · 2021-11-18
> > **[Part 2/2] Thanks for your comments, we address your perplexities in two comments (references in second comment)**
> >
> > Minor:
> >
> > **Q4: “What if we combine 3 or more images/classes?”**
> >
> > **Ans:**  Yes, it is possible to combine more than 2 images. We did not perform this experiment as mixing two was already providing good results and the proper analysis already required us to perform thousands of experiments. Naively mixing more than k images would require dealing with (k-1) hyper-parameters and might not result in improved performance, therefore, a scalable and efficient approach should be devised for the same. We leave this for future work.
> >
> > **Q5: “Why use the beta distribution?”**
> >
> > **Ans:** Any distribution with the following properties would work for Mix-MaxEnt:
> > - unimodal distribution with support [0,1]
> > - we should be able to control the position and the curvature of the mode, so that the interpolation factor can be close to 0.5
> >
> > For instance, we could use a truncated Gaussian centered in 0.5, but we don’t expect we would notice significant differences from using a Beta distribution. We used the Beta distribution because it is available in most machine learning libraries (while truncated Gaussians are not), it potentially allows to model skewed distributions and it’s commonly used in literature.
> >
> > We hope the reviewer will consider our response useful and exhaustive. Please, let us know if anything is unclear or needs further elaboration.
> >
> > Refs:
> >
> > [1] On Mixup Training: Improved Calibration and Predictive Uncertainty for Deep Neural Networks Sunil Thulasidasan, Gopinath Chennupati, Jeff Bilmes, Tanmoy Bhattacharya, Sarah Michalak
> >
> > [2] How Does Mixup Help With Robustness and Generalization? Linjun Zhang, Zhun Deng, Kenji Kawaguchi, Amirata Ghorbani, James Zou
> >
> > [3] Combining Ensembles and Data Augmentation Can Harm Your Calibration Yeming Wen, Ghassen Jerfel, Rafael Muller, Michael W Dusenberry, Jasper Snoek, Balaji Lakshminarayanan, Dustin Tran
> >
> > [4] mixup: Beyond empirical risk minimization, Hongyi Zhang, Moustapha Cisse, Yann N. Dauphin, and David Lopez-Paz.
> >
> > [5] Spectral norm regularization for improving the generalizability of deep learning, Yuichi Yoshida and Takeru Miyato
> >
> > [6] Stable rank normalization for improved generalization in neural networks and GANs, A Sanyal, P H S Torr, and P K Dokania.
> >
> > [7] Simple and principled uncertainty estimation with deterministic deep learning via distance awareness, Jeremiah Zhe Liu, Zi Lin, Shreyas Padhy, Dustin Tran, Tania Bedrax-Weiss, and Balaji Lakshminarayanan.
> >
> > [8] Uncertainty estimation using a single deep deterministic neural network,Joost van Amersfoort, Lewis Smith, Yee Whye Teh, and Yarin Gal Gal.
> >
> > [9] Simple and scalable predictive uncertainty estimation using deep ensembles. Balaji Lakshminarayanan, Alexander Pritzel, and Charles Blundell.
> >
> > [10] Being bayesian, even just a bit, fixes overconfidence in ReLU networks. Agustinus Kristiadi, Matthias Hein, and Philipp Hennig.
> >
> > [11] On Feature Collapse and Deep Kernel Learning for Single Forward Pass Uncertainty, Joost van Amersfoort, Lewis Smith, Andrew Jesson, Oscar Key, Yarin Gal.
> >
> > [12] Class Anchor Clustering: A Loss for Distance-based Open Set Recognition, Dimity Miller, Niko Sunderhauf, Michael Milford, Feras Dayoub
> >
> > [13] Out-of-distribution detection in classifiers via generation. Sachin Vernekar, Ashish Gaurav, Vahda Abdelzad, Taylor Denouden, Rick Salay, Krysztof Czarnecki
> >
> > [14] Likelihood regret: an out-of-distribution detection score for variational auto-encoder, Zhisheng Xiao, Qing Yan, Yali Amit
> >
> > [15] Do generative models know what they don’t know, Eric Nalisnick, Akihiro Matsukawa, Yee Whye Teh, Dilan Gorur, Balaji Lakshminarayanan
> >
> > [16] Evidential Deep Learning to Quantify Classification Uncertainty, Murat Sensoy, Lance Kaplan, Melih Kandemir
> >
> > [17] Do CIFAR-10 Classifiers Generalize to CIFAR-10?, Benjamin Recht and Rebecca Roelofs and Ludwig Schmidt and Vaishaal Shankar
> >
> > [18] Harder or Different? A Closer Look at Distribution Shift in Dataset Reproduction" by Shangyun Lu, Bradley Nott, Aaron Olson, Alberto Todeschini, Puya Vahabi, Yair Carmon and Ludwig Schmidt

---

> > > ### Author Response · Authors · 2021-11-28
> > > **Follow-up with reviewer L8Yz**
> > >
> > > We would like to thank the reviewer once again for their reviews.
> > >
> > > We hope that our replies to the reviewer's concerns were satisfying. We wanted to reach out to ask if there were any further questions/comments from the reviewer. We would be happy to reply.
> > >
> > > Thank you

---

### Official Review · Reviewer_qdpu · 2021-11-07

**Correctness:** 3
**Technical Novelty And Significance:** 3
**Empirical Novelty And Significance:** 3
**Recommendation:** 6
**Confidence:** 3

**Main Review:**

Strengths:
- The paper is clear and easy to understand.
- The proposed technique is simple and can be easily integrated into other NN architecture.
- The performance of the proposed model is competitive compared to the baselines.
- The author conducts experiments on multiple use-cases with multiple datasets.

Weakness:
I have a few comments, concerns, and questions regarding the paper:
1) The authors start the paper with an analysis of some of the observations they found on CIFAR and SVHN data. How much of these observations are applicable to other datasets?
2) The authors made an observation that the OOD and data-shifted samples are mapped by NN to the hypersphere S. Since we are working on high dimensional feature space, a hypersphere is likely to cover more volumes than the area where the data lies. Therefore, this observation should not come as a surprise, shouldn't it?
3) The technique focuses on developing proxies for OOD samples that lie between different clusters of data. However, most of the OOD samples should lie in the area that is not in between two clusters (at least based on the areas). Has the author considered this case? For example, create experiments on this type of OOD and see if the proposed approach could cope with it.
4) The closest previous work to this paper is the DUQ paper. However, the author did not properly compare the model with the DUQ (with code base as the reason). All the DUQ numbers mentioned in the paper are based on directly taking the numbers from the DUQ paper. This may cause problems in assessing the fairness of the comparisons, since the setup in the DUQ paper may be a little different from the experiment in this paper.
5) The comparison of several metrics for evaluating OOD in section 5.1.3 is a bit confusing. What "metric" is used to evaluate the performance of the metrics so the author can say that one is better than the others?
6) As presented by the author, the methods create a high-entropy barrier between two class labels. What if in the classification problems, the transition from one label to another is more or less smooth. For example, in the case of digits 1 and 7 in handwriting recognition. Is applying the proposed approach counter-productive in these cases?
7) Presentation. The tables are too small to clearly see. Some of the results in the main paper refer to a table in the appendix, which makes it a bit inconvenient to read.

**Summary Of The Paper:**

The author proposed a new method for improving uncertainty estimation in deep neural network output. The idea is similar to the mixup paper, but rather than augmenting the data by making a new sample interpolated from two samples having the same label; it creates a new sample that interpolates two samples with different labels. The hope is that these new samples serve as proxies to out-of-distribution data. A new loss function is then proposed by the author that augments the standard cross-entropy loss with an entropy regularization term that encourages the prediction of the new augmented samples to be as uncertain as possible (maximizing the entropy of the prediction). The author then performs experiments to demonstrate the effectiveness of the proposed approach.

**Summary Of The Review:**

The paper proposes an interesting approach in uncertainty estimation for deep neural networks. The paper has some weaknesses that I hope the author can fix it soon. For now, I recommend a weak accept.

---

> ### Author Response · Authors · 2021-11-18
> **[Part 1/2] Thanks for your observations, we address your questions in two comments**
>
> We thank the reviewer for praising the **clarity** of our paper, providing valuable feedback and appreciating the **simplicity** and the **superior performance** of our method in **several evaluation settings** and across **multiple architectures**.
>
> Below we provide answers with additional experiments to the questions asked.
>
> **Q1: The authors start the paper with an analysis of some of the observations they found on CIFAR and SVHN DATA. How much do the observations generalise beyond CIFAR/SVHN?**
>
> **Ans:** We added more datasets to the motivating experiment in Section 3 and show that the **conclusions remain the same**, see figures at:
>
> https://tinyurl.com/4b7jrd3k
>
> https://tinyurl.com/ws8eadc
>
> This means that our observations do generalise to other datasets. The additional datasets we consider are **Tiny-ImageNet, FakeData (a dataset made by random gaussian noise) and CelebA**. We also report the **boxplots** associated with each histogram, showing the means and quantiles as requested. Only in the case of Tiny-ImageNet, 2 points fall outside the hypersphere S. We point out that the classes of CIFAR-10 and CIFAR-100 significantly overlap with Tiny-ImageNet classes.
> While providing experiments and metrics on additional datasets have further supported our conclusions in Section 3, we would like to mention that it is a very common practice in the literature to use small scale experiments (most of the times much smaller than ours) to build intuition and then provide extensive experiments on large scale datasets to show the effectiveness of the proposed approach. Therefore, we believe that the motivating example above should now be sufficient.
>
> **Q2: Since we are working on high dimensional feature space, a hypersphere is likely to cover more volumes than the area where the data lies. Therefore, this observation should not come as a surprise, shouldn’t it?**
>
> **Ans:** It was not obvious to us that the **OOD and shifted-samples are mapped within the tightest hypersphere that fits the in-distribution data**. Note, the OOD samples are completely different from the in-distribution ones. A common notion is that the overconfidence of NNs should be attributed to the possibility that the networks embed inputs in regions of the space **arbitrarily far away** from the train data [1,2]. Our observation, however, implies that we might **not have to focus on such arbitrarily large space**. And this is exactly what the **effectiveness of our approach** Mix-MaxEnt supports.
> Additionally, it has also been discussed in the literature that the trained networks generally are low Lipschitz [3], therefore, change in the input should not change the output much. However, most of these previous works have focused on small additive perturbations to show this. We are not aware of any work that showed something similar for OOD and shifted-samples.
>
> **Q3: The technique focuses on developing proxies for OOD samples that lie between different clusters of data. However, most of the OOD samples should lie in the area that is not in between two clusters (at least based on the areas). Has the author considered this case? For example, create experiments on this type of OOD and see if the proposed approach could cope with it.**
>
> **Ans:** Our experiments do contain situations where the OOD do not lie in between two clusters and our approach performs very well on these situations as well.
> To show this, we fit a **hyper-ellipsoid E** (Gaussian covariance matrix) using only the in-distribution data (CIFAR 10) and show that, in the case of vanilla DNN, following percentage of the OOD dataset fall outside the E:
> C100: 3.6%
> SVHN: 41.46%
> This shows that the **experiments we have performed do contain enough variety where many data points lie outside the pair of clusters** (e.g., at least 41.46% of SVHN would lie outside), and many of them might lie within the pairs of clusters.
> Please note that there is no trivial approach to obtain meaningful OOD regions without having access to the OOD dataset, hence our approach uses interpolation of in-distribution clusters to regularize a network. In existing benchmarks, this extremely simple approach provides highly promising results.
>
> Refs:
>
> [1] Why ReLU networks yield high-confidence predictions far away from the training data and how to mitigate the problem, Matthias Hein, Maksym Andriushchenko, Julian Bitterwolf
>
> [2] Being Bayesian, Even Just a Bit, Fixes Overconfidence in ReLU Networks, Agustinus Kristiadi, Matthias Hein, Philipp Hennig
>
> [3] Stable Rank Normalization for Improved Generalization in Neural Networks and GANs. Amartya Sanyal, Philip H.S. Torr, Puneet K. Dokania

---

> > ### Author Response · Authors · 2021-11-18
> > **[Part 2/2] Thanks for your observations, we address your questions in two comments**
> >
> > **Q4: The closest previous work to this paper is the DUQ paper. However, the author did not properly compare the model with the DUQ (with code base as the reason). This may cause problems in assessing the fairness of the comparisons, since the setup in the DUQ paper may be a little different from the experiments in the paper.**
> >
> > **Ans:** We **do compare with DUQ** in experiments where it was **feasible to make DUQ converge** (please check Table 2). However, as mentioned in our paper (page 6, “Code base” paragraph), **DUQ authors did not perform large scale experiments** on datasets with more than 10 classes.
> >
> > - We **spent several days (weeks, precisely) trying to make DUQ converge on CIFAR-100** to the official code base, modified the hyperparameters, but could not manage to make the method work. It would **not converge for any seed**.
> > - We analysed the reasons and found out that the method did not manage to learn distinct cluster centroids for more than a small minority of classes (about 12-20 classes), all the other centroids collapsed.
> > - We believe that the authors of SNGP could make it converge by performing some fundamental changes to the methodology itself, to the point that we would not consider it DUQ anymore.
> > - As for **CIFAR-10**, we managed to achieve convergence for a few seeds, but in most cases the training resulted to be **unstable and underperforming**. For this reason, we preferred to report results from the SNGP paper.
> >
> > Please note that we compare our approach with **other stronger and more stable baselines** as well, so we believe the **lack of comparison with DUQ** to be **negligible**.
> >
> > **Q5: The comparison of several metrics for evaluating OOD in section 5.1.3 is a bit confusing. What “metric” is used to evaluate the performance of the metrics so the author can say that one is better than the others?**
> >
> > **Ans:** We tried several existing metrics and in **cases** in which a metric was **strictly superior **(e.g. for mixup, the entropy is always better) we reported that one, in the other cases there are **no significant differences** on average with very minor differences. There was no solid justification to prefer one over the other in these cases as the differences were insignificant. We have provided performances of all these metrics in **Tables 8 and 9  in Appendix E**.
> >
> > In the same Appendix, we also perform an important analysis showing that **GMM-based metrics are highly unstable** in many cases due to numerical instability. Therefore, approaches using GMM-based metrics should also provide large scale experiments in order to show the efficacy.
> >
> > **Q6: “What if the transition between labels is smooth? For example, in the case of digits 1 and 7 in handwriting recognition. Is applying the proposed approach counter-productive in these cases?**
> >
> > **Ans:** Please note that our loss contains both the **entropy** (on interpolated samples) and the **log-likelihood** (on clean samples) terms. Therefore, we do not expect it to harm the accuracy on the test set. In your specific example, the **clusters** of 1 and 7 will probably be very **close** in the latent space of a NN trained with cross-entropy, and they also might overlap. Our method has been shown to **improve cluster separation** (see the Fisher index in Section 5.2.1). So we might expect the **accuracy** on these two classes to improve as a consequence. There might still be **ambiguous samples** that could lie between the two clusters, but in that area our regulariser would impose a **high entropy** between the two class labels, and hence would correctly represent this ambiguity. Hence, we would expect our regulariser to both improve accuracy and uncertainty estimation in this case.
> >
> > **Q7: “The tables are small to clearly see. Some of the tables are in the appendix, which makes it a bit inconvenient to read”**
> >
> > **Ans:** We present results on several architectures and evaluation settings for multiple seeds, putting some of the tables in the appendix was necessary due to the space constraints. We will improve the formatting of the tables to make them more readable.

---

> > > ### Comment · Reviewer_qdpu · 2021-11-26
> > > **Thanks for the feedback**
> > >
> > > Thank you for the author for providing detailed feedbacks and answering the concerns I have.

---

### Official Review · Reviewer_iva2 · 2021-11-28

**Correctness:** 2
**Technical Novelty And Significance:** 2
**Empirical Novelty And Significance:** 2
**Recommendation:** 5
**Confidence:** 4

**Main Review:**

Strengths:
1. Empirical results demonstrate meaningfuly improved accuracy in both clean and (non-adversarially) corrupted data compared to the baseline methods.
2. The method can be applied without modifying the network architecture, and it seems to perform well without being sensitive to HP tuning.

Weaknesses:
1. Limited evaluation to support claims ("Mix-MaxEnt consistently provides much improved classification accuracy, better calibrated probabilities for in-distribution data, and reliable uncertainty estimates when exposed to situations involving domain-shift and out-of-distribution samples"):

1.a. In the context of classifier calibration, the comparison to Mixup seems to be limited given the similarity between the methods and the calibration results in the paper. Specifically, ECE/AdaECE results of C100-C and C10-C for Wide-ResNet (Tables 1 and 2) show conflicting results (this appears to persist in ResNet50 experiments as well). Moreover, the authors report the average results over all corruption types and intensities, however, Figure-10 shows that Mixup ECE is more consistent when facing severe corruption (>3), this could suggest a tradeoff between the two variants. Finally, the ablation study should be improved to clearly emphasize the importance of the proposed modifications including a clear summary of the empirical results. I would suggest to dedicate a section of the paper for a deep analysis comparing the two methods instead of scattering results and discussions in different sections.

1.b. Evaluation is limited to ResNet architectures and small image classification datasets. As a general regularization method, the authors should provide a wider range of architectures and datasets to demonstrate the method is broadly applicable (at least in the context of supporting better accuracy/calibration claims).

2. OOD results are not very impressive compared to the baselines (e.g., AUROC is comparable to vanilla DNN in Tables 1 & 2). The authors should also justify why they do not compare against other methods that have trained on proxy distributions to increase uncertainty on OOD [e.g., Dan Hendrycks, Mantas Mazeika, and Thomas Dietterich. Deep anomaly detection with outlier
exposure, 2018, Weitang Liu, Xiaoyun Wang, John D. Owens, and Yixuan Li. Energy-based out-of-distribution
detection, 2020 ]. The authors also do not provide a sufficient explanation to avoid comparison to Mukhoti, Jishnu, et al. "Calibrating deep neural networks using focal loss 2020.

3. Some aspects of the method are overlooked. For instance, the importance/impact of regularization strength is not discussed in the paper (i.e., the balance between CE and ERL, there is an implicit balance when setting a specific ratio of mixed and clean data in a batch). Also, the use of beta distribution is not properly motivated when preferring $\lambda$ values that are close to 0.5.

General remark:
The paper can improve segnificantly in terms of readability, flow and focus. For example, some paragraphs contain highly technical details or remarks that can be moved to the appendix. Moreover, some important results are deferred to the appendix and presented in an inconvinient way and with limited discussion (e.g., missing a summary for ablation study results, appendix G. tables are missing baseline results etc.). In contrast, some content that made it to the main paper is arguably beneficial to readers (e.g., SR,SRN given the results are comparable to vanilla DNN).

**Summary Of The Paper:**

The authors propose a method for regularizing DNN classifiers during training by encouraging the classifier to have high entropy on embeddings of mixed samples (from two different classes) as a proxy for the unknown portion of the data distribution (i.e., the portion of the in-distribution that is not represented by the training samples). The objective of the method is to produce better deterministic uncertainty estimates without compromising accuracy.

The suggested approach is similar to the previously proposed Mixup strategy, except regularization is explicitly applied to the mixed samples and labels while the optimization objective of clean samples is unchanged. Moreover, the distribution of $\lambda$ is set to produce values close to ~0.5 (i.e., mixture farther away from the class clusters).

**Summary Of The Review:**

The general idea, insight from empirical analysis and results appear promising. However, important aspects of this work are not properly addressed to support the main claims (limited evaluation, comparison to relevant work, general writing quality). The authors are encouredged to consider revising the paper accordingly and re-submit.

---

> ### Author Response · Authors · 2021-11-29
> **[Part 1/2] Thanks for your encouraging comments, we address your questions below**
>
> Thank you for taking the time to read our paper. We appreciate that the reviewer found the improved accuracy (on both clean and corrupted data), simplicity (no need to modify the architecture) and the stability of our approach in terms of the choice of hyper-parameter as the core strengths of our approach.
>
> There are a few questions raised by the reviewer that we address below to our best. We are confident that our replies will be satisfactory. We slightly restructure the question formulation so that we can provide more informative answers.
>
> **No one method always outperforms others in all the five aspects -- clean-data accuracy/calibration, corrupted-data accuracy/calibration, and OOD**
>
> We would like to emphasize that there is no existing method (including Deep Ensembles (DE)) that outperforms all other approaches in all the five aspects in all the experiments presented in the paper. *However, if we were to choose one that is performing better in most of the cases then it is evident from our experiments that Mix-MaxEnt outperforms others by a large margin.*
>
> We have been very clear about this, please check *section 5.1.4* of our paper. To summarize, in the case of WideResNet (widely used architecture for OOD experiments in all recent papers), **out of 20 different evaluations** presented in Tables 1 and 2 of the main paper
>
> - Mix-MaxEnt outperformed **all** the approaches (including DE) in 11 of them
> - Mix-MaxEnt outperformed **all** the single model approaches in 14 scenarios.
> - **No** existing single deterministic baseline outperformed all other approaches in more than **2** scenarios.
>
> Similar conclusions can be drawn for ResNet (please refer Tables 5 and 6).
>
> **Empirical comparison between Mixup and Mix-MaxEnt**
>
> Before we provide empirical comparisons we would like to emphasize that there are fundamental differences between the two. Please check the updated paragraph (Mix-MaxEnt vs mixup) in Section 4.
>
> *Regarding empirical comparison, the summary is*
>
> - WideResNet: Mix-MaxEnt outperformed Mixup in **18 evaluations out of 20** in Tables 1 and 2.
> - ResNet: Mix-MaxEnt outperforms Mixup in **17 evaluations out of 20** in Tables 5 and 6.
>
> *Regarding the ECE/AdaECE conflicting results*
>
> Firstly, we recall that when the AdaECE and ECE metrics disagree significantly, this indicates the ECE metric is affected by *estimation bias* due to the binning procedure needed to compute it.
> - On all in-distribution experiments (clean data), our method has better AdaECE than Mixup, and the ECE metric is similar except in one case, in which the ECE on Mixup is better but the AdaECE is more than double the ECE.
> - Regarding data-shifted inputs, it is true that in the case of CIFAR10-C on both the architectures mixup shows superior calibration, yet (1) our method still shows significantly better accuracy and, (2) significantly better OOD detection performance.
>
> We do not have a clear answer to why the calibration of Mix-MaxEnt is better than mixup for CIFAR100-C and not CIFAR10-C as there might be various factors playing crucial roles that we (even as a community) do not understand that well, for example,
> - how does the task complexity, model capacity, and dataset impact calibration? We observed that mixup performed better for CIFAR 10-C, not for CIFAR100-C.
> - what kind of inductive biases are needed so that calibration is always improved?
> - most importantly, how does improving calibration on clean-data impact OOD and data-shift performances?
>
> We would suggest the reviewer look at all the evaluations together (accuracy, clean data calibration, shift-data calibration, and OOD) to pick the best performing method as there is no one method that outperforms all others in all the evaluations.
>
> **Does Figure-10 show that the Mixup ECE more consistent when facing severe corruption (>3)?**
>
> Note, the **type of corruption matters**. For example, WideResNet CIFAR10-C (Figure 10), we observe that
> - in the case of Gaussian noise the ECE of mixup is always better than others. It makes sense as mixup end-up perturbing inputs in their vicinity, therefore, should be robust to similar perturbations.
> - in the case of “snow” and “brightness” corruptions, MixMax-Ent is always better than all others even when the corruption level is >3.
> - in the case of “frost” and “defocus” MixMax-Ent is better than others until corruption level 4.
> - negative result for Mix-MaxEnt -- it performed very poorly in the case of “fog” corruption
> - negative result for Mixup -- it performed very poorly in the case of “brightness” corruption. Note, in this case, Mix-MaxEnt outperformed all others.
>
> There are several such observations here. **We find these plots extremely informative and an important addition to our paper** that would provoke thoughts for future research.
>
> Having said that, while we show more elaborated plots, we do follow very standard practice to show the ECE over average corruptions on all the 75 datasets of CIFAR-10/100-C for comparisons.

---

> > ### Author Response · Authors · 2021-11-29
> > **[Part 2/2] Answers to the remaining questions/comments**
> >
> > *A dedicated section to compare different aspects of Mixup and Mix-MaxEnt*
> >
> > We do provide detailed analysis of the different factors involved in Mix-MaxEnt (please see **Section 5.2.3**). We can elaborate the section if space permits. Or else, we will put a more detailed section in the Appendix to compare the two.
> >
> > *Evaluation is limited to ResNet architectures and small image classification datasets.*
> >
> > We provide results and analysis on WideResNet and ResNets using CIFAR10 and CIFAR100. While we can attempt more large-scale experiments (say ImageNet) as our approach (and others like DNN, DE etc) is clearly scalable, we are not sure how scalable recent approaches would be. For example:
> > - DUQ (van Amersfoort et al., 2020) -- Unfortunately, it was not easy to scale this approach to C100 without modifying the approach itself.
> >
> > In this work we do not attempt to go beyond CIFAR100 as making sure that we have the best behaviour of all the baselines we are comparing with (hyper-parameter cross-validation etc.) already required huge effort with thousands of experiments. Also, please note that the most recent papers only show results on CIFAR10, MNIST, FashionMNIST, and SVHN.
> >
> > Having said that, we will try ImageNet and compare with SOTA that are scalable, however, we can not promise anything at this stage as the time and the resources are limited.
> >
> > **Why we do not compare against other methods that have trained on proxy distributions to increase uncertainty on OOD [e.g., Dan Hendrycks, Mantas Mazeika, and Thomas Dietterich. Deep anomaly detection with outlier exposure, 2018, Weitang Liu, Xiaoyun Wang, John D. Owens, and Yixuan Li. Energy-based out-of-distribution detection, 2020 ].**
> >
> > Please note that our approach (and the ones we compare with such as DE, SNGP, KFAC-LLA etc.) **do not require any external real-world OOD dataset during training**. Ours synthesizes the OOD dataset using the convex combination of in-distribution samples itself. However, both the papers you mentioned require external real-world OOD dataset during training itself. Therefore, it belongs to a different category of approaches than ours.
> >
> > **The authors also do not provide a sufficient explanation to avoid comparison to Mukhoti, Jishnu, et al. "Calibrating deep neural networks using focal loss 2020**
> >
> > There were two main reasons we did not compare with Jishnu et al., 2020 (though we used their AdaECE)
> > - We already had too many baselines to compare with and had dedicated a lot of efforts on several SOTA baselines that most recent papers compare with.
> > - Also, we had observed that our approach already outperformed focal loss for ResNet50 CIFAR10 experiments in terms of both accuracy and ECE.
> >
> > However, as pointed out by the reviewer, there is no specific reason that would stop us from creating a new baseline (focal loss) and compare our results with it. We have already set-up the experiments for the same and will update the tables with additional baseline in our final draft.
> >
> > **Importance/impact of regularization strength is not discussed in the paper (i.e., the balance between CE and ERL, there is an implicit balance when setting a specific ratio of mixed and clean data in a batch)**
> >
> > We observed that our approach was not too sensitive to adding an importance multiplicative term, therefore, chose to ignore it (use 1). However, we could definitely provide the ablation study to show the impact of varying this term and the ratio of mixed and clean data. Obviously, this ablation will result in a setting with performance that is at least as good as the performance of the model presented in the main paper.
> >
> >
> > **The use of beta distribution is not properly motivated**
> >
> > Any distribution with the following properties would work for Mix-MaxEnt:
> > - unimodal distribution with support [0,1]
> > - we should be able to control the position and the curvature of the mode, so that the interpolation factor can be close to 0.5
> >
> > For instance, we could use a truncated Gaussian centered in 0.5, but we don’t expect we would notice significant differences from using a Beta distribution. We used the Beta distribution because it is available in most machine learning libraries (while truncated Gaussians are not), it potentially allows to model skewed distributions and it’s commonly used in literature.
> >
> > We hope that we have answered all the questions/comments by the reviewer. Please let us know if you seek further clarifications.

---

> ### Comment · Reviewer_iva2 · 2021-11-29
> **Rebuttal response**
>
> After considering the authors' response, I maintain my original position: the paper shows promise but it does not meet the quality threshold in its current state (keeping the score at 5), see detailed response below.
>
> Performance as an OOD method
> It is understood that it is unlikely to have a single method to outperform others in every aspect. In this case, the argument stems from the limited discussion, I would suggest clarifying this point in the abstract as well as it can be misleading (i.e., this method is not producing "consistently reliable" uncertainty estimation for OOD that are better than existing methods --- even ones that do not require training the model or using OOD data).
>
> Performance as a calibration method:
> ECE/AdaECE conflict refers to conflicting results between cifar10 and cifar100 scenarios for WideResNet while ResNet50 results show a similar pattern. This is a strong result which in my opinion undermines the premise that the proposed method is better than Mixup (as a calibration method). The authors need to cover more scenarios to refute this evidence.
>
> More evaluation:
> I understand that reviews can always ask for more evaluation than provided. However, in this case, the evaluation is limited to a single architecture (ResNet and WideResNet do not really count as different architectures) and a few datasets. Moreover, some of the baselines (e.g., SNGP) provide results on other datasets and models which is not clear why the authors choose to omit.
>
> Comparison to MixUp:
> I understand the differences in the proposed method and Mixup. The authors should be extra diligent given the similarity between the methods to support the changes made are crucial and are directly linked to the improved results. For instance, the authors can show that using a more relaxed beta distribution (e.g., $\alpha=\beta=0.95$ such that $\lambda$ has a higher probability to get values near 0.5) does not reproduce results with Mixup alone to strengthen the novelty of this work. Section 5.2.3 is not really focused on separating Mixup and the proposed method. Finally, the empirical results should accompany this section in a way that is convenient for the reader to process (I had to jump between different tables in the appendix and the main paper to make sense of the results). This is true for tables 1 2 5 6 as well. The authors should find a way to pack the results together and allow the reader to draw general conclusions when considering multiple scenarios.
>
> Corruption type & intensity matters:
> This point is clear to me. However, I find the discussion in the paper to be lacking. it is OK to average the results for tables but the authors cannot ignore patterns that raise from their own analysis when reviewing the results in the paper.
>
> Baselines:
> Too many baselines is not a good answer given that some of the existing baselines do not make sense to me in light of the results. I understand that OE and ES are using a different proxy for the OOD distribution from the one motivated here. Mining for OOD samples is not hard since the data is unlabeled and potentially unlimited, yet there are other implications such as implicit assumptions on the test time OOD distribution. The authors can also consider these methods as baselines to strengthen the observations at the base of the proposed method (i.e., showing a benefit to using mixed ID data vs OOD samples).
>
> Importance/impact of regularization, choice of mixer distribution:
> Please note that these key aspects are entirely missing from the paper, the comment suggests you include them in a revised paper.

---

> > ### Author Response · Authors · 2021-12-02
> > **[Part 1/2] Important clarifications about our statements**
> >
> > Thank you for your response and for spending time clarifying your position. We believe there are still some misunderstandings, therefore we make our thoughts more clear and further elaborate them below.
> >
> > **Q: This is a strong result which in my opinion undermines the premise that the proposed method is better than Mixup (as a calibration method). The authors need to cover more scenarios to refute this evidence.**
> >
> > We observe that **calibration methods should not be considered in a vacuum**. We do report the performance of several uncertainty estimation downstream tasks at the same time for a very important reason. The **OOD detection performance of Mixup is degraded even with respect to DNN**. This provides a crucial insight about the uncertainty estimation properties of Mixup.
> > - The literature has already evidenced that Mixup calibrates NNs because it simply **reduces the confidence for any input** due to the label-smoothing properties of its loss [1]: this **counterbalances the overconfidence of NNs**. If Mixup were applied to underconfident models, it would make them even more underconfident: the fact the overconfidence of NNs and the confidence decreasing effect of Mixup cancel each other does **not imply it produces better uncertainty estimates per se**
> > - Indeed the **OOD detection** performance is **degraded** exactly because since the network becomes **less confident also close to the in-distribution data**, it will be harder to distinguish IND and OOD data. This result is visible in Figure 7 in our paper: Mixup induces higher entropy both close and away from the class clusters.
> > - On the other hand, **Mix-MaxEnt** produces **higher entropy only in the region of the space between class clusters**, while it keeps low entropy close to class clusters. This is the reason why Mix-MaxEnt does **not exhibit worse OOD performance**.
> >
> > We thank the reviewer for pointing out that we must emphasize these observations much more clearly in our paper. We will update our draft accordingly.
> >
> > [1] Combining ensembles and data augmentation can harm your calibration, Wen et al.
> >
> > **Q: OE and ES are using a different proxy [...] Mining for OOD samples is not hard since the data is unlabeled and potentially unlimited [..] consider these methods as baselines [...]**
> > - **Our method makes no assumption on the OOD data**, it only leverages insights about how OOD/data-shifted inputs are embedded by closed-world softmax classifiers. As answered to other reviewers, these assumptions are theoretically reasonable, generalise to several other datasets we considered and do not depend on the choice of the OOD/data-shifted inputs considered.
> > - The **performance of OE/ES can widely vary depending on which training OOD distribution is used**. Selecting a priori the optimal one to use  without assuming knowledge of the test OOD datasets is not possible. Making choices based on knowledge of the test set inflates the evaluation metrics and **risks of overfitting to the specific evaluation setting**. Using part of the training OOD dataset as a validation set would provide little guarantees about the generalisation of the performance of these methods to the test set. While this can be suitable for ad-hoc applications for practitioners, it does **not allow to tackle the general formulation of the problem** we address, for which no assumption can be made on the OOD test set. Therefore we believe comparing these methods is like **comparing apples and oranges**. Furthermore, very different evaluation settings for OOD (like ours) **might require different training OOD sets for best performance in each evaluation setting**, because of the the underlying assumption about the test set each training OOD set implies
> > - Unlabeled data is easy to collect, but non-overlapping between the OOD data and IND data should be guaranteed, and for a fair evaluation the **non overlapping between the OOD training data and test data must be always guaranteed**. These properties are **not easy to check** without making strong assumptions or implying some form of (potentially expensive) “supervision”.
> > - To the best of our knowledge, **no analysis exists about whether these methods improve the accuracy/calibration on clean/data-shifted inputs**. Even if we overfitted the OOD performance to the test OOD datasets, this might not be beneficial for other tasks. The fact we consider all these tasks at the same time is not an unnecessary complication (as we discussed): the same uncertainty estimates extracted by a network must be useful for several downstream tasks. **Our paper is not an OOD detection paper only**: baselines should produce improvements in all evaluation settings.

---

> > > ### Author Response · Authors · 2021-12-02
> > > **[Part 2/2] Other observations**
> > >
> > > **Q: “For instance, the authors can show that using a more relaxed beta distribution (e.g., α=β=0.95 such that λ has a higher probability to get values near 0.5) does not reproduce results with Mixup alone to strengthen the novelty of this work.”**
> > >
> > > This type of analysis is one of the first preliminary analyses we performed when developing our method. We do have these results (one seed), and we present them at this link: https://tinyurl.com/5x956n76 . In these experiments we show there is **no way to get Mixup using sampling $\lambda \approx 0.5$ to achieve comparable performance to Mix-MaxEnt**. We also observe that the Mixup literature almost never uses values bigger than 0.3/0.4 due to the well known performance degradations observed. We have similar experiments ablating all the components of Mix-MaxEnt but on Mixup, we **will include them in the Appendix**. As suggested by the reviewer, it might make our points stronger to some readers and we will include them and discuss them thoroughly.
> > >
> > > **Q: Corruption type & intensity matters [...] the authors cannot ignore patterns that raise from their own analysis when reviewing the results in the paper.**
> > >
> > > As pointed out in our previous response, these patterns **can be inconsistent** depending on the corruptions chosen (we already discussed the calibration of Mixup in Part 1).  However, we agree adding the discussion we provided can be beneficial to the reader and will be included in the paper.
> > >
> > > **Q: “some of the baselines (e.g., SNGP) provide results on other datasets and models which is not clear why the authors choose to omit”**
> > >
> > > The SNGP paper was accepted only reporting results on models trained on CIFAR-10/100 (with corrupted variants) and using SVHN and CIFAR as OOD and doing experiments only on WideResNet. Our choice of models, baselines and datasets is **on par or superior to many peer-reviewed and accepted papers in the field of uncertainty estimation**. ResNets are the most popular architectures in computer vision. We are consider experimenting with other less popular CNNs (VGG, DenseNet) specifically.
> > >
> > > **Q: “the evaluation is limited to a single architecture (ResNet and WideResNet do not really count as different architectures)”**
> > >
> > > We respectfully disagree on this point. Despite being ResNet variants, these models exhibit **very different uncertainty estimation and data-shift behaviour across the baselines and all the evaluation settings**. We also observe that it is typical for uncertainty estimation literature to focus on one or very few architectures.
> > >
> > > **Q: “Importance/impact of regularization, choice of mixer distribution: Please note that these key aspects are entirely missing from the paper, the comment suggests you include them in a revised paper.”**
> > >
> > > - **importance/impact of regularization**: we believe this analysis to be trivial, given the cross-entropy term and the regulariser act on different regions of the input space, but we can obviously satisfy the reviewer's curiosity and include it.
> > > - **choice of mixer distribution**: we respectfully disagree, but we believe this analysis to be **not worth our time**. We have discussed which **properties a distribution must have** to apply our method and we **will include this discussion in the main paper**. We have already provided ablations in the paper showing that most of the improvements are given by the ability of sampling $\lambda$ in an interval centered around 0.5, and that some additional benefits can be achieved having small tails outside this interval. The beta distribution allows to easily specify a distribution with such properties and it is commonly available in libraries. Determining whether a truncated gaussian or a beta or any other similar distribution could be better is very likely insignificant, especially considering these distributions look very similar for the hyperparameters we use. We **do not expect to observe any statistically significant difference**. We also observe **no paper using Mixup we know performs this kind of analysis**.
> > >
> > > **Q: “The authors should find a way to pack the results together and allow the reader to draw general conclusions when considering multiple scenarios.”**
> > >
> > > The paper is dense with insights and empirical results. It is very difficult to pack everything in 9 pages, and different readers find different aspects important or interesting. There is no way to satisfy them all and include everything in the main paper. We prioritised including in the main paper analyses that mostly referred to why our methodology has been developed and what its effect is. We will try our best to squeeze further information in the main paper.
> > >
> > > We hope the reviewer will find our explanations more clear and useful. We are available to respond to any further questions.

---

### Decision · Program_Chairs · 2022-01-20

**Decision:**

Reject

**Comment:**

This paper proposes a new regularizer, based on entropy maximization of samples near the decision boundary, to improve the calibration of neural networks while maintaining their accuracy.

The method seems simple, sufficiently novel, and has promising results. However, based on the review process (described below), I feel the paper needs to significantly improve its evaluation and presentation before it can be accepted.

The review process summary:

* Two reviews were eventually weakly positive about the paper: without major concerns, but not enthusiastic.

* One review (L8Yz) was not sufficiently informative.

* One review (ESue) raised many points. I disagreed with most of these points, following the authors' discussion. However, a few points seemed valid, such as the not-so-impressive performance for OOD detection, which the authors did not address.

* I therefore asked for an additional review (iva2). The review concluded the paper is interesting and potentially useful, but requires another round of revision before it can be accepted, mainly because of its clarity and missing comparisons. I agree with these conclusions.